# PERSON-CENTRIC ANNOTATIONS OF LAION-400M: AUDITING BIAS AND ITS TRANSFER TO MODELS

**Leander Girrbach**[1,3]  **Stephan Alaniz**[2]  **Genevieve Smith**[4]
**Trevor Darrell**[4]  **Zeynep Akata**[1,3]

[1]Technical University of Munich, Munich Center for Machine Learning (MCML), MDSI
[2]LTCI, Télécom Paris, Institut Polytechnique de Paris, France
[3]Helmholtz Munich      [4]University of California, Berkeley

## ABSTRACT

Vision-language models trained on large-scale multimodal datasets show strong demographic biases, but the role of training data in producing these biases remains unclear. A major barrier has been the lack of demographic annotations in web-scale datasets such as LAION-400M. We address this gap by creating person-centric annotations for the full dataset, including over 276 million bounding boxes, perceived gender and race/ethnicity labels, and automatically generated captions. These annotations are produced through validated automatic labeling pipelines combining object detection, multimodal captioning, and finetuned classifiers. Using them, we uncover demographic imbalances and harmful associations, such as the disproportionate linking of men and individuals perceived as Black or Middle Eastern with crime-related and negative content. We also show that a linear fit predicts 60-70% of gender bias in CLIP and Stable Diffusion from direct co-occurrences in the data. Our resources establish the first large-scale empirical link between dataset composition and downstream model bias. Code is available here.

## 1 INTRODUCTION

To what extent are biases a direct consequence of the massive, uncurated pretraining data on which foundational vision and language models are built? To what extent is bias or amplification of bias from real-world settings a result of the model? While data imbalance is widely implicated (Wang et al., 2019; Hirota et al., 2024; Alabdulmohsin et al., 2024), the connection has largely remained an assumption rather than a measurement. A central contribution of our paper is to provide labels for the the LAION-400M dataset (Schuhmann et al., 2021) that enable researchers, for the first time, to test how well different measures of dataset bias predict model behavior.

Concretely, we infer 276,824,258 bounding boxes of person detections, $199,931,986$ automatically inferred perceived binary gender and race/ethnicity labels for detected people, and detailed captions for each detected person (see Fig. 1 for a detailed workflow). Having these annotations provides an empirical foundation to directly link dataset statistics to downstream model behavior, enabling precise queries about visually depicted groups and their co-occurrence with charged terms (e.g., "criminal"). Moreover, it offers a high-resolution characterization of human representation in web-scale multimodal data, with precise insights that so far could only be estimated from small subsets (Birhane et al., 2023; Friedrich et al., 2023). Finally, it allows to extract targeted subsets for applications related to person modeling (Zheng et al., 2022) and dataset rebalancing (Yang et al., 2020).

In summary, our contributions are: (1) We create extensive, high-quality annotations for the entire LAION-400M dataset, i.e. bounding boxes, perceived gender and race/ethnicity labels; (2) We train gender and race/ethnicity classifiers specifically designed to label perceived gender and race/ethnicity in noisy settings; (3) We analyze the demographic distribution of LAION-400M and find that men and individuals with perceived Middle Eastern or Black appearances are more strongly correlated than expected with crime-related content or negative sentiment; (4) We conduct a novel SAE-based analysis to identify themes associated with different intersectional identity groups; (5) We show that a linear fit predicts 60–70% of gender bias in CLIP and Stable Diffusion models trained on LAION-400M from direct co-occurrence of gender in images with demographic categories in text.

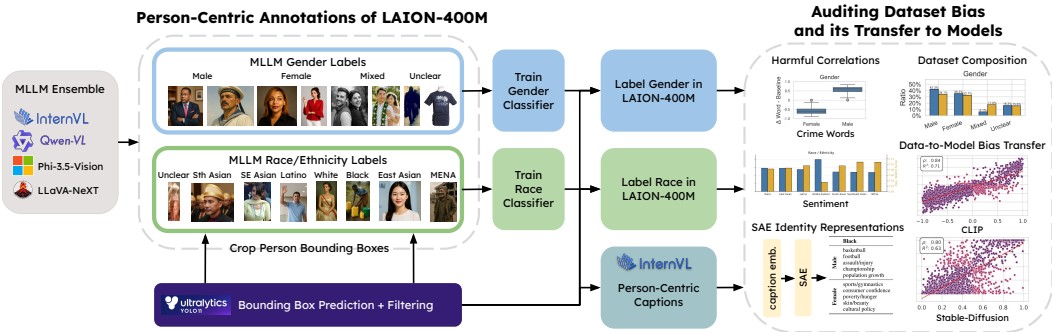

Figure 1: We detect ∼200M person bounding boxes with YOLO11. An MLLM ensemble (Phi-3.5-Vision, LLaVA-NeXT, InternVL3) provides gender and race/ethnicity labels on sampled subsets, and only consensus predictions are used to train SigLIP classifiers. These classifiers then label the full dataset, while InternVL3 generates person-centric captions. The resulting annotations enable systematic analysis of dataset composition, harmful correlations, and bias transfer to downstream models. Images of people are regenerated from the originals by AI to protect privacy.

## 2 BACKGROUND AND RELATED WORK

**LAION-400M.** The LAION-400M dataset (Schuhmann et al., 2021) is a large-scale, publicly available collection of 400 million image-text pairs from Common Crawl. Initial pairs were filtered for semantic relevance, followed by removing duplicates and low-quality entries. All images are resized to a uniform $256 \times 256$ resolution, and the dataset's main application is the pretraining of vision-language foundation models such as Imagen (Saharia et al., 2022), Stable Diffusion (Rombach et al., 2022), and CLIP (Radford et al., 2021). It remains popular in academic research (Sun et al., 2023; Li et al., 2023; Wu et al., 2023; Zhang et al., 2025b) and serves as a representative testbed for analyzing web-scale data (Garcia et al., 2023; Udandarao et al., 2024). As LAION only distributes URLs, images become unavailable over time. Our version collected in September 2022 recovered $375,689,394$ pairs (90.7% of the original), a sufficient portion for generalizable insights.

**Auditing Large-Scale Multimodal Datasets.** Birhane et al. (2023) analyzed text subsets of LAION-400M and LAION-2B, finding significant hateful content. Al Sahili et al. (2025) and Birhane et al. (2024) show that the larger dataset of 2B pairs (Schuhmann et al., 2022) resulted in increased bias, illustrating how larger datasets don't inherently solve issues of stereotyping and bias. Al Sahili et al. report stronger gender and racial skew in OpenCLIP (Ilharco et al., 2021), suggesting data composition matters more than scale. Birhane et al. show larger datasets can amplify harmful stereotypes, e.g., labeling Black and Latino men as "criminal". These findings echo earlier calls for transparency (Birhane & Prabhu, 2021) and underscore the need to audit training data itself. While these studies document consequences of dataset bias, we provide detailed annotations for the entire LAION-400M, enabling direct analysis of its sources.

Other research has also annotated LAION datasets, but with a narrower scope. Seshadri et al. (2024) and Friedrich et al. (2023) studied occupation-related subsets of LAION-2B and LAION-5B, assigning one perceived gender label per image. Díaz et al. (2024) examined co-occurrence of queer identity terms with sexualized content in LAION-400M. Hong et al. (2025) identified sensitive personal information in the DataComp CommonPool dataset (Gadre et al., 2023). Zheng et al. (2022) detected 50M faces in LAION-400M, but mainly to train a face encoder rather than for auditing. Massiceti et al. (2024) and Parashar et al. (2024) focused on rare concepts in LAION-400M, considering only the text portion. Finally, Hamidieh et al. (2024) inferred gender from pronouns in a subset of LAION-400M to study co-occurrence with professions. However, relying only on text is insufficient due to the limited informativeness of alt-text captions (Li et al., 2024).

Our work offers a granular and comprehensive analysis of LAION-400M. Unlike prior efforts, we annotate the full dataset, providing bounding boxes for entire individuals rather than only faces and assigning perceived gender labels to each person instead of a single image-level label. These extensive annotations enable a more direct investigation of LAION-400M's composition and its influence on model behavior. A detailed discussion of related work is in Appendix H.

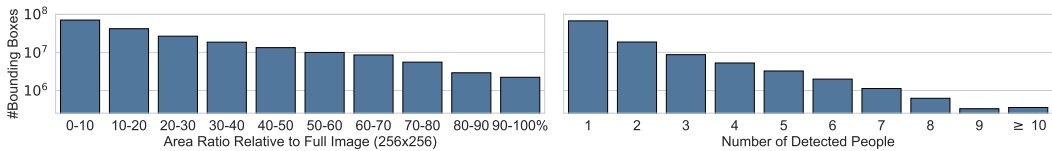

Figure 2: Left: Log-scaled histogram of bounding box areas, expressed as a percentage of the total image area. Right: The distribution of the number of detected people per image.

## 3 AUTOMATICALLY GENERATING BOUNDING BOX, GENDER, RACE/ETHNICITY, AND CAPTION ANNOTATIONS

Here, we describe how we automatically annotate person bounding boxes, perceived gender, and race/ethnicity. We recognize that the gender and racial/ethnic categories used in this study are not exhaustive, and that assigning identity categories to individuals can be ethically fraught and insufficient in capturing the full complexity of personal identity. The purpose of this study is to assess potential biases in existing datasets related to perceived gender and racial identities, and we approach this work with an awareness of the limitations inherent in identity classifications. To mitigate risks, access to our data with assigned gender and race/ethnicity labels alongside the model will only be considered upon request and requires acknowledgement of these ethical considerations. We further discuss the limitations and broader impact of our labels in Appendix A. Examples are in Fig. 3.

**Detecting Person Bounding Boxes.** We detect bounding boxes of people with `YOLOv11-l` (Jocher & Qiu, 2024). `YOLOv11-l` detects all people in each image and returns a bounding box around the person. We keep all bounding boxes with a confidence score of 0.25, the default for `YOLOv11-l`. This threshold is lower than the thresholds used, for example, in Phase (Garcia et al., 2023), because we prioritize recall. Indeed, we show that `YOLOv11-l` achieves high recall in detecting people in annotated real-world datasets (Garcia et al., 2023; Gustafson et al., 2023) (see Appendix C.1). There, we also validate that `YOLOv11-l` does not exhibit gender or racial bias in detection performance by measuring the model's performance on different demographic groups and ensuring that it does not systematically differ between them. Additionally, we ask one human annotator to label 200 images for correctness of the bounding boxes. The annotator decides whether a person is missing (i.e., that did not receive a bounding box), whether a non-human object is assigned a bounding box, or whether the image is correct. We find that 82.5% of images are correct, 10% have a bounding box for a non-human object, and 7.5% of images miss at least one bounding box. Since images can contain multiple bounding boxes, the effective number of missed bounding boxes is actually lower.

After obtaining bounding boxes, we filter out small bounding boxes where person attributes such as gender can no longer be recognized. To filter small bounding boxes, we set a minimum threshold on any bounding box sidelength of 30 pixels. This threshold is based on the performance of automatic labeling methods for perceived gender. In images with a sidelength less than 30 pixels, automatic methods no longer reliably agree with human annotations of perceived gender, i.e. Cohen's $\kappa$ with respect to human labels drops below 0.8, and accuracy below 90%. See Appendix C.2 for details.

In total, after filtering bounding boxes with a minimum side length of less than 30 pixels, we obtain 199,931,986 person bounding boxes in 107,545,236 unique images. In Fig. 2, we show distributions of the bounding box sizes and the numbers of bounding boxes per image. We can see that most bounding boxes are small, i.e. occupying less than 10% of the image area, and in most cases, only one person is visible in an image. However, in some images, up to 55 distinct people are detected.

**Generating Gender Labels.** Modern vision-language models show nearly perfect agreement with human ratings of perceived binary gender. However, processing datasets like LAION-400M presents unique challenges. Automatically detected person bounding boxes can be noisy or imprecise, for instance by including multiple people of different genders. Additionally, images may lack clear gender cues due to occlusion or low quality. To address these issues, we train a custom model using a new dataset specifically curated to handle these cases. Here, we briefly describe dataset construction and model performance, and refer to Appendix D for an extensive discussion.

For our dataset, we sample 3,000,000 person bounding boxes from LAION-400M. We then use three different MLLMs to label the perceived gender in each box with one of four categories: "female",

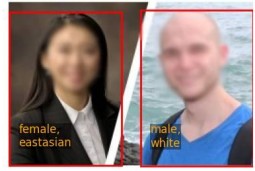 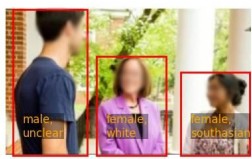 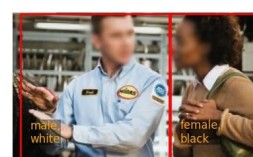 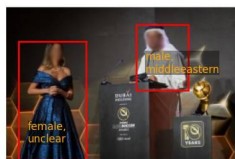

**Caption (left):** The highlighted individual is a woman positioned against a plain background. She has long, straight black hair and is wearing a dark blazer over a white collared shirt. Her expression is neutral, and she is looking directly at the camera. The image appears to be a professional headshot, likely intended for a formal or business context. [...]

**Caption (left):** The highlighted individual is a person with short dark hair, wearing a navy blue t-shirt. They are standing in an outdoor setting, facing slightly to the left of the image. The background includes a brick wall and greenery, suggesting a campus or park-like environment. The person appears to be engaged in a conversation with two others [...]

**Caption (right):** The highlighted person is a woman with dark, curly hair. She is wearing a white collared shirt under a brown sweater with a fuzzy texture. She appears to be standing with her hands clasped together near her chest. The background suggests an indoor setting, possibly a store or service area, with shelves and various items visible. [...]

**Caption (right):** The highlighted individual is a person wearing traditional Middle Eastern attire, specifically a white thobe and a white headscarf with a black agal. The person is standing at a podium with a microphone, addressing an audience. The podium features the text "Dubai World Cup" and "Qatar Foundation," [...]

Figure 3: Example person-centric annotations in LAION-400M. Bounding boxes are in red and contain the gender and race/ethnicity labels. Captions by InternVL (one caption per image) are below. Faces are blurred to protect the privacy of the shown individuals.

"male", "mixed" (multiple people of different genders are present), or "unclear" (no person is visible or gender cues are absent). The models used for labeling are `InternVL3-2B` (Zhu et al., 2025), `Phi-3.5-Vision-Instruct` (Abdin et al., 2024), and `LLaVA-1.6-7B` (Liu et al., 2024). From the labeled bounding boxes, we select 100,000 images, with 25,000 per category, where all three models agree on the label. Since the "mixed" label occurred infrequently, we also included cases where only two of the three models agree on "mixed", which make up $\approx 25\%$ of the images in this category. Finally, we split the dataset into train (80%), validation (10%), and test (10%) sets.

We use the train split to fine-tune a SigLIP model (Zhai et al., 2023) from OpenCLIP (Ilharco et al., 2021) to classify bounding boxes into one of the four gender categories. Our model achieves 97.2% accuracy on the test split. We also evaluate this model on binary gender classification tasks using two other datasets. On Phase (Garcia et al., 2023), it reaches 95% accuracy, while on FACET (Gustafson et al., 2023), it achieves 90% accuracy. The slightly lower performance on FACET can be explained by possible label noise in this dataset, see Appendix B.3.

Our model subsequently assigns a perceived gender label to all person bounding boxes detected in LAION-400M. From these, we also generate image-level gender labels. An entire image is labeled "male" if it only contains people labeled as "male" (and optionally "unclear"). Similarly, an image is labeled "female" if it only contains people labeled as "female". If an image includes at least one male and one female person, it is labeled "mixed". All other images are labeled "unclear".

**Generating Race/Ethnicity Labels.** Similar to our gender labeling, we train a custom classifier to label the perceived race/ethnicity of people detected in LAION-400M. Previous works, e.g. (Karkkainen & Joo, 2021; Garcia et al., 2023), have mostly combined race (such as Black or White people) with ethnicity or cultural context (such as Southeast Asian or Hispanic) into a single category termed "race". We acknowledge that this is inherently inconsistent, but to align with prior work, we use an equivalent labeling schema, which we term "race/ethnicity". As no dataset exists that accounts for missing race/ethnicity cues or noisy bounding boxes, we again create our own.

As race/ethnicity categories for our study, we use the combined set from the most relevant published datasets with demographic labels (Karkkainen & Joo, 2021; Seth et al., 2023; Garcia et al., 2023): Black, East Asian, Hispanic, Middle Eastern, South Asian, Southeast Asian, and White. Note that South Asian is named "Indian" in (Garcia et al., 2023) and (Seth et al., 2023). Next, we map each category to a set of related countries and terms. We retrieve all images from LAION-400M whose alt-text captions contain any keyword associated with a given race/ethnicity category and collect the corresponding person-detection bounding boxes. This increases the probability of including images showing people perceived as belonging to the respective race/ethnicity and ensures that any labels are grounded in textual descriptions of the image.

We then use the same three MLLMs we used to label gender to label the perceived race/ethnicity category for a sample of 7,000,000 images (1,000,000 per category). We retain only the bounding boxes that are labeled with the same category by all three MLLMs. The resulting sets contain images of people who have visual attributes that are consistently perceived as correlated with the respective race or ethnicity, as confirmed by the agreement between the diverse models and the keyword used to retrieve the image. However, we stress that this assessment is only based on visual cues and does not take into account the self-identified race or ethnicity of the shown person, which could also lie

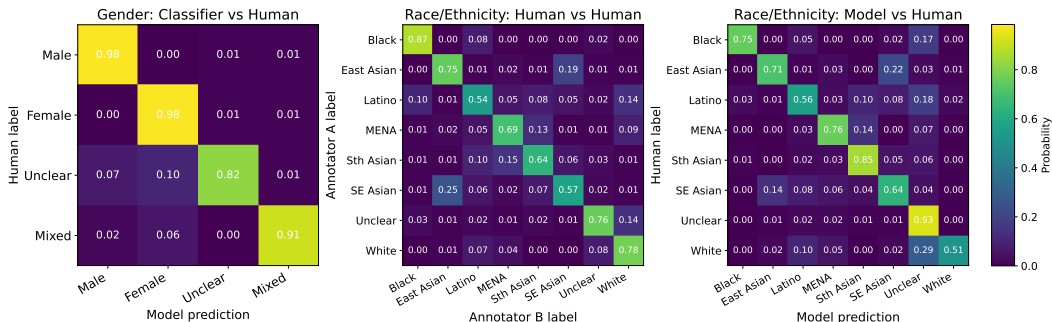

Figure 4: Human validation of classifier annotations. Confusion matrices showing agreement between model and human annotations for gender (left) and race/ethnicity (right). Values are row-normalized. For race/ethnicity, both the human–model (left) and inter-human (right) agreements exhibit similar confusion patterns, which reflect high epistemic uncertainty between groups.

outside our label set. We think our work is necessary to trace the origins of race/ethnicity bias of models in their training data, but we do not aim to draw a conclusion about individual people beyond holistic dataset statistics.

Finally, after obtaining labels from our MLLM ensemble, we sample 25,000 images per category, balanced for gender and keywords, and combine them into our final dataset. We also add 25,000 images labeled as "unclear" by all three MLLMs. The dataset is split using stratified sampling into a training set (80%), a validation set (10%), and a test set (10%). See Appendix E.1 for details on the keywords used for retrieval, the MLLM labeling process, and the final dataset composition.

Our finetuned SigLIP model achieves 87.4% raw accuracy on the test split, which is a strong performance given the ambiguity of perceived race/ethnicity and low agreement among humans (Garcia et al., 2023). To additionally increase precision, we investigated only accepting a race prediction if the predicted logit is higher than a threshold $\tau$ (Hendrycks et al., 2022). However, we noticed that increasing precision through thresholding does not affect our insights (but further increases the ratio of White-appearing people), so we continue using the original, non-thresholded predictions.

**Human Validation Study.** We validate the accuracy of our annotations on a set of 648 images, stratified by gender and race/ethnicity predictions. Two human annotators annotate the gender and race/ethnicity of all images, and we compare these human annotations with the classifier predictions.

Fig. 4 (right) shows confusion matrices for human-classifier agreement (right panel) and human-model agreement (left panel). Confusion counts are row-normalized for better comparability. Overall, Cohen's $\kappa$ for human-classifier agreement is 0.638 (avg.), i.e., indicating moderate to high agreement. However, human-human agreement is only slightly higher, $\kappa = 0.654$. This clearly shows that there is a high epistemic uncertainty, i.e., perceived race/ethnicity is to a substantial degree subjective. Please note that our human-human agreement is higher than reported in other studies (e.g., (Garcia et al., 2023)), because our sampling of images is conditioned on classifier-assigned labels. We also observe similar confusion trends. Disagreement is high for White, Latino, and Middle Eastern groups, and for East Asian and Southeast Asian. Humans also show higher disagreement on Black and Latino, and on Middle Eastern and South Asian. These patterns are unsurprising and can be explained by cultural or geographic similarities, as well as common skin tone associations. Finally, we would like to note that our classifier assigns "unclear" more frequently than humans. However, we think these more conservative judgements are more desirable than overgeneralization.

Fig. 4 (left) shows the human-classifier confusion matrix for gender assignments. Notably, we observe no confusion between the male and female classes. However, disagreements occur where the model predicts male or female, while human annotators select "unclear" or "mixed". We inspect these cases and find that the model sometimes assigns a gender based on contextual cues, particularly clothing, even when the person is mostly occluded. Regarding disagreements on the "mixed" class, human labelers are stricter than the model when considering small people in the background. We acknowledge that both failure modes can yield unintuitively labeled images, but overall, we find no severe or systematic error trends.

Figure 5: Distribution of perceived gender (left) and perceived race/ethnicity labels (right) in 199,931,986 bounding boxes and 107,545,236 unique images.

**Generating Person-Centric Captions.** We generate synthetic descriptions for each detected person. For this, we use MLLMs due to their strong reasoning and captioning performance (Liu et al., 2023; Zhu et al., 2025). A key challenge is providing bounding box information to the MLLM to focus on a specific person while still considering the entire image context. Our approach makes use of the observation that recent MLLMs, specifically InternVL-3 and Qwen-VL-2.5, are aware of visual markers in images, similar to findings by Shtedritski et al. (2023). We draw the bounding box around the person of interest as a red frame and instruct the model to describe the highlighted individual. See Appendix F for examples and the detailed prompt that we use.

To select the most suitable model to generate person-centric captions, we compared four MLLMs on 500 bounding boxes: `InternVL3-2B`, `InternVL3-8B`, `Qwen2.5-VL-3B-Instruct`, and `Qwen2.5-VL-7B-Instruct`. For each bounding box, we presented captions generated by both models to `GPT-5.1` (medium reasoning effort) and asked it to select the better caption, based on correctness, accurate representation of details, and fluency. Win rates for `InternVL3-8B` against components are: 0.756 against `Qwen2.5-VL-3B`, 0.630 against `Qwen2.5-VL-7B` and 0.582 agains `InternVL3-2B`. Therefore, we selected `InternVL3-8B` to generate person-centric captions for all bounding boxes. See Fig. 3 and Table 11 in the supplementary material for examples. Additionally, we examine the captions for potential biases and find a low prevalence of crime-related keywords and an overall positive sentiment, as shown in Appendix F.

# 4 AUDITING LAION-400M FOR DEMOGRAPHIC BIASES

## 4.1 UNVEILING THE GENDER AND RACE/ETHNICITY DISTRIBUTION IN LAION-400M

We show the gender and race/ethnicity distributions resulting from our automatic labeling in Fig. 5. We find that approximately 42% of the bounding boxes depict men, while only 35% depict women. This explains why CLIP models treat men as the default gender (Wu et al., 2024b; Ghosh & Caliskan, 2023), as they are exposed to significantly more male-gendered content than female-gendered content during pretraining. The imbalance is less pronounced in images that exclusively show men or women, but it remains significant. Interestingly, only a fraction of images (17%) show both men and women. This shows that models have ample opportunity to learn gender-specific correlations because alt-text captions describe the entire image, rather than specific individuals within it.

For race/ethnicity, we observe a similar trend. Among the labeled bounding boxes and images, the White category is by far the largest group ($\approx 28\%$) and is approximately four times larger than the next largest group, Black. However, more than 50% of bounding boxes and about 45% of images are labeled as unclear, indicating that no information related to race or ethnicity is visible. This reflects that these are more subjective and have fewer distinct visual cues than perceived gender. Other groups, such as Southeast Asian, Latino, or Middle Eastern, appear in only a small fraction of images, confirming that they are underrepresented in the dataset. This does not only mean that certain appearances of people are omitted, but, combined with our findings on geographical imbalance in Appendix G.1, also implies overrepresentation of Western culture and hence worldview overall (Liu et al., 2021; Mandal et al., 2021; Senthilkumar et al., 2024; Bayramli et al., 2025).

## 4.2 ANALYSING IDENTITY CORRELATIONS WITH CRIME WORDS AND SENTIMENT

**Crime Concepts.** First, we analyze the association between gender, race/ethnicity, and crime, motivated by findings from Agarwal et al. (2021) that models sometimes misclassify images of men into

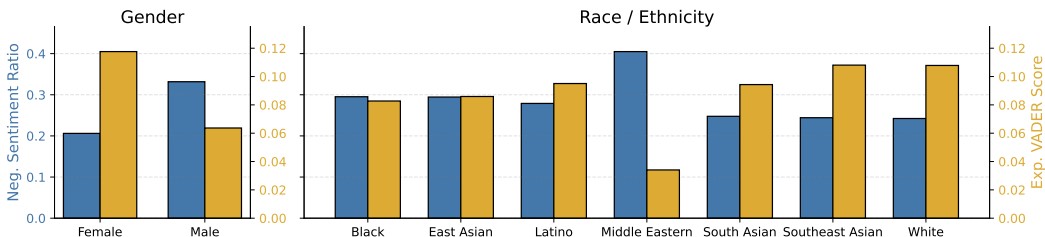

Figure 6: Distribution of the relative change ($\Delta$) in representation for gender (left) and race/ethnicity (right) groups when associated with 63 crime words. $\Delta$ is calculated relative to the baseline distribution of each group. Positive values indicate a stronger association than expected.

Figure 7: Sentiment analysis of alt-text captions for gender (left) and race/ethnicity (right) groups. For each group, we show the ratio of captions with negative sentiment (blue) and the average VADER compound score (amber), excluding captions with a neutral VADER score of zero.

crime-related categories. We expand a set of 63 crime-related words from (Hamidieh et al., 2024) (see Appendix B.1 for the full list) and retrieve all images whose alt-text captions contain these words. For this image set, we calculate the gender and race/ethnicity distribution, discarding images with unclear or mixed identities. We then measure the relative change ($\Delta$) for each group compared to its baseline distribution in the dataset (Fig. 5). $\Delta$ of 1.0 signifies a 100% increase (doubling) in representation, while a $\Delta$ of -0.5 signifies a 50% decrease (halving).

Results in Fig. 6 reveal strong trends. Men appear with crime words far more often than expected (+57%). By race/ethnicity, Black (+51%) and Middle Eastern (+206%) groups show the largest increases, while Latino (+19%), Southeast Asian (+20%), and South Asian (+13%) rise more modestly. White and East Asian groups decrease (both -22%), with the White decrease substantial in absolute terms given group size. These statistics confirm earlier findings (Agarwal et al., 2021; Al Sahili et al., 2025), showing that Black and Middle Eastern individuals are disproportionately linked to crime-related captions.

**Sentiment.** We next examine correlations between gender or race/ethnicity and sentiment, as prior work reported bias in this regard (Kiritchenko & Mohammad, 2018; Birhane et al., 2023; Girrbach et al., 2025a). We compute the VADER compound sentiment score (Hutto & Gilbert, 2014) for all alt-text captions clearly associated with one race/ethnicity or gender, excluding unknown and mixed. Using the NLTK implementation (Bird, 2006), we measure for each group the ratio of negative captions and the average VADER score, discarding neutral cases. In Appendix G.2, we confirm that our findings still hold when using trained sentiment classifiers. Results in Fig. 7 reveal clear differences. Captions linked to females show fewer negatives (0.21) and higher average sentiment (0.12) than males (0.33 and 0.06). Among racial groups, Middle Eastern individuals display the highest negative ratio (0.40) and the lowest average score (0.03). Other groups vary less, with Southeast Asian and White individuals showing the most positive averages (both $\approx$0.11). These findings explain previous observations on male gender and negative sentiment (Kiritchenko & Mohammad, 2018; Girrbach et al., 2025a) and reinforce the negative portrayal of Middle Eastern and Black people seen in our crime-word analysis.

### 4.3 ANALYZING IDENTITY REPRESENTATIONS WITH SPARSE AUTOENCODERS (SAEs)

We aim to uncover themes that are strongly associated with gender and race/ethnicity identities in our dataset. To do this, we start with approximately 200 million automatically generated person captions (from `InternVL3-8B`) and represent each caption as an embedding using

|  | Black | East Asian | Latino | Middle Eastern | South Asian | Southeast Asian | White |
|---|---|---|---|---|---|---|---|
| **Male** | basketball
football
assault/injury
championship
population growth | anime/comics
video games
martial arts
Chinese culture
energy/weapons | bull-/horseriding
luxury cars
enduro
soccer/rugby
mormonism | Islam
firearms/weapons
military
public service
drought/desserts | kabaddi
cricket
Indian religions
bollywood
Islam | Buddhism
sepak takraw
martial arts
meat/seafood
work/labour | men's health
(ice) hockey
aging
auto racing
management |
| **Female** | sports/gymnastics
consumer confidence
poverty/hunger
skin/beauty
cultural policy | anime/comics
Chinese culture
school/languages
Asian car brands
pets/dolls | enduro
lamborghini
culture
nutrition
women's rights | Islam
data protection
hardware modding
archeology
culture | Indian religions
kabaddi
traditional music
bollywood
Islam | culture
sepak takraw
fruits/vegetables
markets
natural medicine | beauty pageants
pregnancy
bmx freestyle
flowers
surfing |

Table 1: Topics with highest PMI per intersectional identity. We condense the top 20 topics into 5 representative keywords. Full results with original topic names are in Appendix B.2.

`granite-embedding-english-r2` (Awasthy et al., 2025). On these embeddings, we train Sparse Autoencoders (SAEs) to discover recurring topics (Peng et al., 2025).

**Discovering Identity-Topic Associations.** To find topics strongly associated with identity groups, we estimate the pointwise mutual information of an identity $i$ and a topic $t$: $\mathrm{PMI}(i,t) = \log \frac{P(i,t)}{P(i)\,P(t)}$. We use the trained SAEs to model $P(i,t)$ and $P(t)$. Specifically, we calculate the probability $P(F_j)$ for each latent feature to be active, the conditional probability $P(i\,|F_j)$ of an identity given that an SAE latent feature $F_j$ is active, and the conditional probability $P(t\,|F_j)$ of a topic associated with feature $F_j$. Through marginalization, these conditional probabilities give the required $P(i,t)$ and $P(t)$. $P(F_j)$ and $P(i\,|F_j)$ are estimated from activation statistics, which are collected by passing caption embeddings for all identity groups through the SAE. To model $P(t\,|F_j)$, we retrieve the five topics most similar to the SAE decoder embedding corresponding to $F_j$ using cosine similarity, following Rao et al. (2024). The resulting cosine similarities are then normalized to form a probability distribution. See Appendix B.2 for the technical details of how we compute $\mathrm{PMI}(i,t)$ and our topic list. There, we also discuss how we ensure robustness of results by combining PMI scores from 24 different SAEs and integrating over topic clusterings of varying granularity.

**Results.** We evaluate 14 intersectional identities, consisting of 2 genders $\times$ 7 race/ethnicity groups. For each identity, we retrieve the 20 most associated topics and manually summarize them in Table 1. The full results with the original topic names are in Appendix B.2.

When comparing male and female identities, sports themes are more frequent for male identities, including topics such as "hockey", "soccer", and "martial arts". In contrast, topics related to culture, such as "culture", "archeology", and "traditional music", are strongly associated with female identities. Across race/ethnicity groups, we find many topics related to regional culture, including different religions ("Islam", "Buddhism", Indian religions), sports ("kabbadi", "sepak takraw", ...), and traditions ("bollywood", "natural medicine"). These are not entirely unexpected, as most groups are tied to specific regions, a factor that is explicitly leveraged when training our labeling model.

Other topics reflect societal or economic circumstances. For example, "firearms/weapons" and "military" are associated with Middle Eastern identities, while "markets" and food categories are associated with Southeast Asian identities. While these associations may reflect existing conditions, they can also be problematic, as they point to a skewed representation of the associated identities. For instance, they portray Southeast Asia as underdeveloped and the Middle East as a region in armed conflict. This observation is further amplified by the occurrence of generic topics like "health", "aging", "pregnancy", and "flowers" for White identities, which confirms the observation that these identities are treated as the default in the LAION-400M dataset (Wolfe & Caliskan, 2022a).

## 5 Measuring the Transfer of Dataset Gender Bias to Models

We aim to develop and test hypotheses about how dataset imbalances lead to model bias, similar to studies on scaling laws that predict foundation model performance (Kaplan et al., 2020; Hoffmann et al., 2022). Our exhaustive labels allow us to systematically study this relationship for the first time, whereas previous work relied on comparisons with proxy statistics, such as data from the U.S. Bureau of Labor Statistics (Luccioni et al., 2023; Cheong et al., 2024). In this work, we evaluate *first-order bias transfer*, which is the extent to which model bias is linearly related to the co-occurrence frequency of a target concept and a bias variable (e.g., gender) within the dataset.

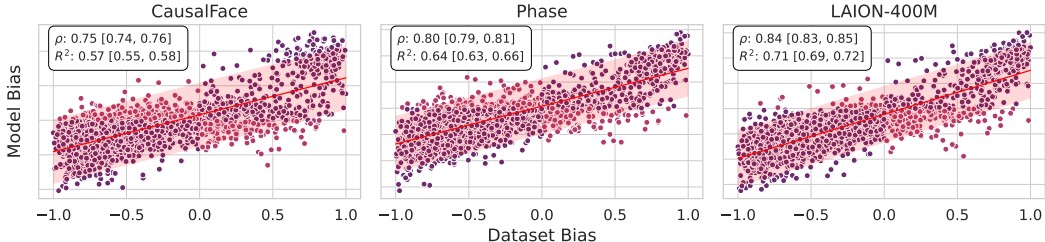

Figure 8: Correlation between gender bias in LAION-400M and CLIP `ViT-B-32-quickgelu` trained on it. The x-axis shows dataset bias as the difference between social categories from (Guilbeault et al., 2024) co-occurring with male- or female-gendered images. The y-axis shows model bias as the difference between average cosine similarities of social categories and male- or female-gendered images, normalized to unit std. dev. We additionally show Pearson correlation $\rho$ and $R^2$ score of the best linear fit (red line). 95%-confidence intervals (in brackets) are tight. Light vs. dark points show categories where the sign (i.e., male- vs. female-leaning) agrees (dark) or disagrees (light) between data and models.

**Social Categories for Bias Evaluation.** We calculate dataset bias and model bias using two taxonomies of social categories. First, Hamidieh et al. (2024) propose So-B-IT, which organizes 405 keywords into nine categories, such as "Appearance" or "Healthcare". Second, Guilbeault et al. (2024) evaluate gender bias in online image search results across 3488 social categories, which include a variety of social roles and occupations like "sailor", "client", or "homeowner". From both resources, we only use words that appear as lemmas in the English WordNet (Miller, 1995). This process yields a total of 3710 combined social categories. Of these, we only consider categories that appear at least 100 times in LAION-400M to reliably estimate dataset bias. This filtering leaves 2261 categories from Guilbeault et al. (2024) and 356 categories from So-B-IT.

**Dataset Bias.** To measure dataset bias for a social category $c$, we retrieve all alt-text captions in LAION-400M that contain $c$. We then filter this set to include only captions whose paired images show either only women or only men. Finally, we calculate the proportion of images showing women among this filtered set. We use this female ratio as a measure of dataset imbalance with respect to the category $c$.

**CLIP Bias.** OpenCLIP (Ilharco et al., 2021) has released several CLIP models pretrained on LAION-400M. To measure gender bias in CLIP, we follow previous work (Caliskan et al., 2017; Wolfe et al., 2022; Hamidieh et al., 2024) by measuring the difference in association strength between a social category $c$ and gender (male or female). We calculate the mean cosine similarity between the text embedding of the social category and a set of images representing each gender. Then, we calculate the difference between these mean cosine similarities and standardize it:

$$d(c) = \frac{\text{mean}_{x \in F} \cos(x, c) - \text{mean}_{y \in M} \cos(y, c)}{\text{stddev}_{w \in F \cup M} \cos(w, c)}, \qquad (1)$$

where $M$ is a set of male-gendered images and $F$ is a set of female-gendered images. For the image sources, we compare embeddings from three datasets: the training portion of the Phase dataset (Garcia et al., 2023), our own dataset sourced from LAION-400M to train our gender labeling model, and CausalFaces (Liang et al., 2023; Hausladen et al., 2024). This comparison allows us to discern the effect of using face data versus images of full people in bias analysis.

**Stable Diffusion Bias.** To measure gender bias in Stable Diffusion models, we generate 100 images for each social category from two models: Stable Diffusion 1.1 and Stable Diffusion 1.4 (Rombach et al., 2022). We chose these models because they were trained on subsets of LAION-5B (Schuhmann et al., 2022), which we expect to have a concept distribution similar to that of LAION-400M. The social categories are inserted into the prompt template "`A picture of a {{category}}`". If the category word is an adjective, we append the word "person".

In total, we generate $2 \times 3710 \times 100 = 742,000$ images. We filter these images, keeping only those that show exactly one person, as detected by `YOLOv11-l`, and where the person is labeled as male or female by `InternVL3-2B`. This ensures that the perceived gender in each image is unambiguous. We continue generating images for each prompt until we have 100 images that meet

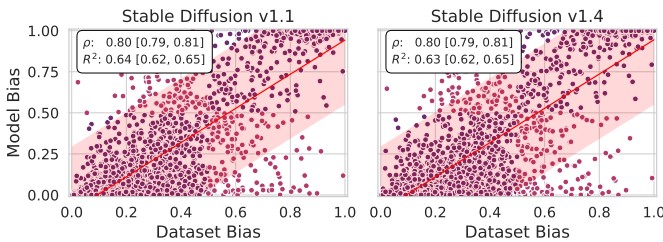 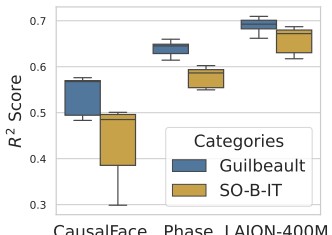

Figure 9: Correlation between gender bias in LAION-400M and Stable Diffusion Models. 95%-CIs (in brackets) are tight.

Figure 10: $R^2$ scores for image and category sources.

the criteria or until a maximum of 500 images have been generated. For each social category, we calculate the ratio of female-gendered images as a measure of the diffusion model's gender bias.

**Results.** In Fig. 8, we show the relationship between data bias and CLIP model bias across three image datasets (CausalFace, Phase, and LAION-400M), using categories from (Guilbeault et al., 2024). We calculate the Pearson correlation $\rho$ and the $R^2$ score between the dataset bias and the model bias. The $R^2$ score indicates the fraction of variance in model bias that is explained by a linear fit from the dataset bias. This score quantifies how much of the model bias can be linearly predicted by first-order co-occurrences in the data.

Model and dataset bias correlate strongly ($\rho \in \{0.75, 0.80, 0.84\}$; $R^2 \in \{0.57, 0.64, 0.71\}$), with higher values for in-distribution images (LAION-400M) than for Phase or CausalFace, highlighting the importance of probing images. As shown in Fig. 10, these trends hold across 12 CLIP models. Results also depend on the choice of social categories: correlations are lower for SO-B-IT than for Guilbeault et al. (2024), likely because SO-B-IT includes more general concepts (e.g., adjectives) that often occur outside people-related contexts.

Stable Diffusion results in Fig. 9 mirror those from CLIP: SD-1.1 and SD-1.4 yield nearly identical outcomes ($R^2 = 0.64, 0.63$, $\rho = 0.80$). Overall, 60–70% of gender bias in CLIP and Stable Diffusion trained on LAION-400M or similar data can be linearly predicted by gender–concept co-occurrences. Future work should test second-order or nonlinear models for improved prediction, and examine race/ethnicity bias in controlled settings with rebalanced data, as our results there were inconclusive due to the low number of concept co-occurrences with non-white people.

**Discussion & Future Work.** This work provides a foundation for large-scale study of dataset-to-model bias transfer. Several directions extend beyond linear prediction of model bias from direct co-occurrences. First, indirect co-occurrences also influence bias (Schrouff et al., 2024; Zhang et al., 2024b). Bias can propagate between words that frequently co-occur when one also appears with a demographic attribute. Modeling this second-order bias transfer requires methods such as label propagation, which we leave for future work. Second, optimizers and data-sampling strategies may alter how strongly models inherit dataset bias. However, evaluating this requires training CLIP from scratch. Third, model training introduces nonlinear dynamics, seen in Figs. 8 and 9, where co-occurrence rates and model bias show nonlinear patterns. As shown in Appendix G.3, nonlinear predictors like Chebyshev polynomials raise $R^2$ by one to three points. Still, advancing methods that model dataset bias beyond simple co-occurrences remains the most promising direction for understanding dataset-to-model bias transfer.

## 6 CONCLUSION

In this work, we created person-centric annotations for the complete LAION-400M dataset, enabling the first systematic audit of demographic representation at web scale. Our contributions include bounding boxes for all detected people, automatically inferred perceived gender and race/ethnicity labels, and detailed person-level captions. With these resources, we quantified demographic distributions, harmful associations with crime and sentiment, and thematic patterns linked to intersectional identities. Importantly, we demonstrated that while 60-70% of model bias can be linearly predicted from direct concept-identity cooccurrences, a significant portion of bias comes from higher-order or nonlinear effects of data, or bias amplification in models. Our labels create the foundation to study more advanced theories of dataset-model interaction, which was so far impossible at web-scale.

## ETHICS STATEMENT

Please see Appendix A for an extensive discussion of the limitations and the broader impact of our work. We use automatic demographic labeling at web scale, which is necessary for LAION-400M but risks embedding societal norms and biases as objective data. We capture only perceived, not actual, gender and restrict to binary categories, since visual cues reflect appearance rather than identity, and current models rarely recognize non-binary expressions, which we acknowledge as a key limitation. Similarly, we capture perceived, not actual, race/ethnicity and focus on certain identity groups. Race/Ethnicity, similar to gender, is socially constructed and inconsistently defined, but we study it as a set of stereotypes because AI models clearly encode such biases. Our labels, therefore, reflect perceived appearance, not true identity, and are intended only for research on dataset and model bias, not for broader purposes, including but not limited to surveillance or profiling. While automated methods may reproduce model biases, we mitigate this through multi-model agreement and validation. Despite these limitations, our annotations are critical to audit large-scale data, trace stereotypes, and enable fairer AI systems. Given that people are best placed to identify themselves, future research, including image data collection, should prioritize having people self-identify, as is possible, and annotations that reflect those self-identifications.

## REPRODUCIBILITY STATEMENT

The data generated for this work is available on HuggingFace.[1] This includes 276,824,258 bounding boxes detected by YOLOv11, the gender and race/ethnicity labeling models, and 199,931,986 gender and race/ethnicity labels with person-centric captions for bounding boxes having a minimum side length of 30 pixels. We also provide the sentiment labels collected for our analyses in Section 4.2 and Appendix G.2, and in addition, we make available the 742,000 images generated for our analysis of gender bias in Stable Diffusion. All annotations are paired with the respective LAION-400M image IDs, if applicable. Our code is available on GitHub.[2]

Due to their sensitive nature, gender and race/ethnicity labels, as well as the trained classifiers, will be provided only upon request to the authors and are conditional on the acceptance of the terms of use. In contrast, bounding boxes, sentiment labels, captions, and generated images will be publicly released on Huggingface.

*Note on LLM Usage:* To improve clarity, the manuscript was polished for grammar and style using a large language model, with all final text reviewed and validated by the authors.

## ACKNOWLEDGEMENTS

This work was partially funded by the ERC (853489 - DEXIM), the Alfried Krupp von Bohlen und Halbach Foundation and Berkeley AI Research (BAIR) Commons, which we thank for their generous support. This work is also supported by Hi! PARIS and ANR/France 2030 program (ANR-23-IACL-0005). The authors gratefully acknowledge the scientific support and resources of the AI service infrastructure *LRZ AI Systems* provided by the Leibniz Supercomputing Centre (LRZ) of the Bavarian Academy of Sciences and Humanities (BAdW), funded by Bayerisches Staatsministerium für Wissenschaft und Kunst (StMWK). The authors also acknowledge the use of the HPC cluster at Helmholtz Munich for the computational resources used in this study.

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

# Supplementary Material

## A    LIMITATIONS AND BROADER IMPACT

In this section, we discuss the potential problems of automatically labeling perceived gender and race/ethnicity, describe the steps taken to mitigate these issues, and examine the broader impacts of providing demographic labels.

**Human vs. Automatic Labels.** While human annotations introduce biases because they reflect the individual annotators' views and current societal norms, automatic labeling can effectively reproduce these subjective labels on a large scale if it aligns well with human judgments. In this situation, a high level of agreement means that the automatic method accurately captures how humans perceive gender. Therefore, automatic methods allow for consistent and scalable labeling. They mirror human understanding, but also their biases. Crucially, this process risks codifying a specific set of societal norms at a massive scale, potentially reifying transient human biases into a seemingly objective ground truth for future AI systems. However, automatic labeling is necessary for our purposes due to the massive scale of LAION-400M, which makes human labeling infeasible. Through controlled experiments, we ensure high agreement with perceived gender and race stereotypes.

**Perceived Gender.** Due to the limitations of visual data, we can only determine the *perceived* gender of individuals in images, not their *actual* gender. Gender is complex, involving biological, psychological, and social aspects, none of which can be fully understood from static visual cues. Images mainly show outward appearance, such as clothing, hairstyles, and facial features. While these are often linked to gender expression, they do not definitively show a person's self-identified gender or biological sex. Therefore, any gender assigned based solely on visual information is an assumption based on societal gender norms and stereotypes. This reflects a perception rather than an objective fact about an individual's true gender identity. Systems that rely on such perceived labels can therefore lead to misgendering, a direct harm to individuals whose gender identity does not align with their perceived gender. Therefore, we only target statistical correlation of entire gender groups in our analyses, and not individual people, thereby reducing potential harm to individuals.

**Binary Gender.** A binary gender system recognizes two distinct genders: male and female. In this system, individuals are categorized into one of these two groups, usually based on the sex they are assigned at birth. This differs from non-binary understandings of gender, which acknowledge a wider range of gender identities. Human gender is more varied than just "male" or "female". First, biological sex itself is not strictly binary. Second, gender identity, which is a person's inner sense of their own gender, can include feeling like a man, a woman, both, neither, or something else entirely, going beyond two fixed categories. In this study, we focus on binary gender for two main reasons:

First, visual cues often provide information consistent with an individual's assigned sex, leading to the perception of binary gender (male/female). However, these visual cues are generally inadequate for discerning perceived non-binary gender. Inferring non-binary identities from images would largely depend on interpreting a person's gender expression, i.e. outward presentations that may defy or combine traditional gender aesthetics. These interpretations are subjective and rely on cultural context, rather than directly revealing a person's internal non-binary gender identity.

Second, labeling perceived non-binary gender presents substantial challenges for current automatic methods. Empirical observations support this: Girrbach et al. (2025b) found that `InternVL2-8B` rarely labels images as "nonbinary", and human annotations in the FACET dataset (Gustafson et al., 2023) also show that less than 1% of perceived gender labels are "nonbinary". This indicates that, while important, the widespread use of "nonbinary" as a perceived gender option is not currently feasible with existing visual data and model capabilities. This limitation is primarily due to the subtle or absent visual cues for perceived non-binary gender in current datasets and the limitations of automated systems in interpreting these complex expressions.

We acknowledge that this necessary methodological limitation contributes to the broader erasure of non-binary individuals within AI datasets and models, highlighting a critical area for future research and data collection. Therefore, we focus on binary perceived gender in this study, but intend to extend our analyses beyond binary gender in the future.

**Race/Ethnicity.** Many datasets provide race labels for their images (Karkkainen & Joo, 2021; Seth et al., 2023; Garcia et al., 2023). However, race is a social construct (Sauer, 1992; Cosmides et al., 2003) with no basis in genetic variation (Lewontin, 1972; Yudell et al., 2016). Its labels are highly unstable and subjective, except for unambiguous distinctions such as (binary) skin color (Edgar et al., 2011; Garcia et al., 2023). Furthermore, race conflates multiple factors like skin tone, ancestry, and nationality in a way that is hard to disentangle. The race categories in our work inherit these inconsistent factors of variation, such as skin color (Black, White), geographical origin (East Asian, Southeast Asian, Middle Eastern), and ethnicity (Hispanic).

Imposing an arbitrary racial labeling system can therefore do more harm than good. Such systems erase identities that do not fit their categories (Scheuerman et al., 2020; Khan & Fu, 2021) and risk perpetuating an inequality-causing worldview. This occurs by reifying these arbitrary labels, which subsequent work may then treat as normative (Duster, 2005; Benthall & Haynes, 2019).

Nevertheless, race as a construct exists in people's minds (Taylor et al., 1978; Hewstone et al., 1991) and has real-world effects (Hanna et al., 2020). Foundation models also exhibit clear biases concerning race, ethnicity, and other visual cues of a person's appearance (Agarwal et al., 2021; Bianchi et al., 2023; Howard et al., 2025). We must therefore understand how these models acquire their learned stereotypes and which traits they associate with concepts of race. Consequently, we operationalize the study of race bias in data and models as the study of visual cues commonly associated with particular races or ethnicities, regardless of whether they are grounded in actual, generalizable visual differences between people. By explicitly referring to such attributes and ethnicity instead of perceived race, we can investigate possible sources of race bias in models without making normative claims about a person's identity.

**Broader Impact.** We provide extensive gender and race/ethnicity labels for LAION-400M to enable better dataset audits and deepen the understanding of how dataset bias relates to model bias. We stress that our labels are intended only for academic research on social biases and similar problems that require such annotations. This includes, for example, auditing datasets for representational disparities, measuring performance gaps in generative and discriminative models (Garcia et al., 2023; Girrbach et al., 2025b), and studying the origins of harmful associations. Conversely, these labels are explicitly not intended for, and should not be used in, any real-world surveillance, predictive policing, biometric identification, or commercial profiling systems.

Furthermore, our labels do not represent claims about the actual or self-identified gender or race/ethnicity of the people shown in the images. Our labels only reflect perceived gender or race/ethnicity based on visual cues correlated with stereotypical appearances and expressions. While we take great care to assess and validate the quality of our labels, models inevitably make mistakes due to factors such as model deficiencies or poor data quality. The use of automated labels risks introducing the models' own biases into the analysis, which can be compounded when multiple tools are used. To counter this, we use multiple MLLMs when creating our dataset and carefully evaluate all models in controlled settings, showing that they provide accurate labels.

Also, a study of this scale has a considerable environmental impact (Strubell et al., 2020; Schwartz et al., 2020). In our experiments, we aimed to mitigate this by using smaller models whenever possible without compromising the quality of our labels. For example, we used `ViT-B-16-SigLIP` as the basis for our trained models instead of larger variants like `ViT-SO400M-14-SigLIP-384`. We also used small MLLMs for data curation, such as `InternVL3-2B` instead of `InternVL3-8B`. We acknowledge that substantial computational effort is an unavoidable cost of AI research, and the goals must be carefully evaluated to justify the expense. We believe this is the case for our research.

Finally, providing demographic labels can be harmful by reinforcing stereotypes and being treated as normative in subsequent work (Duster, 2005; Benthall & Haynes, 2019). This issue is particularly problematic when demographic labels are used for surveillance purposes (Kalluri et al., 2025).

Despite these problems, we believe that our labels are necessary for achieving a better understanding of social bias in AI, which justifies their limited use for this specific purpose. Ultimately, our purpose is to enable fairer and more equitable datasets and models, and we believe understanding and unveiling current inequalities and the underrepresentation of certain identity groups is an important step towards this goal.

# B  ADDITIONAL EXPERIMENTAL DETAILS

## B.1  CRIME WORDS TO IDENTIFY HARMFUL CONCEPT-IDENTITY CORRELATIONS

In Table 2, we list all 63 crime-related keywords we use in Section 4.1 to analyze correlations of gender and race/ethnicity categories and crime. This list was expanded from the crime-related keywords provided by Hamidieh et al. (2024) by adding additional relevant keywords and removing keywords that do not unambiguously point towards crime, such as "insane".

| Crime Keyword Category | Crime Keywords |
| --- | --- |
| General Crime Terms | burglar, criminal, embezzler, felon, fraud, gang-related, gangster, hacker, illegal, lawless, mugger, murderer, robber, shoplifter, terrorist, thief, thug |
| Crimes Against Persons | abduction, arsonist, assailant, assassin, assault, battery, homicide, kidnapper, manslaughter, offender, perpetrator, trafficker |
| Crimes Against Property | arson, burglary, larceny, looting, poaching, trespassing, vandalism |
| Financial & White-Collar Crimes | blackmail, bribery, collusion, counterfeit, corruption, extortion, forgery, insider-trading, money-laundering, perjury, ponzi-scheme, racketeering, smuggler, tax-evasion |
| Cybercrimes | cybercrime, malware, phishing, ransomware, spyware |
| Legal & Procedural Terms | accomplice, acquittal, arrest, contraband, conviction, culprit, felony, fugitive, indictment, inmate, misdemeanor, parole, prosecution, recidivism, suspect, warrant |

Table 2: Crime-related keywords used in Section 4.1.

## B.2  DISCOVERING IDENTITY-TOPIC ASSOCIATIONS WITH SAES

**Topics.** To describe possible themes, we construct a vocabulary of 2392 topics by merging two taxonomies: Google Cloud Natural Language API Categories[3] and IPTC Media Topics.[4]

Many of the resulting topics are semantically similar, such as "basketball", "$3 \times 3$ basketball", and "basketball equipment". We therefore group these topics into semantically coherent clusters. A cluster is valid if the pairwise cosine similarity of the `granite-embedding-english-r2` embeddings for all its topic labels is higher than a given threshold $\tau$.

We noticed that the PMI scores are sensitive to the specific clustering, which changes with the threshold. To account for this, we use a range of 20 thresholds with $\tau \in (0.8, 0.95)$. This interval was manually adapted for our specific embedding space. For each threshold, we create a new clustering, calculate PMI scores for each cluster, and assign the cluster's score to every topic label within it. The scores for each topic are then summed across all clusterings. Finally, we select the 20 individual topic labels with the highest summed PMI scores per intersectional identity.

**Computing Identity-Topic PMI Scores through SAEs.** Each SAE learns latent features $F_j$ that capture recurring patterns in the caption embeddings. To assign semantic meaning to these features, we embed all topic labels in the same space as the captions. In the case where we cluster the topics, we calculate a cluster embedding by concatenating the individual topic labels as a comma-separated list and embedding the entire resulting string. Then, for each SAE feature's decoder embedding, we retrieve its five most similar topics by cosine similarity, following Rao et al. (2024). The respective similarities are normalized to form a probability distribution $P(t \mid F_j)$ over topics for each feature.

---

[3] https://cloud.google.com/natural-language/docs/categories
[4] https://iptc.org/standards/media-topics/

We can use the decoder embeddings here because, through the reconstruction objective of the SAE, they already lie in the same semantic space as the caption embeddings.

We next estimate how strongly each feature relates to demographic groups. For this, we build a $14 \times d$ matrix (14 intersectional identity groups $\times$ $d$ SAE features), where each entry counts how often feature $j$ is activated in captions of group $i$. From this matrix, we derive $P(F_j)$ (activation frequency of each feature) and $P(i \mid F_j)$ (identity distribution given a feature). We combine these distributions to estimate how much a topic $t$ is associated with identity $i$. Using pointwise mutual information (PMI):

$$\text{PMI}(i, t) = \log \frac{P(i, t)}{P(i) \, P(t)}, \tag{2}$$

with probabilities defined as:

$$P(i) = \sum_j P(i \mid F_j) \, P(F_j) \tag{3}$$

$$P(t) = \sum_j P(t \mid F_j) \, P(F_j) \tag{4}$$

$$P(i, t) = \sum_j P(i \mid F_j) \, P(t \mid F_j) \, P(F_j) \tag{5}$$

where we assume identity and topic are conditionally independent given the feature $F_j$. To ensure robustness, we repeat this procedure across 24 SAE models: 12 trained with Top-$K$ sparsity (Gao et al., 2025) and 12 with Batch-Top-$K$ (Bussmann et al., 2024). Within each family, we vary hyperparameters (learning rate $\in \{10^{-3}, 10^{-4}\}$, expansion factor $\in \{16\times, 32\times, 64\times\}$, top-$K$ $\in \{20, 32\}$). The final PMI results are obtained by averaging over all runs. SAEs are trained using the `dictionary-learning` library (Marks et al., 2024).

**Full Results for all Intersectional Identities.** In Table 1, we summarize the top 20 topics associated with each of the 14 intersectional identities using five keywords. The full, unprocessed top 20 topics for each identity, including their PMI scores, are available in Table 3 for female identities and Table 4 for male identities.

The interpretation of most topics is straightforward or can be clarified by inspecting the texts that most activate the relevant SAE features. For example, we discard the topic "java (programming language)" for Southeast Asians, as it serves as a proxy for the island of Java. However, the role of a few other topics, such as "volcanic eruption" for Black women, remains unclear.

### B.3 POSSIBLE LABEL NOISE IN FACET

When evaluating our gender labeling model on the FACET dataset (Gustafson et al., 2023) in Section 3, we observe lower agreement with human labels than on the Phase dataset (Garcia et al., 2023). Consequently, we tested several CLIP models and MLLMs and found a consistent disagreement rate of 10% to 13%. Manual inspection reveals that a portion of the dataset appears to have unintuitive gender labels. Examples of these disagreements are shown in Fig. 11. We therefore conclude that lower agreement for gender labeling models on FACET is expected and is likely due to approximately 10% label noise in the dataset.

## C PERSON DETECTION

This section provides a detailed discussion on our methods to detect people in LAION-400M images (Appendix C.1) and an analysis of the minimum bounding box size required to reliably detect demographic attributes, such as perceived gender.

### C.1 USING YOLO11 TO DETECT PEOPLE IN IMAGES

We use Ultralytics `YOLO11` (Jocher & Qiu, 2024), a state-of-the-art object detection model, to find person bounding boxes in LAION-400m. Here, we provide more details about the bounding boxes detected in LAION-400m images using `YOLO11`. In particular, we report the statistics of detected

| Race (Female) | Top 20 Topics by PMI |
|---|---|
| Black | curriculum (1.28), curling (1.10), volcanic eruption (0.92), consumer confidence (0.80), forms guides & templates (0.79), poverty & hunger (0.78), policy towards indigenous people (0.72), hoop (rhythmic gymnastics) (0.67), skin conditions (0.60), convertibles (0.58), rings (artistic gymnastics) (0.57), poverty (0.54), hip hop (0.53), condiments & dressings (0.52), inventories (0.52), calf roping (0.51), countertops (0.51), hybrid & alternative vehicles (0.48), health and beauty product (0.46), cultural policy (0.45) |
| East Asian | anime & manga (1.29), acupuncture & chinese medicine (0.95), mitsubishi (0.88), traditional chinese medicine (0.83), confucianism (0.83), animated films (0.78), comics (0.76), pets & animals (0.73), dolls & accessories (0.73), suzuki (0.73), comics & animation (0.73), foreign language resources (0.73), school (0.73), adventure games (0.70), java (programming language) (0.63), language resources (0.55), independent and charter school (0.52), luggage & travel accessories (0.43), heart & hypertension (0.42), buddhism (0.41) |
| Latino | enduro (0.87), lamborghini (0.85), cultural policy (0.84), culture (0.83), fruits & vegetables (0.81), cultural development (0.79), women's rights (0.73), women's health (0.72), vitamins & supplements (0.72), eating disorder (0.69), eating disorders (0.69), convertibles (0.62), reproductive health (0.62), women (0.61), steroids & performance-enhancing drugs (0.57), nutrition (0.56), countertops (0.55), obstetrics/gynecology (0.55), reproductive medicine (0.50), health (0.48) |
| Middle Eastern | islam (2.30), freedom of religion (2.12), religious discrimination (1.98), sunni islam (1.96), shia islam (1.87), qur'an (1.72), eid al-adha (1.64), hasidism (1.48), religious facilities (1.43), hanukkah (1.41), data protection policy (1.07), hardware modding & tuning (0.92), religious education (0.91), archaeology (0.90), archeology (0.90), cultural policy (0.89), religion (0.86), cultural development (0.83), tariff (0.82), culture (0.80) |
| South Asian | hinduism (2.13), kabaddi (2.11), sikhism (2.01), folk & traditional music (2.00), jainism (1.91), bollywood & south asian films (1.90), folk music (1.82), traditional dance (1.75), sunni islam (1.75), shia islam (1.70), customs and tradition (1.58), saab (1.49), traditional chinese medicine (1.45), metal and mineral mining and refining (1.38), sepak takraw (1.34), fiber & textile arts (1.33), cultural policy (1.33), cultural development (1.25), textile arts (1.23), madison race (1.20) |
| Southeast Asian | cultural policy (1.03), culture (1.02), java (programming language) (0.98), cultural development (0.98), sepak takraw (0.91), fruits & vegetables (0.90), market and exchange (0.84), plant disease (0.81), energy market (0.80), acupuncture & chinese medicine (0.75), education policy (0.72), alternative & natural medicine (0.72), cooking & recipes (0.70), vitamins & supplements (0.70), kids & teens (0.60), customer services (0.57), medical tourism (0.54), education (0.52), teaching & classroom resources (0.51), flowers and plants (0.50) |
| White | beauty pageants (0.80), contraception (0.78), birth control (0.78), obstetrics/gynecology (0.73), bmx freestyle (0.73), flowers (0.72), women's rights (0.71), pregnancy and childbirth (0.71), women's health (0.70), surfing (0.68), windsurfing (0.65), flowers and plants (0.62), women (0.61), weddings (0.60), dress-up & fashion games (0.59), condiments & dressings (0.57), obgyn (0.56), reproductive health (0.53), surf & swim (0.53), bed & bath (0.52) |

Table 3: Top 20 topics in our person-centric captions by PMI for each female intersectional identity.

| Race (Male) | Top 20 Topics by PMI |
|---|---|
| Black | basketball (1.46), 3x3 basketball (1.29), canadian football (1.06), assault (1.04), basketball equipment (1.04), rugby league (1.03), american football (1.02), play-off championship (1.02), population growth (0.99), heptathlon (0.99), gaelic football (0.95), american football equipment (0.95), security measures (defense) (0.94), final game (0.91), sports officiating (0.90), decathlon (0.90), mormonism (0.89), fighting games (0.86), injury (0.84), rugby (0.82) |
| East Asian | anime & manga (1.56), comics & animation (1.21), comics (1.20), action & platform games (1.16), martial arts (1.08), adventure games (1.08), mixed martial arts (1.05), sombo (martial art) (1.00), action & adventure films (0.95), animated films (0.95), video game development (0.94), confucianism (0.93), energy resources (0.89), acupuncture & chinese medicine (0.88), energy and resource (0.86), video game (0.84), nuclear energy (0.82), action figures (0.79), firearms & weapons (0.78), traditional chinese medicine (0.75) |
| Latino | bullfighting (1.26), lamborghini (1.25), enduro (1.24), soccer (1.19), mormonism (0.92), jaguar (0.89), soccer equipment (0.81), rugby (0.78), citroën (0.76), horseback riding (0.76), bull riding (0.75), continental cup (0.74), horse driving (0.72), canadian football (0.70), livestock (0.69), sports facilities (0.68), saddle bronc (0.62), rugby league (0.62), rugby union (0.62), sport venue (0.59) |
| Middle Eastern | islam (1.57), firearms & weapons (1.40), sunni islam (1.34), shia islam (1.34), eid al-adha (1.30), military service (1.30), qur'an (1.27), military weaponry and equipment (1.26), military (1.23), civil and public service (1.22), zoroastrianism (1.19), combat sports equipment (1.15), military occupation (1.13), military equipment (1.12), drought (1.11), rifle shooting (1.10), judaism (1.04), civilian service (1.03), desserts (0.98), torah (0.90) |
| South Asian | kabaddi (2.10), cricket (1.90), hinduism (1.71), jainism (1.54), cricket equipment (1.53), bollywood & south asian films (1.52), sikhism (1.47), baseball (1.07), hornuss (1.00), judaism (0.94), shia islam (0.93), islam (0.93), international economic institution (0.87), public relations (0.85), international organization (0.84), political polls & surveys (0.82), bar and bat mitzvah (0.82), badminton (0.80), international relations (0.73), judge (0.70) |
| Southeast Asian | java (programming language) (1.17), buddhism (1.08), sepak takraw (1.06), mixed martial arts (0.86), meat & seafood (0.85), meat & seafood substitutes (0.84), taoism (0.78), labor market (0.76), flexible work arrangements (0.73), work & labor issues (0.70), culinary training (0.66), waste management (0.66), ramadan (0.65), yard maintenance (0.61), market and exchange (0.59), construction & power tools (0.59), kickboxing (0.55), workplace health and safety (0.54), martial arts (0.53), occupational health & safety (0.51) |
| White | male impotence (0.62), men's health (0.61), ice hockey (0.60), sledge hockey (0.59), aging & geriatrics (0.58), geriatric medicine (0.57), ageism (0.56), motorboat racing (0.52), men (0.50), senior citizens (0.47), auto racing (0.46), economy (0.45), seniors & retirement (0.42), motor car racing (0.40), motor sports (0.40), hockey (0.40), management (0.39), motorcycle racing (0.39), financial advisory service (0.38), business and finance (0.37) |

Table 4: Top 20 topics in our person-centric captions by PMI for each male intersectional identity.

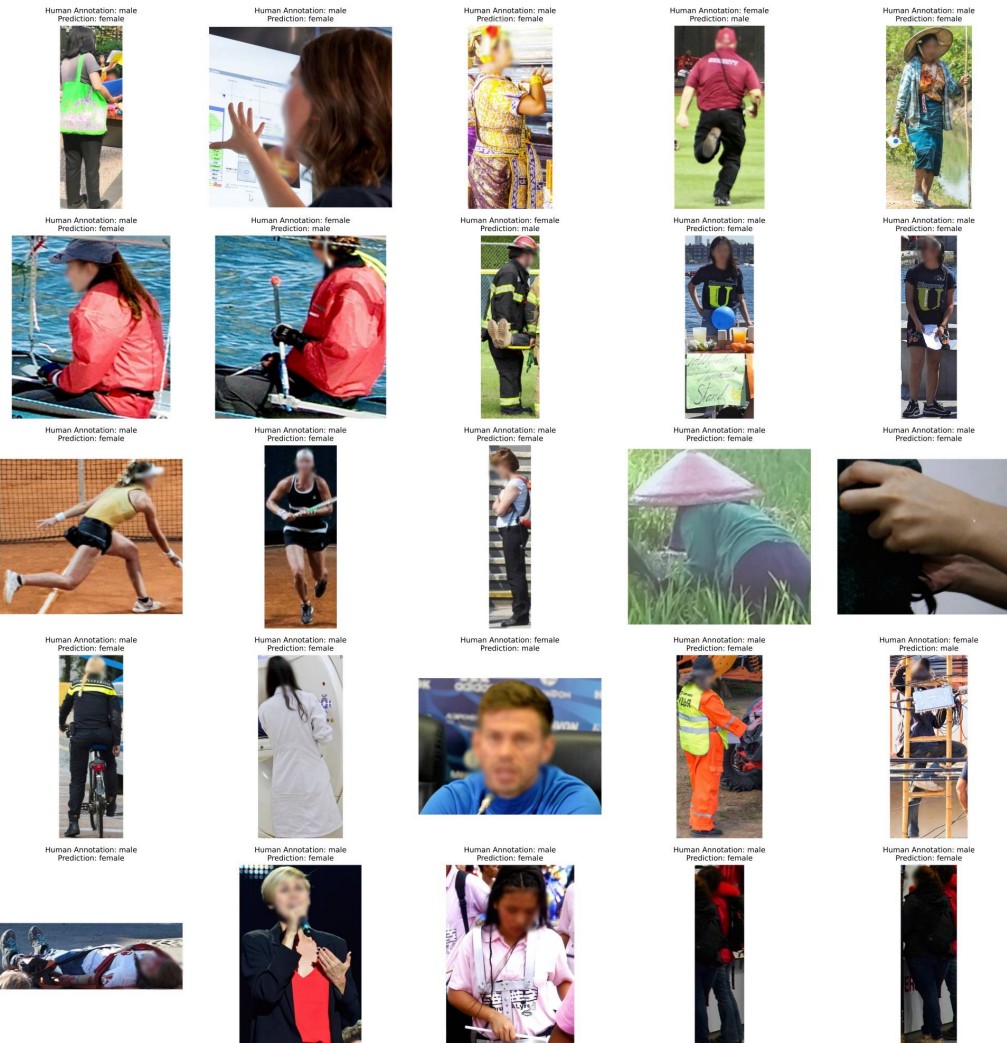

Figure 11: 25 bounding boxes sampled from the Facet dataset where both `InterVL3-2B` and CLIP `ViT-SO400M-14-SigLIP-384` models disagree with the human-provided gender annotations. Potential sources of disagreement are occlusion or ambiguity, but we also find instances where model predictions appear subjectively more accurate than the original human labels. Faced are blurred to protect the privacy of shown individuals.

bounding boxes in LAION-400M, and we show that person detection is accurate, using datasets commonly used to evaluate social bias. Leveraging demographic annotations, we also show that `YOLO11` does not show significantly different performance across gender and race groups.

Specifically, in this work, we use the `YOLO11-l` model, which has 25.3 million parameters. Since we are only interested in detecting people, we limit detections to COCO class 0 ("person"). The reported performance of `YOLO11-l` on the entire COCO 2017 Object Detection Task (Lin et al., 2014) validation set (not limited to people) is 53.4 mAP. This performance is similar to other models in the YOLO series. We further analyze the model's performance and potential biases below.

**Statistics.** `YOLO11-l` detects 267,434,822 person bounding boxes in 112,400,461 unique images. For comparison, we detect people in more than twice the number of images than LAION-Face (Zheng et al., 2022).

The maximum number of detected bounding boxes in a single image is 64. In Fig. 12, we show the number of unique images categorized by the number of detected person bounding boxes in the image. Over 50% of these images (64,507,400) contain only one detection. As expected, the number of images decreases as the number of detections per image increases.

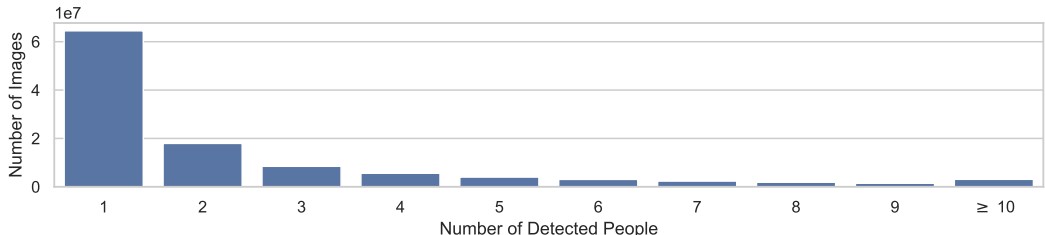

Figure 12: Number of unique images categorized by the number of detected person bounding boxes in the image. Over 50% of the images contain only one detection. The number of images decreases as the number of detections per image increases.

Fig. 13 shows the number of bounding boxes returned based on their covered image area (left) and the confidence score assigned by `YOLO11-l` (right). The results indicate that most detected bounding boxes are small. The left chart shows that 68.3% of bounding boxes cover less than 20% of the image area. The number of bounding boxes drops significantly as the area ratio increases. Fig. 13 (right) shows that 43.5% of bounding boxes are predicted with high confidence, as they fall within the 80-90% and 90-100% confidence intervals. These confidence scores represent the model's predicted probability that the detected object is a person and that the bounding box accurately localizes it. This implies that the model is highly confident in a large portion of its detections.

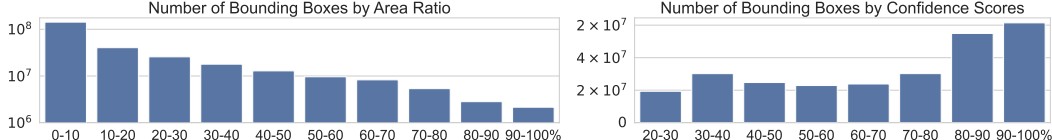

Figure 13: (Left) Number of bounding boxes by percentage of the original image (size $256 \times 256$) that each bounding box occupies. (Right) Number of bounding boxes by confidence score given to the bounding box by `YOLO11`. The lowest possible confidence score is 0.25. All ratios and confidence scores are grouped into 10% intervals.

**Model performance.** To assess the performance of `YOLO11-l` on more relevant tasks beyond the COCO 2017 Object Detection Task validation set, we also conduct a more specific evaluation using data that only contains bounding boxes of people. For this, we use the FACET (Gustafson et al., 2023) and Phase (Garcia et al., 2023) datasets. Both datasets include manually verified bounding boxes of people. They also have demographic information, such as gender, ethnicity, and skin tone: FACET contains 49,551 bounding boxes of people, and Phase contains 35,339 bounding boxes of people. We do not expect these datasets to significantly overlap with COCO, which was used to train `YOLO11`. This is because FACET images come from the Segment Anything 1 Billion dataset

(Kirillov et al., 2023), and Phase images come from Google Conceptual Captions (Sharma et al., 2018). However, neither dataset includes all the people in the provided images, but we still use these datasets to evaluate the recall of YOLO11. In this stage, focusing on recall is important because our priority is to detect all relevant people. False positives (i.e., bounding boxes that do not actually show people) can be filtered out later. Specifically, we calculate Recall@50 (where a detection is considered correct if its IoU with a ground truth box is at least 0.50), Recall@75, Recall@90, and mean recall (the average recall across various IoU thresholds from 0.5 to 0.95). In addition to YOLO11-l, we also evaluate YOLO11 models of various sizes.

The results are shown in Table 5. All models perform similarly on Phase, with smaller models having a slight advantage. On FACET, the models also perform similarly, but larger models show better performance. Performance of larger models (x, s variants) is in line with results reported by Gustafson et al. (2023). Since these results do not clearly favor any specific model, and we do not observe significant differences in processing speed (except for the largest model, YOLO11-x, which is noticeably slower), we chose the largest model that did not process data significantly slower than the others, i.e. YOLO11-l.

Models generally show strong recall performance on both the Phase and FACET person detection datasets, especially with less strict Intersection over Union (IoU) thresholds. Recall@50 and Recall@75 scores are consistently high. However, performance drops significantly at Recall@90. This suggests that there is some disagreement regarding the precise location of people in the images. Still, the mean recall values are good and indicate a solid overall balance across different IoU thresholds. We conclude that these results show that YOLO11 models perform well in person detection.

| | **Phase** | | | | **FACET** | | | |
| Variant | R@50 | R@75 | R@90 | mR | R@50 | R@75 | R@90 | mR |
|---|---|---|---|---|---|---|---|---|
| x | 0.98 | 0.91 | 0.62 | 0.82 | 0.96 | 0.89 | 0.71 | 0.84 |
| l | 0.98 | 0.91 | 0.63 | 0.83 | 0.95 | 0.89 | 0.69 | 0.84 |
| m | 0.99 | 0.92 | 0.64 | 0.84 | 0.95 | 0.87 | 0.67 | 0.82 |
| s | 0.99 | 0.93 | 0.67 | 0.85 | 0.93 | 0.84 | 0.61 | 0.79 |
| n | 0.98 | 0.93 | 0.67 | 0.84 | 0.89 | 0.78 | 0.53 | 0.73 |

Table 5: Recall of YOLO11 models (x, l, m, s, n variants) on Phase and FACET datasets.

**Qualitative Examples.** Fig. 14 shows examples of ground-truth bounding boxes in Phase that were detected by YOLO11-l using different IoU (Intersection over Union) thresholds. At IoU thresholds of 0.9 and 0.75, the predicted bounding boxes closely match the ground truth. However, at an IoU threshold of 0.5 (right), the predicted bounding box covers the entire person, whereas the ground-truth box focuses only on the torso and head. Additionally, in this image, YOLO11-l accurately detects a second person who is not annotated in Phase.

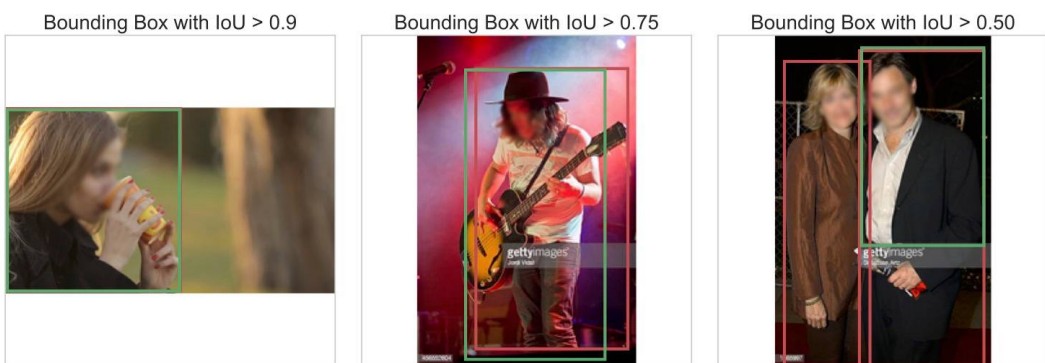

Figure 14: Person detection examples for Phase. Predicted bounding boxes are in red. Ground-truth bounding boxes are in green for IoU thresholds of 0.9, 0.75, and 0.5. Faces are blurred to protect the privacy of the shown individuals.

**Model Bias.** In addition to evaluating recall in person detection, we also assess whether `YOLO11-l` shows biases in detection performance. For example, we check if performance varies based on a person's perceived gender or ethnicity. It is important to evaluate this to prevent introducing bias into our annotations through the models used for automatic labeling. This is a significant issue, as previous research has found considerable bias in person object detection models (Buolamwini & Gebru, 2018; De Vries et al., 2019; Schwemmer et al., 2020), and in the COCO dataset (Meister et al., 2023), which was used to train `YOLO11`. To evaluate gender bias in `YOLO11-l`, we use demographic information from the Phase and FACET datasets. Phase provides annotations for perceived gender and ethnicity, while FACET provides perceived gender and skin tone. Skin tone is measured using the Monk Skin Tone scale, which ranges from 1 (lightest) to 10 (darkest).

Figure 15a shows the results for Phase. We find that `YOLO11-l` performs slightly differently based on gender and ethnicity. However, this difference in performance is very small, with the mean recall ranging from 82 to 83 points. It is worth noting that performance is better for male images than for female images, and among the annotated perceived ethnicities, recall is highest for people labeled "White". However, these trends are opposite in the FACET dataset (Fig. 15b). In this dataset, recall is higher for female-labeled images than for male-labeled images. Additionally, performance is better for darker skin tones compared to lighter skin tones. The generally higher performance for skin tone labels versus gender labels can be attributed to the fact that over 9,000 images in the FACET dataset (about 10% of the dataset) do not have skin tone annotations.

We conclude that performance differences between demographic groups are generally small. These differences seem to reflect variations in the datasets rather than biases of `YOLO11-l`. This is supported by the contradictory trends observed in the two evaluated datasets. Therefore, automatic person detection using `YOLO11` does not appear to introduce significant bias.

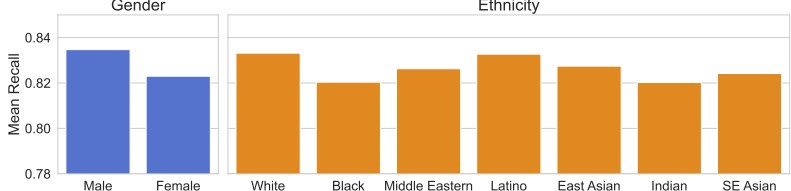

(a) Recall by perceived gender and ethnicity for the `YOLO11-l` model on the Phase dataset. The difference in performance across demographic groups is small, with mean recall ranging from approximately 0.82 to 0.83.

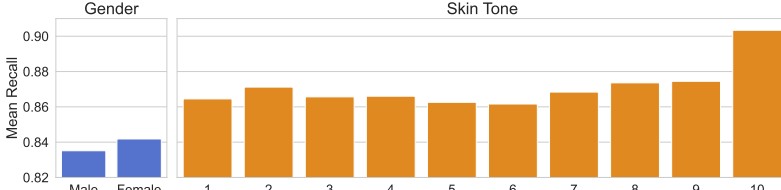

(b) Recall by perceived gender and skin tone for the YOLO11-l model on the FACET dataset. Around 10% of the dataset does not have skin tone annotations, which explains why performance for gender labels is around 3 points lower on average than performance for skin tone labels.

Figure 15: Recall of `YOLO11-l` by social categories (gender, ethnicity, and skin tone) on Phase and FACET datasets.

## C.2 FILTERING SMALL BOUNDING BOXES

Some detected person bounding boxes are very small. To set a minimum side length for a bounding box and remove all boxes with a side shorter than this threshold, we determine the minimum bounding box side length by investigating how well automatic perceived gender labeling agrees with human annotations at different image sizes. For this purpose, we sample a subset of 100 bounding boxes for each gender/ethnicity combination from the training split of the Phase dataset (Garcia et al., 2023). The sampled bounding boxes have a minimum side length of 70 pixels. Each bounding box is cropped and treated as a separate image. A few gender/ethnicity combinations do not have enough bounding boxes in the dataset, so we have a total of 1286 images. Each image is resized

while maintaining its aspect ratio to have a minimum side length from the following list (in pixels): 5, 10, 15, 20, 25, 30, 35, 40, 50, 60, 70, or the original size.

We use an MLLM (`InternVL3-2B`) and a CLIP model (`ViT-SO400M-14-SigLIP-384`) to predict the perceived binary gender of each image. We then evaluate the agreement between the predicted perceived gender and human labels of perceived gender. We find the minimum side length that achieves almost perfect agreement (Cohen's $\kappa > 0.8$ and accuracy $> 0.9$). The minimum side length determined in this way is the side length used for filtering the bounding boxes.

The results are shown in Fig. 16. As the minimum side length of the image increases, both CLIP and InternVL show better agreement. InternVL consistently performs better than the CLIP model at all resolutions, except for minimum side lengths of 5 and 10 pixels, where both models perform poorly. InternVL achieves a $\kappa$ value greater than 0.8 and an accuracy of 90% at the 30-pixel minimum side length. Because InternVL's performance reaches the desired level of agreement, we chose a minimum side length of 30 pixels for filtering the detected bounding boxes. Therefore, the smallest bounding box size is $30 \times 30$ pixels, which is about 1.4% of the standardized area of LAION-400m images ($256 \times 256$ pixels).

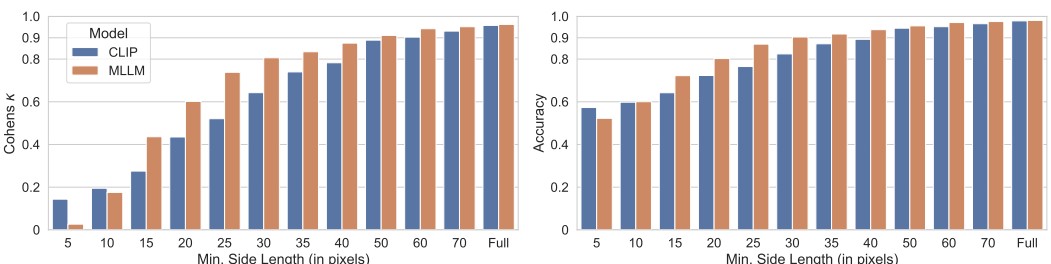

Figure 16: Agreement between automated and human-annotated perceived gender as a function of image resolution. We evaluate `InternVL3-2B` and CLIP `ViT-SO400M-14-SigLIP-384` based on Cohen's $\kappa$ (left) and accuracy (right) against human labels. Performance is shown for 1286 Phase training images resized to varying minimum side lengths in pixels.

## D  AUTOMATIC GENDER LABELING

To automatically label perceived binary gender, we train a custom classifier on a dataset built from LAION-400M. This ensures that the LAION-400M bounding boxes are in-distribution for the classifier's training data. This section describes how we build the dataset and details the hyperparameter study conducted to select the best model.

**Dataset Construction.** In a preliminary step, we label each bounding box as "male", "female", or "unclear" using `InternVL3-2B` (Zhu et al., 2025). This preliminary labeling, conducted as a feasibility study, allowed us to enrich our sample pool with images likely to contain our target classes before applying the more expensive multi-model consensus labeling.

We use these labels to sample 1,000,000 bounding boxes from each category, for a total of 3,000,000. These are then labeled by multiple MLLMs: `InternVL3-2B`, `Phi-3.5-Vision-Instruct` (Abdin et al., 2024), and `LLaVA-1.6-7B` (Liu et al., 2024). Using models with different base LLMs and training data avoids confounders, where models behave similarly due to their training.

For each of the 3,000,000 bounding boxes, we obtain a single gender label. The possible labels are "male" for a bounding box showing only male people, "female" for a bounding box showing only female people, "mixed" for a bounding box showing both male- and female-appearing people, and "unclear" for bounding boxes that do not show people or show people without clearly recognizable gender cues. These labels are exhaustive for perceived binary gender but do not cover nonbinary gender, as explained in Section 6. To obtain labels from MLLMs, we use the following prompt:

```
What is the gender of the person in the image? Only answer if you are
    sure and there are visual cues in the image about the person's gender
    . If you are not sure, choose the option 'gender is unclear / cannot
    tell / no person visible'.
```

```
A. only male person visible
B. only female person visible
C. both male and female people visible
D. gender is unclear / cannot tell / no person visible
Answer with a single letter.
```

To avoid label bias, such as a model preferring option "A" over "D" (Dominguez-Olmedo et al., 2024; Reif & Schwartz, 2024), we randomly permute the options in each prompt while keeping the letters fixed. After obtaining the response, we map the predicted letter to the corresponding label.

Next, we present the agreement statistics for the three models. The number of images where all models agree on the same label is as follows: Models agree on "male" for 703,756 bounding boxes, on "female" for 739,915 bounding boxes, on "unclear" for 735,713 bounding boxes, and on "mixed" for 18,997 bounding boxes. To evaluate the agreement among the models on all 3,000,000 bounding boxes, we calculate Fleiss' $\kappa$ (Fleiss & Cohen, 1973). The result is $\kappa = 0.735$, which indicates substantial agreement according to (Landis & Koch, 1977). In addition, we present pairwise agreement results for the three model combinations. This includes Cohen's $\kappa$ (Cohen, 1960) between each pair of models and the Jaccard Index for each label (male, female, mixed, unclear) on the bounding boxes predicted with that label.

| | `InternVL-Phi` | `InternVL-LLaVA` | `Phi-LLaVA` |
|---|---|---|---|
| Cohen's $\kappa$ | 0.748 | 0.750 | 0.719 |
| Jaccard Index male | 0.705 | 0.739 | 0.716 |
| Jaccard Index female | 0.757 | 0.764 | 0.721 |
| Jaccard Index mixed | 0.269 | 0.018 | 0.008 |
| Jaccard Index unclear | 0.668 | 0.651 | 0.649 |

Table 6: Pairwise agreement statistics between the three MLLMs used for data labeling: `InternVL` (`InternVL3-2B`), `Phi` (`Phi-3.5-Vision-Instruct`), and `LLaVA` (`LLaVA-1.6-7B`). The table shows Cohen's $\kappa$ for overall agreement across all four labels and the Jaccard Index for the set of bounding boxes assigned each specific label. All metrics are calculated on the 3,000,000 bounding boxes sampled for dataset construction.

The results are in Table 6. They show that the models have substantial, but not perfect, agreement across all categories. The only outlier is `LLaVA-1.6-7B`, which rarely predicts "mixed". We attribute this to the generally weaker performance of `LLaVA-1.6-7B`, which is also the oldest model and based on the weakest LLM, `Vicuna-7B` (Chiang et al., 2023). We therefore add `Qwen2.5-VL-7B-Instruct`, which is a recent and strong MLLM, as a fourth model to generate higher-quality labels for the "mixed" category.

Interestingly, the agreement for the "male" and "female" categories is also imperfect. This is due to the instability of the "unclear" category, as there is no clear standard regarding what qualifies as a visual gender cue. These results highlight the importance of using a consensus from multiple models, as labels from a single model can be heavily influenced by its specific behaviors and potential spurious correlations.

From the labeled bounding boxes, we sample a balanced dataset of 100,000 images, composed of 25,000 samples for each of the four categories. For the "male", "female", and "unclear" categories, these were randomly sampled from the pool of images where our initial three MLLMs were in unanimous agreement. To form the "mixed" category, we took all 18,997 images where `InternVL3-2B`, `Phi-3.5-Vision-Instruct`, and `Qwen2.5-VL-7B-Instruct` agreed, and supplemented them with 6,003 randomly sampled images where two of these three models agreed.

We note that this modified protocol for the "mixed" category, while necessary to gather sufficient samples, results in labels that are not validated to the same level of consensus as the other three categories. The strong performance of our final classifier on this category (see Table 7) suggests this approach was nonetheless effective.

**Model Training.** We finetune pretrained CLIP models from OpenCLIP (Ilharco et al., 2021), specifically `ViT-B-16`, `ViT-B-16-SigLIP`, and `ViT-L-14-quickgelu`, on the training

split (80%) of our dataset. We conduct a hyperparameter study to find the best model. For each of the three pretrained models, we evaluate the following hyperparameter grid:

| Batch Size | Number of Epochs | Learning Rate |
|---|---|---|
| 16, 32, 128 | 2, 5, 10, 15 | $5 \times 10^{-5}, 1 \times 10^{-5}, 5 \times 10^{-6}$ |

In all cases, we use the AdamW optimizer (Loshchilov & Hutter, 2019) with a OneCycle learning rate scheduler (Smith & Topin, 2019). The scheduler uses a cosine annealing strategy, with annealing beginning at 30% of the total training steps. During the first epoch, we train only the new classification layer using the AdamW optimizer with a constant learning rate of $1 \times 10^{-3}$. This aligns the classification weights with the pretrained embedding space and helps mitigate catastrophic forgetting in the encoder during the early training phase. After each epoch, we evaluate the accuracy on the validation split (10%) and keep the best checkpoint.

After training models for all hyperparameter combinations, we select the one with the highest accuracy on the validation split. The best model is a `ViT-B-16-SigLIP` trained for 2 epochs with a batch size of 32 and a learning rate of $5 \times 10^{-6}$. This model achieves 97.24% accuracy on the validation split and 97.18% accuracy on the test split. The following table shows the detailed class-wise metrics: These results show that the trained model performs well on all four categories. Precision

| | Precision | Recall | F1-Score | Support |
|---|---|---|---|---|
| Female | 0.97 | 0.97 | 0.97 | 2500 |
| Male | 0.98 | 0.96 | 0.97 | 2500 |
| Mixed | 0.95 | 0.97 | 0.96 | 2500 |
| Unclear | 0.99 | 0.99 | 0.99 | 2500 |
| Accuracy | | | 0.97 | 10000 |
| Macro Avg | 0.97 | 0.97 | 0.97 | 10000 |
| Weighted Avg | 0.97 | 0.97 | 0.97 | 10000 |

Table 7: Gender labeling performance of the finetuned model on the 10,000-image test split.

for the "mixed" category is slightly lower than for other categories. However, this is expected, as this category exhibited the lowest agreement during the MLLM annotation phase (see Table 6). Given these strong results, we use this model to label the perceived binary gender for all bounding boxes in LAION-400M.

## E    LABELING RACE/ETHNICITY

### E.1    COLLECTING A RACE/ETHNICITY DATASET

We create a dataset of people perceived as belonging to a certain race or ethnicity by cropping person bounding boxes from the LAION-400M dataset. The selection process has two stages. First, we identify images from LAION-400M whose alt-text captions contain keywords associated with specific race/ethnicity categories. These keywords include country names and other terms related to the category. From these images, we crop all person bounding boxes and sample one million of them per category for the next stage. In the second stage, three MLLMs label each cropped bounding box. If all three MLLMs assign the same perceived race/ethnicity as suggested by the image caption, we assume the person in the image has visual properties commonly associated with that race or ethnicity.

**Race/Ethnicity Categories.** It is impossible to create an objectively valid set of race/ethnicity categories. Any practical set of categories will inevitably exclude many of the world's populations and their identities. Despite this fundamental limitation, we aim to create a set of categories that is as useful as possible for the research community. Therefore, we closely follow previous work (Karkkainen & Joo, 2021; Seth et al., 2023; Garcia et al., 2023) and use the following categories:

Black, East Asian, Latino / Hispanic, Middle Eastern / North African, South Asian, Southeast Asian, White

However, this list inherits inconsistencies from previous labeling systems. Some categories, such as Black and White, are linked to skin tone as a clear visual feature. Other categories refer mainly to geographical ancestry (East Asian, Southeast Asian, South Asian) or ethnicity (Latino / Hispanic), making their visual cues less clear.

**Identifying Image Candidates.** For each race/ethnicity category, we retrieve candidate images from LAION-400M that might contain people stereotypically associated with that race or ethnicity. For this, we use keywords that are either country names or terms related to the category.

For country names, we map categories to geographical regions according to the United Nations geoscheme subregions. Each race/ethnicity category is linked to the subregions with which it is most stereotypically associated. We then use the names and adjectives of all countries in the matched subregions as keywords to search for candidate images. The list of race/ethnicity categories, matched subregions, and country names is in Table 8 (we only list country nouns there but also use country adjectives). We make the following changes to the UN list: Iran is moved from South Asian to Middle Eastern / North African; Afghanistan is removed from South Asian; Israel is removed from Middle Eastern / North African; Sudan is moved from Middle Eastern / North African to Black; and the United States is added to Black to include people perceived as African American. These individual choices are subjective, but we believe they result in a more useful keyword list.

The race/ethnicity-related terms are listed in Table 9. Some of these terms are derogatory or offensive. We include them because racial stereotypes are often most harmful when combined with such content. Additionally, some terms overlap with country names or adjectives, but we keep the two lists separate for clarity.

For each keyword, we retrieve all images from LAION-400M whose alt-text captions contain that keyword. If a keyword is a multi-word term, we retrieve an image only if its caption contains all the individual words. Both captions and keywords are lowercased and lemmatized, and stop words are removed. From all retrieved images, we extract the person bounding boxes. We only keep boxes that are labeled as "male" or "female" and discard those labeled "unclear". Finally, we sample a set of 1,000,000 bounding boxes per race/ethnicity category (7,000,000 in total) for labeling.

**Labeling Images.** To label these 7,000,000 images for perceived race/ethnicity, we use three different MLLMs: `LLaVA1.6-7B` (Liu et al., 2024), `Phi-3.5-Vision-Instruct` (Abdin et al., 2024), and `InternVL3-2B` (Zhu et al., 2025). By selecting models with different language components and training data, we aim to identify images that consistently show common visual cues associated with certain races or ethnicities. All three models receive the same prompt:

```
What is the race of the person in the image? Only answer if you are sure
    and there are visual cues in the image about the person's race or
    ethnicity. If you are not sure, choose the option 'race is unclear /
    cannot tell / no person visible'.
A. White, European or Caucasian
B. Black, African, or African American
C. East Asian, such as Chinese, Japanese, or Korean
D. Latino, Hispanic or South American
E. Middle Eastern, Arab or North African
F. Indian, Pakistani, or Bangladeshi
G. Southeast Asian, such as Filipino, Thai, or Vietnamese
H. race is unclear / cannot tell / no person visible
Answer with a single letter.
```

To mitigate label bias (Dominguez-Olmedo et al., 2024), we randomly permute the options for each image while keeping the letters and their order fixed. The prompt explicitly instructs the models to select a race/ethnicity only when clear visual cues are present and provides detailed descriptors for each category. Models predict a single letter, which is then mapped to a normalized category.

An image,i.e. a cropped person bounding box, is considered to represent features related to a race or ethnicity if all three models agree on the category, and this agreed-upon category also matches the one suggested by the keyword used to retrieve the image. For instance, if an image was retrieved using the keyword "India", all three models must classify the person's race/ethnicity as South Asian for it to be included. However, the agreement rates differ significantly across the categories. For

| Race/Ethnicity | Regions | Countries |
| --- | --- | --- |
| Black | Western Africa; Eastern Africa; Southern Africa; Middle Africa | Angola; Benin; Botswana; Burkina Faso; Burundi; Cabo Verde, Cape Verde; Cameroon; Central Africa; Chad; Comoros; Congo; Cote d'Ivoire, Ivory Coast; Djibouti; Equatorial Guinea; Eritrea; Eswatini, Swaziland; Ethiopia; Gabon; Gambia; Ghana; Guinea; Guinea-Bissau; Kenya; Lesotho; Liberia; Madagascar; Malawi; Mali; Mauritania; Mauritius; Mozambique; Namibia; Niger; Nigeria; Rwanda; Sao Tome, Principe; Senegal; Seychelles; Sierra Leone; Somalia; South Africa; South Sudan; Tanzania; Togo; Uganda; Zambia; Zimbabwe |
| East Asian | Eastern Asia | China; Japan; Mongolia; North Korea; Korea; Taiwan |
| Latino | Central America; South America; Caribbean | Antigua, Barbuda; Argentina; Bahamas; Barbados; Belize; Bolivia; Brazil; Chile; Colombia; Costa Rica; Cuba; Dominica; Dominican Republic; Ecuador; El Salvador; Grenada; Guatemala; Guyana; Haiti; Honduras; Jamaica; Mexico; Nicaragua; Panama; Paraguay; Peru; Saint Kitts, Nevis; Saint Lucia; Saint Vincent, Grenadines; Suriname; Trinidad, Tobago; Uruguay; Venezuela |
| Middle Eastern / North African | Western Asia; Northern Africa | Algeria; Armenia; Azerbaijan; Bahrain; Cyprus; Egypt; Georgia; Iran; Iraq; Israel; Jordan; Kuwait; Lebanon; Libya; Morocco; Oman; Palestine; Qatar; Saudi Arabia; Sudan; Syria; Tunisia; Turkey; United Arab Emirates, UAE; Yemen |
| South Asian | Southern Asia | Bangladesh; Bhutan; India; Maldives; Nepal; Pakistan; Sri Lanka |
| Southeast Asian | South-eastern Asia | Brunei; Cambodia; Indonesia; Laos; Malaysia; Myanmar, Burma; Philippines; Singapore; Thailand; Timor-Leste, East Timor; Vietnam |
| White | Western Europe; Eastern Europe; Southern Europe; Northern Europe; Northern America | Albania; Andorra; Australia; Austria; Belarus; Belgium; Bosnia, Herzegovina; Bulgaria; Canada; Croatia; Czechia, Czech Republic; Denmark; Estonia; Finland; France; Germany; Greece; Hungary; Iceland; Ireland; Italy; Kosovo; Latvia; Liechtenstein; Lithuania; Luxembourg; Malta; Moldova; Monaco; Montenegro; Netherlands, Holland; New Zealand; North Macedonia; Norway; Poland; Portugal; Romania; Russia; San Marino; Serbia; Slovakia; Slovenia; Spain; Sweden; Switzerland; Ukraine; United Kingdom, UK, Great Britain; United States, USA, America; Vatican, Vatican City, Holy See |

Table 8: Country keywords associated with each race/ethnicity category used to retrieve candidate images from LAION-400M for labeling.

| Race/Ethnicity | Related Expressions |
|---|---|
| White | Aryan; Caucasian; Caucasoid; European; Europid; Indo-European; Of European descent; Paleface; White |
| Latino | Hispanic; Latino; Latina; Latinx; Latine; Latin@; Chicano; Chicana; Xicano; Xicana; Latin American; Tejano; Nuyorican |
| Black | Black; African American; Afro-American; African; People of African descent; Person of color; Afro-Caribbean; Afro-Latino; Black British; African Canadian; Colored; Negro |
| East Asian | East Asian; Oriental; Asian; Far East Asian; East Asian person; Person of East Asian descent; Han Chinese; Japanese; Korean; Mongolian |
| Southeast Asian | Southeast Asian; Person of Southeast Asian descent; ASEAN national; Indochinese; Filipino; Vietnamese; Cambodian; Thai; Malaysian; Indonesian; Singaporean; Burmese; Laotian; Hmong; Timorese; Bruneian |
| South Asian | South Asian; Desi; Indian; Pakistani; Bangladeshi; Person of South Asian descent; Indian subcontinent native; Sri Lankan; Nepalese; Hindustani; Asian |
| Middle Eastern / North African | Middle Eastern; West Asian; Arab; Persian; Turkic; Kurdish; Levantine; Maghrebi; North African; Person of Middle Eastern descent; MENA |

Table 9: Race/Ethnicity-related expressions associated with each category used to retrieve candidate images from LAION-400M for labeling.

example, the models agree on the "Black" category in 335,004 cases (33.5%), but on the "Latino" category in only 26,809 cases (2.7%). Agreement statistics for all categories are shown in Fig. 17.

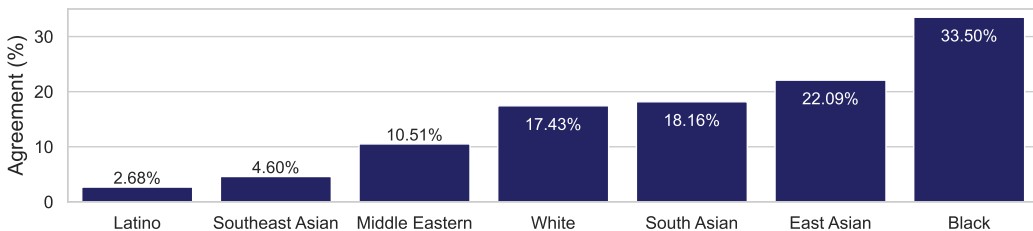

Figure 17: Agreement rates of MLLMs for the different race/ethnicity categories, i.e. the ratio of 1,000,000 images per category where all 3 MLLMs give the same label as suggested by the keyword that retrieved the image.

**Compiling the Final Dataset.** Using the labels obtained from the MLLMs, we compile a balanced dataset with 25,000 cropped person bounding boxes per race/ethnicity category. We chose 25,000 samples because the category with the fewest images, "Latino", has only 26,809 agreed-upon labels. The images are randomly sampled while attempting to balance keywords and gender to ensure geographical diversity and gender representation. A perfect balance is not possible, as some keyword-gender combinations have very few retrieved or labeled images.

Furthermore, we add two additional sets of images. The first set contains 25,000 images where all three MLLMs agreed on the "unclear" label. The second set contains 16,676 images where the models all disagreed with each other and with the race/ethnicity category suggested by the retrieval keyword. We label this set "disagreement". In total, the dataset contains 216,676 images that represent either visual cues associated with our considered race or ethnicity labels or, in the case of the two control groups, an absence of them. Finally, we create three stratified splits for training, validation, and testing/calibration. Each class has 20,000 training images, 2,500 validation images, and 2,500 test images.

### E.2 Training a Race/Ethnicity Classifier

Using the training portion of our dataset collected in Appendix E.1, we fine-tune a pretrained CLIP model to predict the race/ethnicity labels from images. As base model, we use the `ViT-B-16-SigLIP` model trained on `webli`. This model is provided by OpenCLIP. To optimize performance, we conduct an extensive hyperparameter study involving the learning rate, batch size, and the number of training epochs. We train models using two grids of parameters. The first set is

| Batch Size | Number of Epochs | Learning Rate |
|---|---|---|
| 16, 32, 128 | 1, 10, 20, 50 | 1e-4, 5e-5, 1e-5 |

and this set determined that 10 or 20 epochs and a learning rate of 1e-5 consistently yields best results, while the batch size has less impact. Based on these findings, we used a second grid

| Batch Size | Number of Epochs | Learning Rate |
|---|---|---|
| 32, 64, 128, 256 | 10, 15, 20 | 2e-5, 1e-5, 5e-6 |

to further refine performance. In all cases, we use the AdamW optimizer (Loshchilov & Hutter, 2019) paired with a OneCycle learning rate scheduler (Smith & Topin, 2019) using a cosine annealing strategy and starting learning rate annealing at 30% of overall training steps. During the first epoch, we only train the newly introduced classification layer using AdamW with constant learning rate (1e-3) to align the classification weights with the pretrained embedding space and mitigate catastrophic forgetting in the encoder in the early phase of training. Finally, for training the classifier, we do not use the images labeled "disagreement", but we include them for validation.

After training models for all hyperparameter combinations, we select the best-performing model on the validation split, which uses batch size 16, 20 epochs, and learning rate 1e-5. This model achieves 87.4% raw accuracy on the validation split and 87.3% accuracy on the test split. The label-wise accuracies on the test split are in Table 10, and the detailed confusion matrix in Fig. 18.

These results show that the model achieves strong overall performance across race/ethnicity groups, but with notable differences between some categories. Interestingly, classes that are underrepresented in LAION-400M, such as Latino and Middle Eastern, achieve the highest precision and recall rates. In contrast, the East Asian and Southeast Asian categories perform notably worse, with F1-scores of $0.87$ and $0.84$, respectively, mainly due to high confusion between them. This demonstrates the difficulty of distinguishing between these groups based on visual features alone. The "unclear" class has moderate precision and high recall, but the confusion matrix shows that White individuals are misclassified as "unclear" more often than individuals from other groups. This suggests that the classifier may systematically under-identify White individuals in practice, which implies that the proportion of White people in LAION-400M is likely even higher than the considerable overrepresentation we measured in Section 4.1.

| | Precision | Recall | F1-score |
|---|---|---|---|
| Black | 0.94 | 0.91 | 0.92 |
| East Asian | 0.88 | 0.86 | 0.87 |
| Latino | 0.92 | 0.91 | 0.91 |
| Middle Eastern | 0.92 | 0.92 | 0.92 |
| South Asian | 0.92 | 0.89 | 0.91 |
| Southeast Asian | 0.84 | 0.84 | 0.84 |
| Unclear | 0.76 | 0.83 | 0.79 |
| White | 0.90 | 0.87 | 0.89 |
| Accuracy | | | 0.87 |
| Macro avg | 0.89 | 0.88 | 0.88 |
| Weighted avg | 0.88 | 0.87 | 0.87 |

Table 10: Performance in perceived race/ethnicity labeling of the finetuned model on the 20,000-image test split.

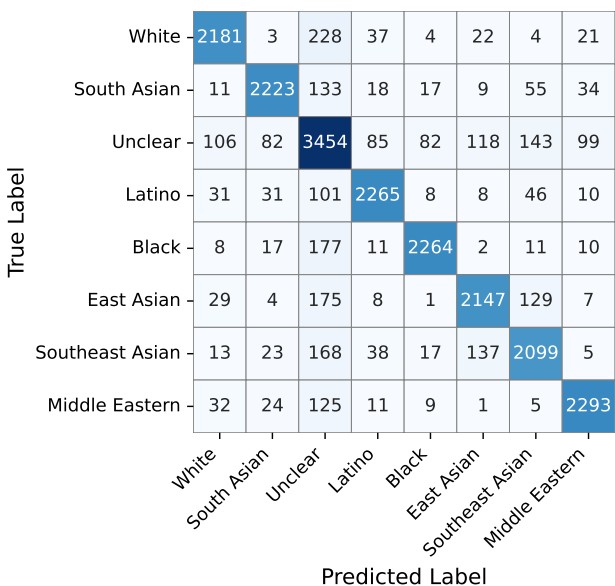

Figure 18: Confusion matrix of the race/ethnicity classifier on the test split.

We additionally analyze the correlation of logits on all bounding boxes in LAION-400M. The result in Fig. 19 shows that correlations partially agree with human-human disagreements identified in Section 3, particularly in Fig. 4. Logits for East Asian and Southeast Asian are clearly correlated. Also, logits for White are slightly positively correlated with logits for Latino and Middle Eastern. This also holds for the logits for Southeast Asian and the logits for Latino and South Asian. All these confusions are also found in Fig. 4. One notable case is the correlation of Black and Unclear, which could indicate that the classifier classifies more individuals perceived as Black as Unclear.

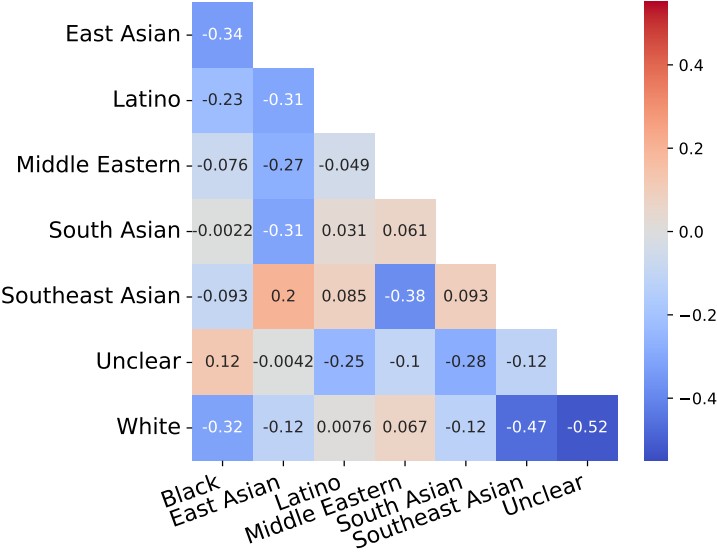

Figure 19: Pairwise correlation of logits for race/ethnicity classes on all bounding boxes.

# F  PERSON-CENTRIC CAPTIONS

We generate synthetic descriptions for each detected person. Synthetic captions are generated by MLLM captioning models. This approach is similar to existing recaptioning methods used to improve the quality of multimodal pretraining datasets (Chen et al., 2024; Liu et al., 2023; Fan et al., 2023; Nguyen et al., 2023; Li et al., 2024). To provide informative captions, we prompt models to include rich context and detailed descriptions. This requires using the strongest available MLLMs for captioning that fit our computational budget. However, the main challenge is how to provide bounding box information to the model so that it focuses on the specified person while also considering the entire image context. Although models have been developed that allow specifying bounding boxes as coordinates in their prompts (Chen et al., 2023a; Zhang et al., 2024a; Peng et al., 2024) or visually within the image (Cai et al., 2024; Lin et al., 2025), the performance of these models is significantly lower than that of generalist models like InternVL (Zhu et al., 2025) or Qwen-VL (Bai et al., 2025). This suggests to use the stronger generalist models to generate synthetic captions, even though they do not currently support bounding boxes natively in their prompts.

However, previous research (Shtedritski et al., 2023) found that large vision-language models are already aware of visual markers in images. While Wu et al. (2024a) find that this may not be true for all MLLMs, we find that the latest generation of open MLLMs, specifically InternVL and Qwen-VL-2.5, indeed show strong awareness of visual markers. Using this ability, we draw a red bounding box around the person we want to caption and ask the model to describe the highlighted person. Although drawing the bounding box as a frame in the image risks creating occlusions, we find this problem to be generally not significant. The detailed prompt used to generate captions, which was manually engineered using a small sample of bounding boxes, is:

```
Describe the highlighted person in one continuous paragraph. Describe the
context, how the person looks and what the person is doing. Focus on
objective details and avoid any subjective or emotional descriptions.
Only focus on the highlighted person. Be as detailed and precise as
possible. Write in plain, objective and scientific language.
```

Using this prompt, we compare four MLLM models on a subset of 500 bounding boxes to find the MLLM that provides the highest quality captions. We present the image alongside captions generated by two models to GPT-5.1 (medium reasoning effort) and ask it to select the better caption. The exact prompt we use is below.

We then use the model with the highest win rate to caption all bounding boxes in LAION-400M. The models we compare are `InternVL3-2B` and `InternVL3-8B`, and `Qwen2.5-VL-3B-Instruct` and `Qwen2.5-VL-7B-Instruct`. `InternVL3-8B` achieves the highest win rates: `0.756` against `Qwen2.5-VL-3B`, 0.630 against `Qwen2.5-VL-7B` and 0.582 against `InternVL3-2B`. Overall, while hallucinations and inaccuracies are present in captions generated by any model, especially when image quality is low, many people are visible in the image, or the image content is complex, we find that InternVL models suffer from fewer hallucinations than Qwen-VL models. Comparing `InternVL3-2B` and `InternVL3-8B`, we find that both models generally provide accurate descriptions, but captions by `InternVL3-8B` appear to get details correct more often than captions generated by `InternVL3-2B`.

In Table 11, we show examples of person descriptions generated by `InternVL3-8B`. The model correctly infers the intended person from the added bounding box, even if there is some overlap between the bounding box and other unrelated people in the image. The descriptions are mostly correct and contain relatively few speculations or subjective statements about the images. Descriptions also contain elements of the background or other aspects of the image, which is intended, so the context can be understood.

> **LLM-as-a-judge for Caption Comparison Prompt**
>
> You are judging two captions for a single person highlighted by a red frame in the provided image. Decide which caption better describes the framed person, or declare a tie.
>
> Return exactly one of the following and nothing else: Model A, Model B, or TIE.

Scope: Evaluate only the person and visual evidence inside the red frame. Ignore unframed regions and do not use outside knowledge.

Rubric (in priority order):
1) Faithfulness: Does the caption accurately match what is visible about the framed person (appearance, attributes, actions, relationships to visible context)?
2) No hallucinations / correctness: Penalize any invented or unverifiable claims (e.g., names, occupations, backstories, emotions, ages, nationalities, medical or moral judgments) unless explicitly visible (e.g., a readable name tag).
3) Detail & specificity: Prefer precise, correct details (colors, garments, accessories, pose, interactions with clearly visible objects/context relevant to the framed person).
4) Linguistic quality: Prefer fluent, grammatical, concise, and helpful wording.

Tie guidance:
- Output TIE if quality is indistinguishable or both captions have similar severity of errors.
- If one caption has a faithfulness error and the other does not, choose the faithful one.
- If both are faithful and correct, choose the one with better correct detail and clearer language.

Edge cases:
- Treat unverifiable inferences as hallucinations unless clearly supported by visible text/symbols.
- If both captions are largely incorrect or speculative, output TIE.

Think through your decision silently and output only the final label with no punctuation, whitespace, or explanation.

Model A: {caption_a}
Model B: {caption_b}

Final answer (one of: Model A, Model B, TIE):

**Caption Bias Analysis.** To investigate potential bias in generated captions, we examine their sentiment and the ratio of crime-related keywords across gender and race/ethnicity groups. For sentiment analysis, we compute the VADER score, as described in Section 4.2. For crime-related keywords, we verify whether any keyword listed in Table 2 appears in the caption. We then calculate the average VADER score for each gender and race/ethnicity group, along with the ratio of captions for each group that contain at least one crime-related keyword.

The results are in Fig. 20. The prevalence of crime-keywords is very low, between $0.1\%$ and $0.3\%$. Note that such keywords appear for several reasons, not all of which are harmful. For example, the word may be visible as text in the image, or a keyword might have multiple meanings that are benign. None heless, the ratio of captions containing crime-related keywords is nearly three times higher for male-gendered bounding boxes than for female-gendered bounding boxes. It is also twice as high for persons labeled Middle Eastern as for persons labeled White.

We observe similar trends for VADER sentiment scores. Average sentiment is lower for male-gendered bounding boxes than for female-gendered ones. It is also lower for individuals labeled as Middle Eastern than for those labeled as White. However, the average sentiment for White and Black groups is approximately the same. Overall, sentiment scores are significantly higher in generated captions than in the original alt-text captions because the generated captions are longer and provide more space for positive expressions. The MLLM employed is also finetuned for alignment, i.e. safety and helpfulness, which mitigates potential issues with inappropriate content such as slurs or hate speech.

Overall, it is challenging to distinguish between data-induced bias and model-induced bias. For example, images featuring Middle Eastern-looking people often contain content indicating armed conflict, which is inherently negative. Due to the low prevalence of crime-related keywords and generally high sentiment scores, we conclude that the effects from the model are not pronounced and the captions are overall safe.

| Image | Caption by `InternVL3-8B` |
|---|---|
| 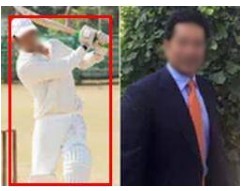 | The highlighted individual is a person dressed in traditional white cricket attire, including a helmet, gloves, and pads, indicating they are engaged in playing cricket. The person is holding a cricket bat and appears to be in the midst of executing a shot, with their body positioned in a batting stance. The background suggests an outdoor cricket field, with a visible stumps setup, reinforcing the context of a cricket match or practice session. The person's focused posture and equipment clearly denote active participation in the sport. |
| 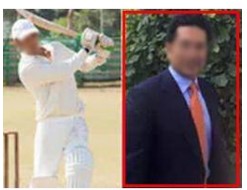 | The highlighted person is a man standing outdoors with a background of green foliage. He is dressed in formal attire, consisting of a dark suit jacket, a light blue dress shirt, and an orange tie. His hair is dark and styled neatly. The man is facing forward, and his expression appears neutral. The image is a close-up, focusing on his upper body and head. |
| 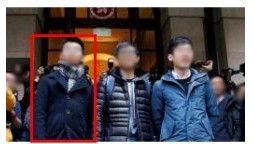 | The highlighted individual is a person standing in a group of three, positioned slightly to the left. This person is wearing a dark blue overcoat and a scarf wrapped around their neck. They have short hair and are looking straight ahead. The background suggests they are outside a building with a crowd and photographers around, indicating a public or media event. The person's posture is upright, and they appear to be part of a formal or significant occasion, as evidenced by the presence of onlookers and cameras. |
| 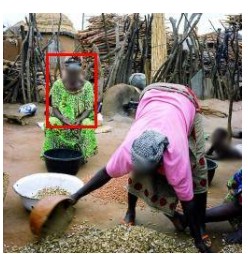 | The highlighted individual is a person seated on the ground, wearing a vibrant green dress with a patterned design. The person has a head wrap that appears to be dark in color. They are seated in an outdoor setting, surrounded by what looks like a rural environment with wooden structures and scattered materials in the background. The person's hands are resting on their lap, and they are positioned near a black basin and a large white bowl filled with grains or seeds. The scene suggests an agricultural or food processing activity, with other individuals nearby engaged in similar tasks. |
| 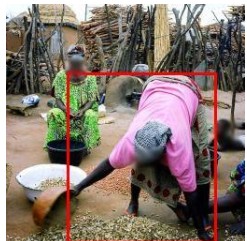 | The highlighted individual is engaged in manual labor, specifically sifting or sorting grains or seeds. This person is wearing a pink shirt and a patterned skirt with red circular designs. A gray headscarf covers their head. They are bent over, using a wooden bowl to transfer contents onto the ground, which is covered with scattered grains. The setting appears to be an outdoor rural area, with traditional structures and wooden materials in the background. Another person, dressed in a green patterned outfit, is seated nearby, observing or waiting. The ground is bare earth, and various containers are visible, indicating a communal or agricultural activity. |
| 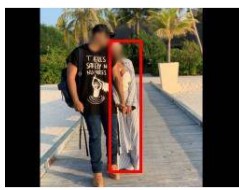 | The highlighted person is wearing a long, striped garment that reaches the ankles, paired with yellow footwear. The individual is holding hands with another person, who is dressed in a black t-shirt with white text and graphics, blue jeans, and brown shoes. They are walking on a wooden pathway, surrounded by a tropical setting with palm trees and greenery in the background. The person's head is covered with a white cloth, and they appear to be engaged in a leisurely stroll. |

Table 11: Examples of images with person descriptions generated by `InternVL3-8B`. Faces are blurred to protect the privacy of the shown individuals.

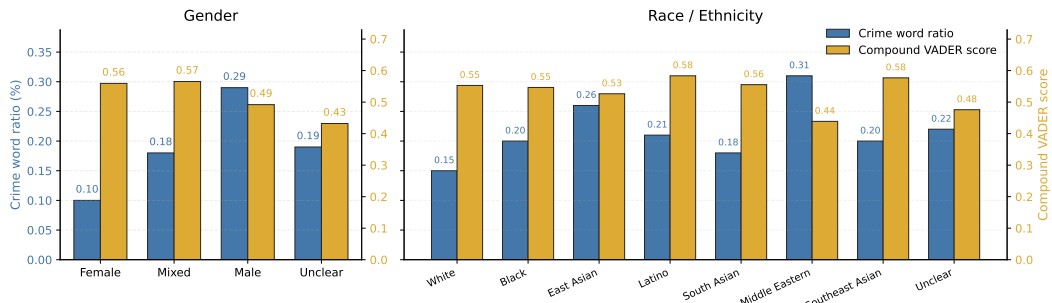

Figure 20: Analysis of sentiment and crime-related keywords in generated captions across gender (left) and race/ethnicity (right) groups. The left y-axis indicates the Crime word percentage (blue bars), which is the percentage of captions for each group containing at least one crime-related keyword. The right y-axis indicates the Compound VADER score (orange bars), which measures the average sentiment of the generated captions for each group. Prevalence of crime-related keywords is relatively low overall, ranging between 0.1% and 0.3%.

# G  ADDITIONAL INSIGHTS

## G.1  GEOGRAPHICAL BIAS IN LAION-400M

We analyze the geographical bias in LAION-400M by identifying all images whose alt-text captions contain a country name. For each country, we count the number of images in the dataset that contain its name or the corresponding adjective (e.g., "France" and "French"). We also consider common synonyms of country names, for example, "Holland" and "Netherlands". We lemmatize the country names and captions before matching them using the SPACY library. If a country name consists of multiple words, we treat both captions and country names as bags of words, meaning we count captions that include every word from the country name at least once. After collecting all images whose alt-text caption contains a country name, we calculate the number of these images where we detected at least one person labeled as male or female.

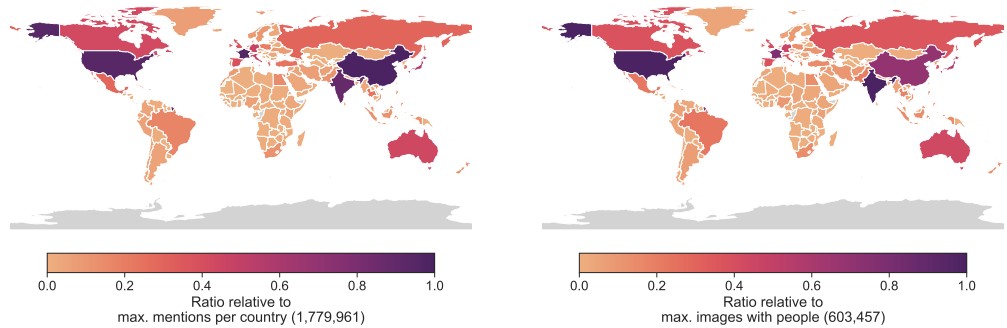

Figure 21: Geographical distribution of country mentions in the LAION-400M dataset, identified by matching country names within image alt-texts. Left: The total number of images associated with each country. Counts are normalized relative to the country with the most mentions (China: 1,779,961). Right: The number of images from the left set that also contain at least one detected person. Counts are normalized relative to the country with the most images containing a person (USA: 603,457).

The results are in Fig. 21. As expected from LAION-400M's focus on English captions, countries with large English-speaking populations are overrepresented. In contrast, countries in Africa, South America, Southeast Asia, and other regions of the Global South are underrepresented relative to their population size. Besides the English-speaking world, industrialized countries in Europe and East Asia also have a high number of associated images. These observations confirm a strong geographical bias in LAION-400M toward industrialized and English-speaking nations.

| Country | Total Images | | Images with Person | |
|---|---|---|---|---|
| | Num. Images | Rank | Num. Images | Rank |
| United States of America | 1,674,653 | 3 | 603,457 | 1 |
| India | 1,591,710 | 4 | 601,939 | 2 |
| France | 1,713,622 | 2 | 469,266 | 3 |
| China | 1,779,961 | 1 | 421,725 | 4 |
| Japan | 1,037,322 | 5 | 339,600 | 5 |
| Germany | 837,709 | 7 | 282,629 | 6 |
| Australia | 783,392 | 8 | 263,397 | 7 |
| United Kingdom | 662,397 | 11 | 244,353 | 8 |
| Canada | 761,375 | 9 | 229,914 | 9 |
| Spain | 688,016 | 10 | 228,842 | 10 |

Table 12: Top 10 countries ranked by the number of associated images in LAION-400M that contain a detected person. For each country, the table provides the absolute counts and overall rank for two categories: (1) the total number of images with a country mention in the alt-text, and (2) the subset of those images that include a person.

Table 12 shows the 10 countries with the most associated images that include at least one person. Although there are minor differences in the rankings, both sets of results confirm the same overall trend. Notably, the USA is the country with the most associated images showing a person, yet it is only the third most frequent country overall. Conversely, China has the most mentions overall, but only 23% of images with "China" in the alt-text caption also show a person.

These insights on geographical imbalance are important, as they show that some cultures are better represented than others. Consequently, models trained on such skewed data provide a poorer experience for a large part of the world's population (Senthilkumar et al., 2024; Bayramli et al., 2025; Dehdashtian et al., 2025; Nayak et al., 2025).

### G.2 SENTIMENT ANALYSIS USING TRAINED SENTIMENT CLASSIFIERS

In Section 4.2, we use the rule-based VADER sentiment analyzer (Hutto & Gilbert, 2014) to analyze the relationship between the sentiment of alt-text captions and the social groups visible in the corresponding image. Here, we complement this analysis with trained sentiment classifiers from the PYSENTIMIENTO library (Pérez et al., 2021). This library was previously used by Birhane et al. (2023) to discover that LAION-400M contains a significant amount of hate speech.

PYSENTIMIENTO provides a sentiment classifier trained on the SemEval 2017 Task 4A dataset (Rosenthal et al., 2017). Assuming a multiclass classification scenario, this classifier returns a probability distribution over 3 sentiment classes: positive, negative, and neutral sentiment. Additionally, PYSENTIMIENTO includes a model to detect hate speech, which is trained on data from SemEval 2019 Task 5 (Basile et al., 2019). This model assesses hatefulness, targetedness, and aggressiveness, particularly concerning women and immigrants. Hatefulness measures the presence of hate speech. Targetedness indicates whether the hate is directed at an individual or a group. Aggressiveness measures whether the content refers to violence, contains overt hostility, or legitimizes discriminatory attitudes. All three scores range from 0 to 1.

**Sentiment.** Figure Fig. 22 presents the sentiment classification results. The left panel shows the distributions of softmax probabilities $\geq 0.15$ for positive sentiment, calculated from the alt-text captions of female- and male-gendered images. The right panel displays the corresponding distributions for negative sentiment. These results confirm our observations in Section 4.2: positive sentiment is more prevalent in captions of female-gendered images, while negative sentiment is more common in those of male-gendered images. This finding supports the hypothesis that associations between the male gender and negative sentiment, and the female gender and positive sentiment, are due to correlations in the pretraining data.

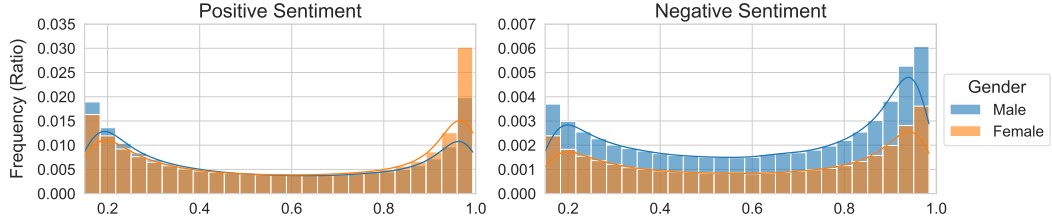

Figure 22: Distribution of softmax probabilities for positive sentiment (left) and negative sentiment (right), factorized by the perceived gender of the subjects in the corresponding images.

**Hate Speech.** To quantify the amount of hate speech in LAION-400M, Birhane et al. (2023) define the Hate Content Rate (HCR) as follows:

$$\text{HCR}_{\mathfrak{T}}(\tau) = 100 \times \frac{1}{N} \sum_{i=1}^{N} \mathbb{I}[P_{\mathfrak{T},i} > \tau], \tag{6}$$

where $N$ is the number of images, $\mathfrak{T}$ is the score type from {hateful, targeted, aggressive}, and $P_{\mathfrak{T},i}$ is the score for the image with index $i$. This metric measures the percentage of images with a hatefulness, targetedness, or aggressiveness score higher than a given threshold $\tau$.

We present HCR scores over a range of thresholds in Fig. 23, with separate results for male- and female-gendered images. All scores are consistently higher for female-gendered images, which aligns with the model's focus on identifying hate speech against women. Confirming observations from Birhane et al. (2023), hateful captions are much more prevalent than aggressive or targeted ones. We also observe that a large portion of high hatefulness scores identifies NSFW content. Our results, based on the full dataset rather than a subset as in Birhane et al. (2023), agree with their findings. Furthermore, our person labels allow us to confirm that these findings hold for images showing people. Hate speech is present for both genders, but is more prevalent for female-gendered images.

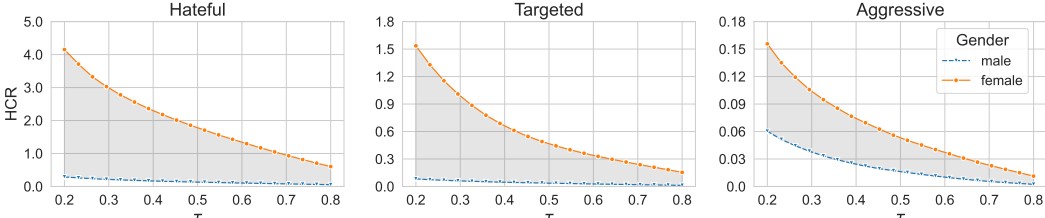

Figure 23: Hate Content Rate (HCR) as a function of the detection threshold $\tau$ for alt-text captions associated with images of male- and female-presenting people in LAION-400M. Each panel corresponds to a different score: hatefulness, targetedness, and aggressiveness. Note that the y-axis scales differ significantly across the three plots.

### G.3 POLYNOMIAL TRANSFER OF DATASET GENDER BIAS TO MODELS

In Section 5, we measure how much of the variance in model gender bias can be linearly predicted from direct co-occurrences of gender and social categories in LAION-400M. We find that, depending on the setup, 60% to 70% of the variance in model bias can be linearly predicted in this manner. A natural question is whether the fit can be improved by considering more expressive functional relationships beyond the linear regime. Generally, we find gains are modest. However, Chebyshev polynomial fits of degree 3 can improve $R^2$ by 1 to 3 points, while higher-order polynomials do not substantially increase the goodness of fit.

In Fig. 24a, we show the resulting best fit for CLIP gender bias, and in Fig. 24b, we show the resulting best fit for Stable Diffusion gender bias. The polynomial fit for CLIP gender bias improves the tail behavior, particularly where bias increases rapidly, i.e., very male-associated social categories in

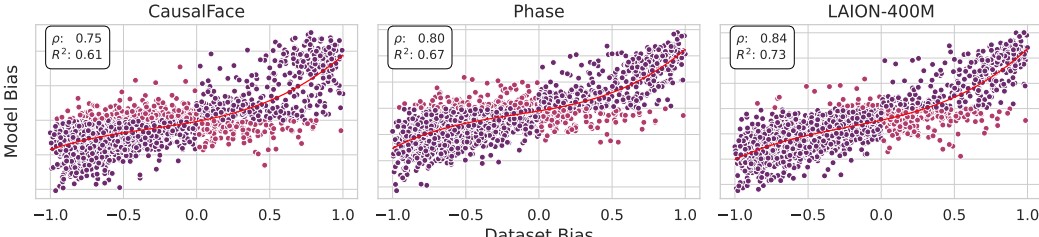

(a) Chebyshev polynomial fit (degree 3) of CLIP gender bias as a function of dataset gender co-occurrence.

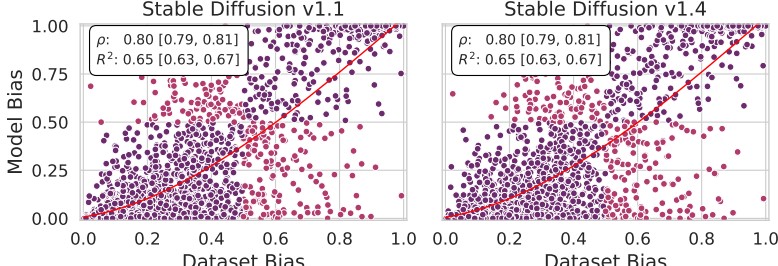

(b) Chebyshev polynomial fit (degree 3) of Stable Diffusion gender bias as a function of dataset gender co-occurrence.

Figure 24: Predicting model bias in CLIP and Stable Diffusion from dataset cooccurrences with nonlinear functions (polynomials).

the data result in a superlinearly stronger model bias than less strongly associated social categories. The fit also better represents the social categories that co-occur roughly equally with female- and male-gendered images, which tend to be more male-aligned in CLIP embeddings.

Regarding Stable Diffusion gender bias, we think that the relationship is governed by an S-shaped curve, where dataset co-occurrence ratios are squashed toward the extremes in generation ratios from Stable Diffusion. However, this is not fully reflected in the fit, likely because many social categories that are female-dominated in the data still yield mainly generations of male-gendered images. Outlier removal could help identify trends more clearly.

In summary, we find that trends clearly exhibit nonlinear patterns, but this is not easily addressed by assuming simple nonlinear relationships, such as polynomials or sigmoid functions. This leads us to conclude that better modeling of dataset biases, such as through second-order co-occurrences, is the more promising direction for better understanding and modeling gender bias transfer from data to models.

## G.4 DISTRIBUTION OF DEMOGRAPHIC GROUPS IN CC12M

We utilize YOLOv11 and our gender and race/ethnicity classifiers to examine the composition of the CC12M dataset (Changpinyo et al., 2021) with respect to demographic groups. CC12M is a web-scraped image dataset containing 12 million image-text pairs. As in Section 3, we detect person bounding boxes using YOLOv11 and label those with a minimum side length of 30 pixels using our fine-tuned classifiers. In total, we find 12,367,045 bounding boxes in 4,159,826 unique images. Interestingly, this means the average number of people per image (with at least one person) is higher in CC12M ($\approx$ 3) than in LAION-400M ($\approx$ 2).

The resulting statistics are in Fig. 25. We observe that, except for minor differences, the composition is similar to that of LAION-400M (see Fig. 5). Differences include a higher number of "mixed" images in CC12M, more people labeled "White", and fewer non-"White" categories. Overall, the high similarity demonstrates that our findings on LAION-400M generalize to similarly sourced web-scale datasets.

Figure 25: Distribution of perceived gender (left) and perceived race/ethnicity labels (right) in 12,367,045 bounding boxes and 4,159,826 unique images in the CC12M dataset.

### G.5 DATASET AND MODEL GENDER BIAS IN CRIME WORDS AND PERSONALITY TRAITS

We compare the dataset gender bias and the CLIP gender bias for crime words and personality traits using the same methods described in Section 5. The crime words come from the keywords listed in Table 2. The personality traits are taken from (Rudman et al., 2012, Study 1). To measure dataset bias, we evaluate a total of 59 crime words and 42 personality traits, all of which appear at least 100 times in the alt-text captions of LAION-400M.

Model gender bias is measured as the standardized difference in cosine similarities between CLIP word embeddings and male-gendered image embeddings. Dataset bias is measured as the difference between two ratios: the ratio of captions associated with female-gendered images that contain the word, and the ratio of captions associated with male-gendered images that contain the word.

The results appear in Fig. 26. We find that crime keywords are almost all male-biased in both the dataset and the CLIP model ("felony" is the sole exception). This confirms the analysis in Section 4, indicating that CLIP learns to associate crime words with male-appearing individuals based on the corresponding dataset bias. A similar pattern emerges for personality traits, with most traits being associated with males in both the data and models. A caveat is that linear fits do not perform well in these particular regimes due to the small sample size. Specifically, we observe a low but still positive correlation between dataset bias and model bias. This suggests that the analysis presented in Section 5 holds on a global level for social roles, although patterns may differ for particular subgroups.

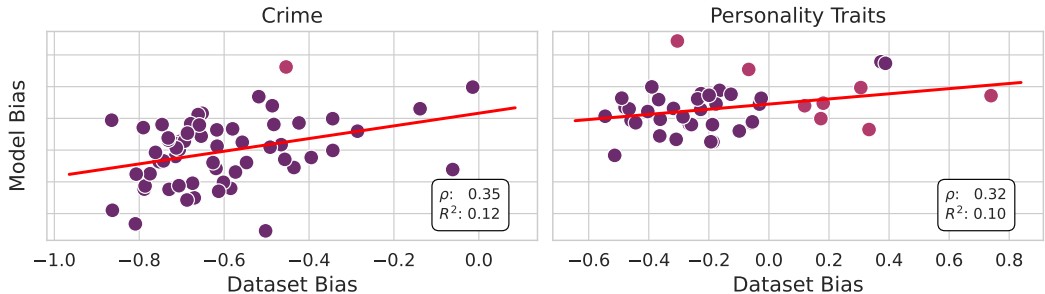

Figure 26: Relationship between dataset gender bias and CLIP gender bias for crime words and personality traits. Each point is a keyword appearing at least 100 times in LAION-400M alt-text. Model bias is the standardized difference in cosine similarity between CLIP word embeddings and male-gendered image embeddings and dataset bias is the difference in word frequencies between female- and male-associated captions. Lines show linear fits for each group.

## H EXTENDED DISCUSSION OF RELATED WORK

### H.1 ANALYSIS OF LAION-400M

Quantitative auditing and analysis of large-scale multimodal datasets such as LAION-400M is an underexplored area of research. Understanding the pretraining distribution of multimodal foundation models is highly relevant, especially for the social implications of large-scale AI applications. Con-

sequently, previous work has scrutinized certain aspects of LAION-400M and LAION-2B. However, these analyses generally target only a subset or the text modality of LAION-400M.

**Birhane et al. (2023)** investigate the prevalence of hateful content in LAION-400M and LAION-2B. Using a pretrained hatefulness scoring model, they detect racism and misogyny in the alt text of a subsample of image-text pairs (3.2M for LAION-400M and 12M for LAION-2B). They calculate the ratio of samples where the hatefulness score exceeds a given threshold. Their results suggest that hatefulness is more prevalent in LAION-2B compared to LAION-400M, implying that dataset scaling can have detrimental effects on fairness.

Our study provides richer insights into LAION-400M by providing annotations for the entire dataset, not just a subset, and by focusing on the visual domain in addition to the text.

**Al Sahili et al. (2025)** systematically investigate social bias in CLIP-style models by disentangling the effects of model size, training data scale, and data composition. They compare CLIP models, trained on 400M proprietary image-text pairs, with OpenCLIP models, trained on LAION-400M and LAION-2B, across various bias benchmarks. When comparing models of equivalent size trained on 400 million image pairs, the OpenCLIP model using LAION-400M shows greater gender bias, whereas the CLIP model is more racially biased. For OpenCLIP models, expanding the dataset from LAION-400M to LAION-2B doubles the gender skew and significantly increases race skew, contrary to the belief that more data leads to fairer outcomes. Al Sahili et al. conclude that corpus composition is a more dominant driver of social bias than the scale of the model or its training data alone.

Our additional labels can enhance analyses such as the one conducted by Al Sahili et al. by providing detailed information about the pretraining dataset. This allows for claims about the effects of particular datasets to be extended by tracing the underlying dataset statistics that cause models to learn their biases.

**Birhane et al. (2024)** investigate how training on LAION-400M and LAION-2B impacts racial bias in 14 CLIP-style models, using the Chicago Face Database (Ma et al., 2015) as a probe for evaluation. The results show that while larger datasets reduce the misclassification of people as animals, the effect on labeling individuals as "criminal" depends on the model's architecture. For some large models, scaling the dataset increases the probability of labeling Black and Latino men as "criminal" by 65% and 69% respectively. Conversely, for some smaller models, the same dataset scaling decreased the probability of these harmful classifications. Birhane et al. conclude that simply scaling up datasets can dangerously amplify historical and systemic biases. They underscore the urgent need for transparent curation practices and open access for independent audits to ensure model safety and fairness. A similar argument is made in (Birhane & Prabhu, 2021), where the authors show various ethical problems in ImageNet (Deng et al., 2009). Together, these observations call for comprehensive auditing of popular, openly available datasets used for pretraining foundation models.

Our study answers this call by providing extensive annotations and analysis of the entire LAION-400M dataset. By openly releasing our annotations, we allow researchers and the community to explore the content and biases of LAION-400M in unprecedented detail.

**Seshadri et al. (2024)** retrieve captions from LAION-2B that match specific occupation-related prompts. They automatically label the perceived gender in the retrieved images using a CLIP variant to obtain gender distributions for their prompts. Similarly, Friedrich et al. (2023) identify 1.83M images in LAION-5B representing occupations. They use a gender classifier trained on FairFace (Karkkainen & Joo, 2021) to label perceived binary gender in images and use this data to identify gender and intersectional imbalances. However, both studies label only a small part of LAION-2B or LAION-5B, specifically the images for the set of prompts considered. Furthermore, they assign a single holistic gender label to an entire image, whereas we provide bounding boxes and individual gender labels for multiple people in an image. Finally, FairFace consists mainly of images showing a person's face, which incurs a significant domain shift compared to LAION-400M.

**Udandarao et al. (2024)** calculate the frequency of concepts within large-scale pretraining datasets, including LAION-400M, to analyze the zero-shot capabilities of CLIP. To analyze the LAION-400M dataset, they first compile 4,029 concepts from various downstream evaluation tasks and then estimate each concept's frequency in images and alt-text captions. For the text modality, they create

an inverted index of all captions to count concept mentions, while for the image modality, they use the RAM++ open-set tagging model (Huang et al., 2023) to identify the visual presence of each concept. By intersecting the text and image search results, they obtain a matched image-text frequency for every concept. However, Udandarao et al. also find many cases where images and text are not well aligned.

While Udandarao et al. also label the LAION-400M dataset, their focus differs from ours. They provide image-level attributes that describe visible objects and their features. In contrast, we offer fine-grained, person-specific annotations, including bounding boxes and captions for all individuals, and their gender and race. Our approach allows for a more detailed analysis of person representations, whereas their method supports a broader but less precise analysis of diverse concepts.

**Dulhanty & Wong (2019)** introduce a framework to automatically annotate perceived age and gender in two ImageNet subsets by classifying face crops: the 2012 Large Scale Visual Recognition Challenge (ILSVRC) training set and the "person" hierarchical category. The findings for the ILSVRC subset reveal a significant demographic imbalance. Faces appearing as female comprise 41.62% of the data, while individuals over 60 account for only 1.71%. The largest identified subgroup is males aged 15 to 29, representing 27.11% of the faces. This imbalance is more pronounced in the 'person' subset, where only 31.11% of individuals appear female.

**Alabdulmohsin et al. (2024)** calculate gender and profession labels for one billion image-text pairs in the WebLI dataset (Chen et al., 2023b) using an unspecified image classifier. However, they do not release this data or provide enough detail to reproduce their annotations. Furthermore, their labels are holistic and do not account for multiple people within an image.

**Zheng et al. (2022)**, finally, use a face detection model to detect 50 million faces in LAION-400M. We go beyond this by providing bounding boxes for entire people, not only faces. We also provide additional labels for perceived gender and person descriptions. Lastly, we use our automatically generated labels to generate insights into the relationship between bias in pretraining datasets and the models trained on them, whereas Zheng et al. primarily use their dataset for training a general face encoding model.

## H.2 SOCIAL BIAS IN CLIP MODELS

**Background on CLIP.** CLIP, or Contrastive Language-Image Pre-training, (Radford et al., 2021) is a multimodal model that learns to understand images and text together through contrastive pre-training. At its core, CLIP has two separate encoders: one for images, typically a Vision Transformer (Dosovitskiy et al., 2021), and one for text (Vaswani et al., 2017). These encoders map their inputs into a shared, high-dimensional embedding space. During training on a large dataset of (image, text) pairs, the model processes a batch of size $N$. The image encoder produces image embeddings $v_i$, and the text encoder creates text embeddings $u_j$. The training objective is to maximize the similarity for the $N$ correct $(v_i, u_i)$ pairs while minimizing it for the incorrect pairings.

This goal is formalized with a symmetric cross-entropy loss calculated on the cosine similarity scores between image and text embeddings. For a given batch, the loss for predicting the correct text for each image is:

$$L_{\text{image-to-text}} = -\frac{1}{N} \sum_{i=1}^{N} \log \frac{\exp((v_i \cdot u_i)/\tau)}{\sum_{j=1}^{N} \exp((v_i \cdot u_j)/\tau)} \tag{7}$$

Simultaneously, the loss for predicting the correct image for each text caption is:

$$L_{\text{text-to-image}} = -\frac{1}{N} \sum_{i=1}^{N} \log \frac{\exp((u_i \cdot v_i)/\tau)}{\sum_{j=1}^{N} \exp((u_i \cdot v_j)/\tau)} \tag{8}$$

In these equations, $\tau$ is a learnable temperature parameter that scales the logits before the softmax operation, controlling the sharpness of the probability distribution. The total loss for CLIP is the average of these two components:

$$L_{\text{CLIP}} = (L_{\text{image-to-text}} + L_{\text{text-to-image}})/2. \tag{9}$$

By optimizing this objective, the model learns to pull related image and text representations closer together in the embedding space while pushing unrelated ones apart. This process builds a rich,

semantically organized representation space that enables CLIP's powerful zero-shot learning capabilities.

However, this contrastive objective also means that CLIP can learn to associate images with biased or harmful information frequently contained in the paired text. For example, if many images of women are paired with text containing derogatory language, CLIP will learn to associate images of women with negative sentiment. Similarly, if images of men are frequently paired with text referring to crime or criminals, CLIP will, to some extent, learn to associate images of men with these categories. Here, we discuss in detail recent works that have analyzed and found social bias in CLIP models.

**Summary.** Previous work collectively demonstrates that CLIP models internalize and reproduce a wide spectrum of human-like societal biases, spanning gender, race, age, occupation, religion, and other social categories. A predominant methodological trend is the quantitative measurement of these biases using the cosine similarity between the model's image and text embeddings. Many studies adapt variants of the Word Embedding Association Test (WEAT) (Caliskan et al., 2017) to calculate association scores and effect sizes that reveal stereotypical linkages (Steed & Caliskan, 2021; Wolfe et al., 2022; Janghorbani & De Melo, 2023). Another common approach involves analyzing biases in image retrieval tasks, where metrics like Bias@K or MaxSkew quantify the demographic skew in top-ranked results for neutral queries (Wang et al., 2021; Berg et al., 2022; Hall et al., 2023). These investigations frequently utilize established datasets like FairFace (Karkkainen & Joo, 2021) and the Chicago Face Database (Ma et al., 2015), as well as custom-built or synthetic datasets, to systematically probe for specific stereotypes (Wolfe & Caliskan, 2022b; Hausladen et al., 2024). A near-universal conclusion is that CLIP learns these biases from stereotypical representations present in its vast, web-scraped training data, thereby reflecting and perpetuating societal inequities (Mandal et al., 2023; Hamidieh et al., 2024). Researchers also find that larger dataset scales can cause models to learn more nuanced and subtle human biases (Wolfe et al., 2024). Furthermore, these intrinsic biases are shown to propagate into downstream applications, such as text-to-image generation and visual question answering, where they can amplify harmful stereotypes (Wolfe & Caliskan, 2022a). Finally, several papers note that the manifestation of these biases is highly sensitive to experimental design, including the choice of text prompts and class labels, highlighting the critical role of careful application design in mitigating harm (Agarwal et al., 2021; Hausladen et al., 2024).

**(Wang et al., 2021)** analyzes gender bias in CLIP by evaluating its performance on image search tasks using gender-neutral text queries. They conduct experiments on MS-COCO (Lin et al., 2014) and Flickr30K (Plummer et al., 2015), where they create a purely gender-neutral test corpus by programmatically removing or replacing gender-specific words in the image captions. To determine the gender attribute of an image, they rely on its associated human-annotated captions, labeling an image as "male" or "female" based on the presence of masculine or feminine words. The primary metric used to quantify bias is `Bias@K`, which measures the normalized difference between the number of masculine and feminine images in the top-K retrieved results for a given gender-neutral query. Their findings reveal that the CLIP model exhibits significant gender bias: for example, on the MS-COCO 1K test set, the results for gender-neutral queries skewed towards men, with about 6.4 out of 10 retrieved images depicting males. Additionally, the paper evaluates CLIP on an occupation dataset from (Kay et al., 2015), measuring a "similarity bias" defined as the difference in the expected cosine similarity between an occupation term (e.g., "doctor") and images of men versus women in that occupation. This analysis also shows that the CLIP model has a severe similarity discrepancy for various occupations like "telemarketer" and "chemist".

**(Steed & Caliskan, 2021)** proposes the Image Embedding Association Test (iEAT) to quantify social biases in image embedding models. They evaluate biases such as racial and gender stereotypes by measuring the differential association of two sets of target concept images, $X$ and $Y$ (e.g., 'male' vs. 'female'), with two sets of attribute images, $A$ and $B$ (e.g., 'career' vs. 'family'). The test statistic is defined as

$$s(X, Y, A, B) = \sum_{x \in X} s(x, A, B) - \sum_{y \in Y} s(y, A, B), \qquad (10)$$

where

$$s(w, A, B) = \text{mean}_{a \in A} \cos(w, a) - \text{mean}_{b \in B} \cos(w, b) \qquad (11)$$

measures the association of a single image embedding $w$ with the attribute sets based on cosine similarity. The study uses stimuli from psychological tests and web searches and demonstrates that

models pre-trained on ImageNet learn significant human-like biases, including racial, gender, and intersectional stereotypes. Steed & Caliskan conclude that these biases are learned automatically from stereotypical portrayals of people on the web. The iEAT framework can also be applied to CLIP by using CLIP's image encoder to generate the embeddings for the stimulus sets. These embeddings would then be fed into the iEAT equations to measure the magnitude and statistical significance of biases.

**(Dehouche, 2021)** investigates social stereotypes in CLIP by generating a dataset of 10,000 synthetic portrait photographs and performing zero-shot classification of these images. This process yields class-belonging probabilities, which are the model's computed likelihoods that an image belongs to a given textual label (e.g., "Attractive" or "Unattractive"), derived from the cosine similarity between CLIP's image and text embeddings. The analysis primarily uses Pearson correlation on these probabilities to measure associations between protected attributes (gender, ethnicity) and other labels. The study finds significant correlations reflecting cultural stereotypes, such as strong positive associations between Female and Attractive, Male and Rich, and White Person and Attractive. Dehouche concludes that because CLIP reflects biases present in culture, fairness is application-dependent, proposing that specific, undesirable stereotypes should be neutralized geometrically at the point of deployment.

**(Agarwal et al., 2021)** evaluates bias in CLIP through experiments that target denigration harms and gendered associations. To measure denigration, they prompt CLIP in a zero-shot setting to classify 10,000 images from the FairFace dataset (Karkkainen & Joo, 2021) while adding non-human and crime-related labels to the class set. Analysis of misclassification rates showed that Black individuals are most frequently classified into non-human categories, while males are more often associated with crime-related labels. A second experiment investigates gender by classification of images of U.S. Members of Congress with two distinct label sets, one based on occupations and another set containing a mix of occupations and attributes. By analyzing label distributions at varying probability thresholds, Agarwal et al. found that lower-probability labels introduced gender stereotypes like 'nanny' for women and 'mobster' for men. CLIP also associates appearance-related descriptions more with women, while linking high-status occupations like 'executive' and 'doctor' more often to men. However, the study concludes that the manifestation of such biases is highly sensitive to class design, as adding the label 'child' significantly reduced harmful classifications for younger individuals, implying that design choices are a key determinant of how model biases manifest.

**(Berg et al., 2022)** proposes ranking metrics to measure bias in CLIP, primarily using MaxSkew and Normalized Discounted Cumulative KL-Divergence (NDKL). MaxSkew originates from Skew@k, which is defined as

$$Skew_{A@k}(\tau_T) = \ln \frac{p_{\tau_T,T,A}}{p_{d,T,A}}. \tag{12}$$

This formula calculates the log-ratio between the actual proportion of an attribute A, $p_{\tau_T,T,A}$, and its desired proportion, $p_{d,T,A}$, within the top-k ranked results. For example, if a search for "engineer" should ideally show 50% women ($p_{d,T,A}$) but the top 100 results only include 10% women ($p_{\tau_T,T,A}$), the Skew would be $\ln(0.10/0.50)$. This negative value indicates a significant under-representation. Accordingly, MaxSkew@$k$ is defined as

$$MaxSkew_{A@k}(\tau_T) = \max_{A_i \in \mathcal{A}} Skew_{A@k}(\tau_T), \tag{13}$$

measuring the maximum skew of any attribute. $MaxSkew_{A@k}$ continues to be a popular metric in evaluating retrieval bias and has been adopted by several papers (Chuang et al., 2023; Smith et al., 2023; Seth et al., 2023; Hirota et al., 2025; Al Sahili et al., 2025; Zhang et al., 2025a). Also proposed by Berg et al., the NDKL metric is defined as

$$NDKL(\tau_T) = \frac{1}{Z} \sum_{i=1}^{|\tau_T|} \frac{d_{KL}(D_{\tau_{T_i}} || D_T)}{\log_2(i+1)}. \tag{14}$$

This metric aggregates the Kullback-Leibler (KL) divergence between the observed attribute distribution, $D_{\tau_{T_i}}$, and the desired distribution, $D_T$, across all ranked positions. A logarithmic discount is applied, meaning that bias in higher-ranked results carries more weight. For instance, if the first ten images of a search all belong to one demographic group, this contributes more to the total bias score than if images 91 through 100 showed the same skew. This prioritizes penalizing bias that

users encounter first. Berg et al. use these metrics to quantify skewed associations between sensitive text queries, like "a photo of a criminal", and the demographic attributes of faces in the FairFace and UTKFace datasets. In addition to these ranking metrics, they assess bias by measuring the rate of harmful zero-shot misclassifications. This occurs when images of people are incorrectly categorized into non-human or criminal classes. The analysis of CLIP revealed significant harmful associations. For example, the model misclassified images of Black individuals into crime-related categories at a disproportionately high rate compared to other ethnic groups.

**(Wolfe et al., 2022)** investigates whether CLIP exhibits hypodescent, the social principle of assigning multiracial individuals to a minority racial group. The analysis employs a face-morphing experiment using images from the Chicago Face Database (Ma et al., 2015), creating 21-step series of synthetic faces that transition between self-identified Asian, Black, or Latino/a individuals and White individuals. Bias is primarily measured by calculating the cosine similarity between a morphed image's embedding and competing racial text labels (e.g., "a photo of a Black person" vs. "a photo of a White person"). Here, a higher similarity with the minority label at the 50% morph point signifies hypodescent. Results demonstrate that CLIP strongly adheres to this rule, classifying the majority of intermediate images with the minority-group label, particularly for female faces (e.g., 89.1% of 50/50 Asian-White female morphs are labeled "Asian"). Furthermore, the study reveals that "White" functions as a default race, evidenced by a high Pearson correlation ($\rho \leq 0.82$) between an image's similarity to the label "person" and its similarity to "White". Valence biases are quantified using an adapted Word Embedding Association Test (Caliskan et al., 2017), defined as

$$s(i, A, B) = \frac{\text{mean}_{a \in A} \cos(i, a) - \text{mean}_{b \in B} \cos(i, b)}{\text{stddev}_{x \in A \cup B} \cos(i, x)}, \tag{15}$$

where $i$ is the image embedding and $A$ and $B$ are sets of unpleasant and pleasant words. This metric shows that an image's association with unpleasantness correlates positively with its association with the "Black" label, demonstrating that CLIP encodes human-like racial stereotypes.

**(Wolfe & Caliskan, 2022b)** analyzes CLIP for biases related to social markedness across age, gender, and race or ethnicity, using images from the FairFace dataset (Karkkainen & Joo, 2021). The primary method involves measuring which individuals are "marked" with a demographic label versus being described by the default, unmarked term "a person". This is quantified by comparing the cosine similarity between an image embedding and text prompts: an image is considered unmarked if the prompt "a photo of a person" is ranked higher than prompts specifying a demographic (e.g., "a photo of a Female person"). A second metric, representational self-similarity, is used to analyze the structure of the embedding space, defined as the mean pairwise cosine similarity for a group of images $G$ with the CLIP embeddings of the $i$-th image denoted as $v_i$:

$$s(G) = \frac{1}{n^2 - n} \sum_i \sum_{j \neq i} \cos(v_j, v_i) \tag{16}$$

where a higher value indicates that the model's representations are more tightly clustered around a shared trait. The study finds that CLIP disproportionately leaves individuals who are White, male, and middle-aged unmarked, treating them as the default, while preferentially marking individuals from other races, females, and those at the extremes of age. These markedness patterns directly correlate with higher self-similarity, indicating that the model's representations for these groups are less varied and more defined by their demographic characteristics. The authors conclude that CLIP internalizes and reflects societal biases from its training data, where dominant groups are treated as the norm and others are marked as deviations, with these biases compounding at the intersection of identities.

**(Wolfe & Caliskan, 2022a)** investigates whether CLIP internalizes and reproduces the societal bias that equates American identity with being White. The analysis primarily utilizes images of self-identified Asian, Black, Latina/o, and White individuals from the Chicago Face Database (Ma et al., 2015). Bias is quantified using Embedding Association Tests (EATs), which measure the differential association between sets of images and attribute words by calculating the normalized difference between mean cosine similarities, yielding an effect size $d$. For instance, the EAT assesses whether image embeddings of White individuals are more closely associated with text embeddings of in-group words (e.g., "we", "our") compared to images of other racial groups. The findings reveal that, across all evaluated models, images of White individuals are more strongly associated with patriotism and in-group terminology, while images of Asian, Black, and Latina/o individuals are more

associated with egalitarianism and being an immigrant. These latent biases manifest in downstream tasks that use CLIP as image encoder: a visual question answering model identified 97% of White individuals as "American" but only 3% of Asian individuals, while a text-guided image generator prompted with "an American person" consistently lightened the skin tone of input images across all racial groups. The authors conclude that these multimodal models learn, reflect, and amplify significant human-like biases, which propagate to their real-world applications with potentially harmful consequences.

**(Wolfe et al., 2023)** presents a quantitative analysis of sexual objectification bias in CLIP models, replicating several experiments from the psychology literature. The study primarily measures bias using an Embedding Association Test (EAT), which calculates an effect size $d$ representing the differential association between two sets of target concepts (e.g. objectified vs. non-objectified images) and two sets of attributes (e.g. text with vs. without emotion). The metric is formally defined as

$$d = \frac{\text{mean}_{x \in X}\, s(x, A, B) - \text{mean}_{y \in Y}\, s(y, A, B)}{\text{stddev}_{w \in X \cup Y}\, s(w, A, B)}, \tag{17}$$

where $s(w, A, B)$ computes the difference in mean cosine similarity of an image embedding $w$ to the attribute sets $A$ and $B$. Using the standardized SOBEM database (Ruzzante et al., 2022) of "objectified" (partially clothed) and "non-objectified" (fully clothed) women, Wolfe et al. find that models significantly disassociate human characteristics like emotion from objectified women, a conclusion supported by saliency maps showing the model's attention shifting from the face to the chest. Further experiments demonstrate that images of female professionals are more strongly associated with sexual descriptions than their male counterparts and that text-to-image generators using CLIP text encoders produce sexualized images of teenage girls by default. The study concludes that CLIP models trained on web-scraped data learn, reflect, and can amplify harmful, human-like societal biases.

**(Janghorbani & De Melo, 2023)** introduces a framework to analyze stereotypical biases in CLIP beyond common gender and race analyses. They compile a multimodal benchmark, MMBias, which contains images and textual phrases for 14 population subgroups across religion, nationality, disability, and sexual orientation. To measure bias, they calculate an effect size $d$, which quantifies the differential association between two target groups (e.g., images of Islam, $X$, and Christianity, $Y$) and two attribute sets (e.g., pleasant words, $A$, and unpleasant words, $B$). This is defined as the normalized difference in mean association scores, where the score for a single item $w$ is

$$d(w, A, B) = \text{mean}_{a \in A} \cos(w, a) - \text{mean}_{b \in B} \cos(w, b), \tag{18}$$

using cosine similarity between embeddings. For instance, the analysis reveals that CLIP more strongly associates images representing Islam with negative-valence words like "terrorist" and "oppression" compared to images representing Christianity, which are associated with positive words like "peace" and "blessing". The study concludes that CLIP has internalized and can reproduce significant, harmful societal stereotypes pertaining to a wide range of demographic groups, reflecting biases present in its training data.

**(Hall et al., 2023)** introduces Visogender, a benchmark dataset designed to quantify occupational gender bias in vision-language models by leveraging a collection of images balanced by the perceived gender of individuals in professional roles. The analysis is conducted through two primary tasks: pronoun resolution and image retrieval. The resolution task assesses a model's ability to correctly associate gendered pronouns in captions with the corresponding subjects in an image, where resolution bias is measured by the resolution accuracy gap, defined as $\Delta(o) = RA_m(o) - RA_f(o)$, which is the difference between the resolution accuracy for masculine-presenting subjects ($RA_m$) and feminine-presenting subjects ($RA_f$) for a given occupation $o$. The retrieval task, conversely, measures bias by analyzing the gender representation in the top K images returned for a gender-neutral query (e.g., "the doctor and their patient"), using metrics like Bias@K to quantify the over-representation of one gender. For instance, a positive Bias@K value indicates that more images of masculine-presenting professionals were retrieved. Benchmarking several CLIP-style models, the study finds that these models struggle with pronoun resolution in complex scenes with multiple people and exhibit significant biases in both resolution and retrieval. The results indicate that while the direction of bias differs across models, models generally show a tendency to retrieve more masculine-presenting individuals, and their performance correlates with societal gender stereotypes found in U.S. Labor Force statistics.

**(Mandal et al., 2023)** create a diverse probing dataset by scraping 1,260 images from Google using the search terms "man" and "woman" translated into various languages across nine geographical regions to analyze social bias in CLIP. They let CLIP predict labels for these images from two lexicons: one for occupations and one for personality-describing adjectives. To measure bias, they primarily utilize the Word Embeddings Association Test (WEAT) (Caliskan et al., 2017), which quantifies the association between target words (the predicted labels) and attribute words (gendered terms). The core metric is the WEAT association score for a single word $w$ with attribute sets $A$ (male terms) and $B$ (female terms), calculated as

$$s(w, A, B) = \text{mean}_{a \in A} \cos(w, a) - \text{mean}_{b \in B} \cos(w, b), \tag{19}$$

where a positive score indicates a male bias. This is aggregated into a differential association score for the set of male-associated labels $X$ and female-associated labels $Y$, given by

$$s(X, Y, A, B) = \sum_{x \in X} s(x, A, B) - \sum_{y \in Y} s(y, A, B), \tag{20}$$

which is then normalized to produce the final WEAT effect size. The findings indicate that CLIP demonstrates significant stereotypical gender bias, associating adjectives like 'knowledgeable' with men and 'feminine' with women. Furthermore, the occupational biases discovered in CLIP have a strong correlation with real-world statistics: occupations more associated with men in the model corresponded to higher median salaries, while those associated with women aligned with lower pay and higher female workforce participation. The paper concludes that training models on unfiltered internet-scale data does not eliminate bias but instead mirrors and perpetuates societal inequities.

**(Wolfe et al., 2024)** investigates the emergence of human-like facial impression biases in 43 CLIP vision-language models by comparing their outputs to human judgments from the One Million Impressions (OMI) dataset (Peterson et al., 2022). The analysis measures the model-human similarity for 34 attributes, defined as the Spearman's rank correlation, $sma = \rho(m^a, h^a)$, between the vector of average human ratings ($h^a$) and a corresponding vector of model associations ($m^a$) for all 1,004 OMI images. A model's association for a single image is calculated as the difference in cosine similarity between the image embedding and text embeddings for opposing poles of an attribute (e.g., "trustworthy" vs. "devious"). Wolfe et al. find that CLIP models learn facial impression biases, including for visually unobservable traits like trustworthiness, and demonstrate that the degree to which a model reflects a bias is strongly predicted by the inter-rater reliability (IRR) of that bias among humans. Furthermore, the study reveals that dataset scale is a critical factor: models trained on larger datasets (LAION-2B) learn more subtle, subjective biases and develop an internal structure of correlated attributes that more closely resembles that of human social perception. These findings extend to generative models that use CLIP as text encoder, showing that Stable Diffusion (Rombach et al., 2022) inherits these facial impression biases, which can intersect with and amplify demographic stereotypes. The paper concludes that increasing data scale causes vision-language models to more faithfully reproduce nuanced and potentially harmful societal biases embedded in their training data.

**(Hamidieh et al., 2024)** introduces So-B-IT, a taxonomy of 374 words across nine categories of potential social bias in CLIP, such as occupation, criminal justice, and appearance. The core methodology measures the associations between these terms and demographic groups by using text prompts (e.g., "a photo of a [word]") to retrieve the top 100 most similar images from the demographically balanced FairFace dataset. Bias is then quantified by analyzing the demographic distribution of the retrieved images and by calculating a Caption-Association Score,

$$CASC(c, G) = \frac{\text{mean}_{g \in G} \cos(c, g) - \text{mean}_{g' \in \overline{G}} \cos(c, g')}{\text{stddev}_{x \in G \cup \overline{G}} \cos(c, x)} \tag{21}$$

This metric measures the normalized difference in mean cosine similarity $\cos(c, \cdot)$ between a caption $c$ and images of a target group $G$ versus all other images. In this context, the set $G$ represents the specific demographic group under investigation (e.g., 'Middle Eastern men'), while its complement, $\overline{G}$, includes all other images in the dataset that are not part of group $G$. For instance, a high CASC score for the caption "a photo of a terrorist" and the demographic group "Middle Eastern men" indicates the model strongly associates that term with that group. Note that the CASC score is equivalent to the Word Embedding Association Test adaptation proposed by (Wolfe et al., 2022) except for using a single group of social categories instead of two attribute groups. The study concludes that CLIP

models exhibit significant and intersectional biases, such as associating "homemaker" specifically with Indian women and "CEO" with white men. The authors trace these learned stereotypes back to skewed demographic representations within the model's pre-training data.

**(Hausladen et al., 2024)** investigates social perception biases in CLIP by measuring associations between synthetically generated face images and psychologically-grounded text prompts. The analysis utilizes the synthetic dataset introduced in (Liang et al., 2023), where images are systematically and independently varied across protected attributes (race, gender, age) and non-protected, confounding attributes (facial expression, lighting, pose). Bias is quantified by measuring the cosine similarity between CLIP's image and text embeddings, introducing a normalized metric defined as

$$\Delta \cos(I, D) = \text{mean}_{i \in I, d \in D, t \in T} \left( \cos(i, \mathbf{t}(d, t)) - \cos(i, \mathbf{t}(\emptyset, t)) \right),$$

(22)

which isolates a social trait's effect by subtracting the similarity of the image to a neutral prompt. For example, this measures how much "friendlier" a face is perceived, independent of how much it looks like a generic "person". Hausladen et al. conclude that CLIP makes fine-grained, human-like social judgments that are systematically biased by protected attributes, finding a particularly strong and unique pattern of extreme perceptions for Black women across different ages and expressions. Critically, the study demonstrates that non-protected attributes like facial expression can influence social perception more than protected attributes like age, suggesting that analyses on wild-collected data without controlling for such confounds may yield misleading conclusions.

