# OpenReview forum: "Person-Centric Annotations of LAION-400M: Auditing Bias and Its Transfer to Models"
_ICLR.cc/2026/Conference — ICLR 2026 Poster_

### Official Review · Reviewer_DJbC · 2025-10-26

**Soundness:** 4
**Presentation:** 3
**Contribution:** 3
**Rating:** 6
**Confidence:** 4

**Summary:**

The paper annotates the images of LAION 400M by providing bounding boxes of person detections, inferred gender and race labels, as well as detailed captions for each detected person. The annotations reveal significant issues: males and people with Middle Eastern or Black appearances are more strongly correlated crime-related content or negative sentiment, a lot of which is shown to propagate into the models trained on the studied dataset. The authors also study the themes of captions concerning the racial and gender combinations, and uncover interesting patterns.

**Strengths:**

1. The paper provides gender and racial annotations for images in LAION-400M, which is a significant contribution given that LAION400M is web-scraped.
2. Instead of using off-the-shelf gender and race classifiers, authors particularly finetune models (e.g., SIGLIP for gender) to make them more aware of the domain, and to handle female, male, mixed and unclear cases (and the equivalent for race).
3. Analysis reveals more association of males and Middle-Eastern/Black races with crime and negative sentiments
4. Authors also attempt to tie dataset bias with model bias (CLIP and Stable Diffusion), and find a significant overlap.

**Weaknesses:**

1. The paper examines LAION400M, which was used to train earlier versions of OpenCLIP and Stable Diffusion, hardly used now. The biases are expected to grow with scale, as emphasized by previous papers [a], and a similar experiment on LAION 2B, DataComp, etc would have helped us analyse biases in more modern models.
2. The identity-topic associations are dependent on the automatically generated captions from pretrained caption generators. Such models may carry their own biases, and it is hard to say if the generated captions are accurate. Similarly, topic analysis on the original captions may have revealed more patterns.
3. The authors study biases in downstream models via social categories. Crime-related and Sentiment-based analysis would have been valuable too. It often does not guarantee that presence of social category c in captions would ensure that the persons present in the corresponding images actually belong to category c, due to misalignment issues [b].
4. The authors mention that 60-70% of the biases in downstream models can be explained by those in the datasets. However, they do not discuss what leads to the rest of the biases, especially in Stable Diffusion.

[a] Birhane et al., 'Into the LAION’s Den: Investigating Hate in Multimodal Datasets', NeurIPS D&B 2023

[b] Udandarao et al., 'No “Zero-Shot” Without Exponential Data: Pretraining Concept Frequency Determines Multimodal Model Performance', NeurIPS 2024

**Questions:**

1. What if the gender and race classifiers were not finetuned, and instead the ensemble VLMs were used to annotate the entire dataset? Do the authors avoid it for computational costs? What are their thoughts on analysis on some other dataset like CC-12M or DataComp - is a separate finetuning required for those cases too?
2. How do the authors verify that YOLOv11-l does not have gender/racial biases?

---

> ### Author Response · Authors · 2025-11-20
> **Author Response to Reviewer DJbC (1/2)**
>
> We thank the reviewer for the positive evaluation of our work, highlighting its “significant contribution” and methodological strengths (fine-tuned classifiers, dataset bias to model transfer analysis, …).
>
> Below, we discuss the individual concerns raised by the reviewer.
>
> ---
>
> > Similar experiment on LAION 2B, DataComp, etc would have helped us analyse biases in more modern models. Do the authors avoid it for computational costs? What are their thoughts on analysis on some other dataset like CC-12M or DataComp - is a separate finetuning required for those cases too?
>
> Fortunately, a separate fine-tuning of our classifiers is not necessary. To annotate datasets in the same way we did for LAION-400M only requires bounding box detection and gender and race/ethnicity classification using our classifiers. We restricted ourselves to LAION-400M for computational reasons, but we would like to stress that our efforts yield a dataset that is orders of magnitude larger than any existing dataset, e.g. both FACET and Phase both contain less than 100,000 images.
>
> In our revision (Appendix G.4), we include an analysis of the detected gender and race/ethnicity labels on CC12M. We find that while the average number of detected people in images with at least one person is higher in CC12M ($\approx$ 2.97) than in LAION-400M ($\approx$ 1.86), the distribution of predicted gender and race/ethnicity labels is very similar. This confirms our insights generalize to similarly sourced web-scraped datasets.
>
> ---
>
> > The identity-topic associations are dependent on the automatically generated captions from pretrained caption generators. Such models may carry their own biases, and it is hard to say if the generated captions are accurate.
>
> Thank you for raising this concern. We agree that automatically generated captions may introduce their own biases. However, modern MLLMs are generally strong captioning models, and recaptioning is now a standard practice for improving dataset quality (e.g., [1, 2, 3, 4]). Our topic analysis further focuses on broad, easily identifiable themes (e.g., religion, sports), which reduces sensitivity to fine-grained captioning errors. Moreover, as the reviewer notes, the original alt-text captions contain well-documented issues, so analyzing recaptioned data provides a more reliable signal. Additionally, in our revision (Sec. 3, “Generating Person-Centric Captions”), we have now added a quantitative experiment that justifies our choice of captioning model. Please see our General Response for details.
>
> To more directly evaluate whether our captioning model introduces harmful bias, we added a quantitative experiment in the revision (Appendix F, “Caption Bias Analysis”) measuring (1) the ratio of captions containing crime-related keywords and (2) average VADER sentiment across gender and race/ethnicity groups. Results are in Fig. 20. Crime-related keywords are extremely rare in synthetic captions (0.1%–0.3%), though ratios are higher for male vs. female and for Middle Eastern vs. White individuals. Crucially, these disparities are far smaller than in the original alt-text captions, where ratios are 0.17% (female) vs. 0.83% (male) and 0.38% (White) vs. 1.71% (Middle Eastern), indicating that synthetic captions reduce associations of identity and crime rather than amplify them.
>
> Sentiment scores show similar trends, but overall sentiment is much higher than in alt-text captions because synthetic captions are longer and MLLMs are alignment-tuned. While disentangling dataset- from model-induced bias remains difficult (e.g., war-related image content affects sentiment even of “objective” captions), the very low crime prevalence, reduced disparities relative to alt-text, and consistently positive sentiment suggest that model effects are limited and that the generated captions are safe and appropriate for analysis.
>
>
> [1] Chen et al.: _Sharegpt4v: Improving large multi-modal models with better captions_. In ECCV, 2024.\
> [2] Fan et al.: _Improving clip training with language rewrites_. In NeurIPS, 2023.\
> [3] Nguyen et al.: _Improving multimodal datasets with image captioning_. In NeurIPS 2023.\
> [4] Li et al.: _What if we recaption billions of web images with llama-3?_ In arXiv 2024.\
> [5] Udandarao et al.: _'No “Zero-Shot” Without Exponential Data: Pretraining Concept Frequency Determines Multimodal Model Performance'_. In NeurIPS 2024.

---

> > ### Author Response · Authors · 2025-11-20
> > **Author Response to Reviewer DJbC (2/2)**
> >
> > > The authors study biases in downstream models via social categories. Crime-related and Sentiment-based analysis would have been valuable too. It often does not guarantee that presence of social category c in captions would ensure that the persons present in the corresponding images actually belong to category c, due to misalignment issues.
> >
> > We agree with the reviewer that presence of a social category in the alt-text caption does not mean that the category is actually represented by a person in the image. In fact, the existence of such potentially harmful biases is exactly what our analysis aims to measure. The contrasting objective of CLIP models causes the model to learn these relations. If certain social categories (in text captions) frequently cooccur with visual cues associated with some genders or races/ethnicities, the model will learn this, and accordingly the CLIP embeddings for images containing such cues will be more similar to text embeddings of descriptions mentioning the social category. Through our data, it becomes possible to measure on a large scale, and we provide an analysis regarding to which extent direct cooccurrences linearly relates to model bias.
> >
> > In the revision, we have added an explicit evaluation of crime keywords and personality traits in Appendix G.5. There, we confirm that both CLIP and LAION-400M associate crime keywords more with male-gendered images, and this holds also for most personality traits.
> >
> > ---
> >
> > > The authors mention that 60-70% of the biases in downstream models can be explained by those in the datasets. However, they do not discuss what leads to the rest of the biases, especially in Stable Diffusion.
> >
> > Thank you for this suggestion. In Sec. 5, we discuss several directions how this analysis can be expanded, and we added one new experiment in Appendix G.3. Please see our [general response](https://openreview.net/forum?id=t3ZMiHhqXm&noteId=1qAhljvH85) for details.
> >
> > ---
> >
> > > How do the authors verify that YOLOv11-l does not have gender/racial biases?
> >
> > To evaluate this, we check if the person detection performance of Yolo-v11 is the same across different demographic groups in the FACET and Phase datasets, i.e., we evaluate subgroup fairness. Details on this experiment are in Appendix C. We invite the reviewer to check the details stated there, which may answer the question. In short, we find only very small differences between demographic groups and no systematic trends that raise concerns about biased detections by Yolo-v11.
> >
> > ---
> >
> > We would like to thank the reviewer again for the great suggestions to improve our paper. We are looking forward to a constructive discussion.

---

> > > ### Comment · Reviewer_DJbC · 2025-11-28
> > >
> > > Thanks to the authors for clarifying most of my doubts.
> > >
> > > *'The identity-topic associations are dependent on the automatically generated captions from pretrained caption generators. Such models may carry their own biases, and it is hard to say if the generated captions are accurate.'* - Here, by biases i did not mean over-use of crime-related words against some communities. I meant issues like the captioning model accurately recognizing all religious attires/sports/etc from across the world (e.g., baseball vs cricket, attire of police officer vs security guard). Since this section performs topic modelling, and different topics are attached to the different genders and races, it is crucial that the captions are correct, especially in case of races, as races could be associated with multiple cultures across the world. Rather, the original captions would have been a more accurate descriptor of the activities performed by the persons in the images.

---

> ### Author Response · Authors · 2025-12-03
>
> We thank the reviewer for acknowledging that our rebuttal has addressed all their concerns, except the suggested topic analysis on the original alt-text captions using our novel SAE-based method.
>
> To assess feasibility, we calculated the pointwise mutual information between individual lemmas in the alt-text captions and intersectional identities. However, we found that most meaningful associations are with words that appear as few as 20 times across all captions, which necessarily introduces noise and also comments on the sparsity and uninformativeness of alt-text captions. Below, we show the resulting table of top 10 most associated keywords for each identity. We find some expected ones, such as “ mujahedin” for Middle Eastern identities, but most of them are noise.
>
> Based on these results, we strongly believe that synthetic captions are strictly more meaningful than the often uninformative alt-text captions, and the costs of performing this experiment (embedding 400 million captions, training SAEs, SAE inference on all captions, expected GPU-h: 200) is not justified by the expected insights.
>
> We nevertheless thank the reviewer for the continued engagement and clarifying the original comment. We hope our explanation sufficiently explains why we didn’t include this particular experiment in the revision.
>
> ---
>
> |           | Female                                                                                                               | Male                                                                                                           |
> |:---------------|:---------------------------------------------------------------------------------------------------------------------|:---------------------------------------------------------------------------------------------------------------|
> | Black          | nicety, fulbe, doornail, kabala, shithead, rickrack, prideful, immunise, passionflower, seychellois                  | counterpunch, striving, aflatoxin, chunga, eudaimonia, treasonable, aloes, patronizing, modifiable, dogmatism  |
> | East Asian      | slattern, topee, horseshoer, faithlessness, prudish, dyspnea, tidiness, rhombic, mollycoddle, shortish               | yanan, tekki, tange, trapezium, troglodyte, mongoloid, sightedness, centralise, faithlessness, cortical        |
> | Latino         | tribade, huascaran, hollering, greengrocery, babassu, barbecuing, michigander, cochineal, mesoamerican | packhorse, boilerplate, salvadorean, oxcart, plutocrat, pachuco, broncho, torreon, contras, vocalize           |
> | Middle Eastern  | fellah, asala, mujahedin, pentangle, druse, mamilla, wimple, theosophical, hizbollah, shamash                        | sukkoth, cheops, lethality, overstretch, thwarted, ashur, peachwood, advisement, mujahedin, levite             |
> | South Asian     | mynah, incandescence, glassed, unmatchable, serail, undamaged, stocktaking, fineness, underpay             | unburied, indic, regularisation, lilting, vaccinium, victimisation, aave, prosthodontics, dhal, overconfidence |
> | Southeast Asian | aiai, altimeter, secularisation, chloroquine, padda, hummock, fealty, nepenthes, elongation, greengrocery            | lengthwise, scag, madrasah, shiah, longyi, elephas, volcanology, tornillo, aleatory, prizefight                |
> | White          | nonconforming, gritstone, diploid, lipizzan, jabiru, enol, aboveground, habitude, mucous, afterbirth                 | enol, patellar, faltering, pathogenesis, terabyte, meerschaum, jakobson, croaker, benzene, flatfoot            |

---

### Official Review · Reviewer_gDXh · 2025-10-29

**Soundness:** 2
**Presentation:** 3
**Contribution:** 3
**Rating:** 4
**Confidence:** 4

**Summary:**

The paper presents a large-scale study that adds person-centric annotations to the LAION-400M dataset. It automatically labels each detected person with perceived gender and race or ethnicity using a combination of object detection models and MLLMs. The paper then analyzes demographic distributions, harmful associations such as links to crime or negative sentiment, and the relationship between dataset bias and model bias in CLIP and Stable Diffusion. The main finding is that about 60–70% of model bias can be explained by co-occurrence patterns in the data.

**Strengths:**

- The paper tackles an important and timely topic.

- While prior work has provided protected-attribute annotations for smaller datasets such as COCO or GCC, this paper scales the effort to LAION-400M, a much larger and more representative dataset, aligning with recent trends in large-scale data research.

- The analysis is comprehensive and provides multiple insights. For example, the paper uncovers detailed demographic distributions, harmful associations, and correlations between data composition and model outputs.

- The study linking dataset bias and model bias is particularly interesting. The finding that a significant portion of model bias can be explained by dataset co-occurrences highlights the need for the community to address dataset-level bias more seriously.

**Weaknesses:**

- There is a potential risk that biases from the MLLMs used for demographic attribute annotation and caption generation propagate into the resulting annotations. For instance, if these models make more errors for certain genders or races, their biases may directly influence the final dataset. Although the paper validates agreement with human-labeled datasets, three concerns remain:

1. The agreement is relatively low for race annotations, and it is questionable to dismiss this easily. If certain racial groups have higher error rates, the final demographic distribution could diverge significantly from reality.

2. Relatedly, the paper only reports aggregate error rates but does not analyze error trends. It would be important to know whether errors are uniformly distributed or concentrated on specific groups, as this strongly affects the reliability of MLLM-based annotations.

3. No human study was conducted to verify the quality of the obtained annotations. While comparison with datasets such as FACET provides a proxy for human validation, a small-scale human study (even around 1K samples) on LAION-derived annotations would greatly strengthen confidence in their accuracy.

- In the dataset–model bias correlation experiment, the paper mentions that “the remaining 30–40% of bias stems from nonlinear or higher-order effects,” but does not provide any quantitative or qualitative analysis of these effects. Since this bias-transfer analysis is one of the central contributions, including at least some empirical investigation or hypothesis testing for these unexplained components would significantly reinforce the paper’s impact.

**Questions:**

**Overall assessment and suggestions**

This paper presents an interesting and valuable attempt to provide protected-attribute annotations and large-scale demographic analysis for a dataset of the scale of LAION-400M. The topic is timely and important, and the effort to enable systematic auditing of web-scale data is commendable.

However, I believe the paper does not sufficiently analyze (or mitigate) the potential biases introduced by the automatic annotation pipeline itself, especially those arising from the MLLMs used for labeling gender and race. Given that this work focuses on human-centric annotations, such limitations are critical and cannot be easily overlooked.

If the rebuttal provides a convincing analysis or additional validation addressing this issue, I would be happy to raise my score.

---

> ### Author Response · Authors · 2025-11-20
> **Author Response to Reviewer gDXh (1/2)**
>
> We thank the reviewer for the constructive feedback, highlighting the “important and timely topic”, the alignment with “recent trends in large-scale data research”, and the “comprehensive analysis”. We are pleased the reviewer sees our study on gender bias transfer as “particularly interesting”.
>
> The reviewer provided valuable suggestions that are shared by other reviewers, so we respond to most of them in our [general response](https://openreview.net/forum?id=t3ZMiHhqXm&noteId=1qAhljvH85). In particular, the reviewer’s main concern involves the validation of automatically inferred gender and race/ethnicity labels. In our revision, we include an evaluation involving human annotators in Sec. 3 (“Human Validation Study”) to demonstrate the validity of our findings, see the [general response](https://openreview.net/forum?id=t3ZMiHhqXm&noteId=1qAhljvH85) for details on this.
>
> ---
>
> > The agreement is relatively low for race annotations, and it is questionable to dismiss this easily. If certain racial groups have higher error rates, the final demographic distribution could diverge significantly from reality.
>
> We absolutely agree with the reviewer on this point, and this constitutes the main motivation for our experimental setup. First, we would like to note that literature (discussed in Appendix A) agrees that perceived race/ethnicity is a subjective category, with limited objective grounding in “reality”. Different (human) observers frequently disagree on (perceived) race/ethnicity. This means any concrete label assignment will conflict with opinions of a significant number of human observers, but they will also disagree among themselves. There is no way to eliminate this fundamental epistemic uncertainty. To respond to this problem, our work uses and enables the following mitigation strategies:
> 1. In contrast to prior work, we use MLLM consensus labels to train classifiers. This ensures these labels are not only based on the perspective of a single MLLM.
> 2. A simple post-processing strategy is to collapse labels with frequent confusions, such as East Asian and Southeast Asian. We intentionally chose a fine-grained label set to enable both levels of analysis.
> 3. Our classifier returns a distribution over labels (different from MLLM, and we will make these logits available to researchers in addition to the maximum-likelihood labels). As discussed in Sec. 3 (lines 243 to 246), it is possible to threshold on logits, but we did not find advantages for our analysis, while other researchers may. Additionally, we analyzed the pairwise correlation of logits of the race/ethnicity classifier and found correlation patterns that partially recover common disagreements of human annotators, for example logits for Southeast Asian and East Asian are correlated. These results are in Sec. E.2 (Fig. 19) in the revision.
>
> We believe that these measures, together with our quantitative and qualitative evaluation, yield high-quality labels that will be useful for future research.
>
> ---
>
> > Relatedly, the paper only reports aggregate error rates but does not analyze error trends. It would be important to know whether errors are uniformly distributed or concentrated on specific groups, as this strongly affects the reliability of MLLM-based annotations. No human study was conducted to verify the quality of the obtained annotations.
>
> In the revision, we have added a human validation study in Sec. 3 and show that disagreement between human annotators and our race/ethnicity labels mirrors disagreement between human annotators. Please see our [general response](https://openreview.net/forum?id=t3ZMiHhqXm&noteId=1qAhljvH85) for details. Specifically, we find the following patterns regarding race/ethnicity:
> * Both human-human and human-classifier disagreement is (relatively) higher for White, Latino, and Middle Eastern groups, and for East Asian and Southeast Asian.
> * Human-human disagreement (but not human-classifier disagreement) is also higher on Black and Latino, and on Middle Eastern and South Asian.
> * These patterns are unsurprising and can be explained by cultural or geographic similarities, as well as common skin tone associations.
> * Finally, our classifier assigns “unclear” more frequently than humans. However, we think these more conservative judgements are more desirable than overgeneralization
>
> For gender, we find very high agreement (Cohen’s $\kappa = 0.95$ for human-classifier agreement). We notice almost perfect agreement on “male” and “female” labels, but humans are in some cases more conservative than classifiers in assigning “unclear” and “mixed”. Since “male” and “female” are overall more frequent, this is less serious.

---

> > ### Author Response · Authors · 2025-11-20
> > **Author Response to Reviewer gDXh (2/2)**
> >
> > > Since this bias-transfer analysis is one of the central contributions, including at least some empirical investigation or hypothesis testing for these unexplained components would significantly reinforce the paper’s impact.
> >
> > Thank you very much for this valuable suggestion. In Sec. 5 (“Discussion & Future Work”), we now discuss several directions on how to better model bias transfer, and we included a new experiment in Appendix G.3 using nonlinear instead of linear fit. Please see the [general response](https://openreview.net/forum?id=t3ZMiHhqXm&noteId=1qAhljvH85) for more details on this. Specifically, we identify the following directions:
> > * Indirect co-occurrences influence bias, i.e. bias transfers between co-occurring words where one also frequently cooccurs with a demographic in the visual domain. To evaluate this, we need to model this second-order bias, for example, by label propagation on the cooccurrence graph. We haven’t developed a full theory for this yet, but it is an exciting topic for future research.
> > * Optimizer or data sampling may affect the extent to which models acquire dataset bias. We aren’t aware of research on this, but it’s well known that SGD-trained models generalize better than Adam-trained models. Similar effects may hold for bias. To evaluate this, we need to train CLIP models from scratch, which is not within our compute budget.
> > * The model (together with its training, e.g., the loss) induces nonlinear dynamics. Fig. 8 and Fig. 9 clearly show nonlinear relationships of co-occurrence rates and model bias. We found Chebyshev polynomials improve the $R^2$ fit by 1 to 3 points (see App. G.3). However, we think that modeling dataset bias beyond direct co-occurrences is the most interesting way to gain further insights into dataset-to-model bias transfer.
> >
> > ---
> >
> > We thank the reviewer once again for the great suggestions, and we hope our answers adequately address the reviewer’s concerns. We are looking forward to a constructive discussion.

---

> > > ### Comment · Reviewer_gDXh · 2025-11-25
> > >
> > > Thank you very much for the detailed response. The rebuttal addresses most of my main concerns, although I still think there remains some risk of bias being introduced through the automated demographic attribute annotation process, which the authors also acknowledge. That said, I believe the paper has taken reasonable steps within its scope, and the overall contribution is still clear and meaningful. I have therefore increased my score.

---

> > > > ### Author Response · Authors · 2025-11-25
> > > >
> > > > We thank the reviewer for the positive assessment of our work and the support. We are happy that we address the concerns and that the reviewer acknowledges that the paper "has taken reasonable steps within its scope" to mitigate the "risk of bias being introduced through the automated demographic attribute annotation process".
> > > >
> > > > Thank you very much again for the great review and suggestions, which have helped improve our work.
> > > >
> > > > Sincerely,\
> > > > the authors

---

### Official Review · Reviewer_uTpU · 2025-11-01

**Soundness:** 2
**Presentation:** 2
**Contribution:** 3
**Rating:** 6
**Confidence:** 3

**Summary:**

This work aims to examine demographic patterns in a large web dataset and how they reflect in downstream generative systems. The paper augments LAION-400M image-caption dataset by including person-level annotations, encompassing bounding boxes around individuals, with automatically inferred gender and ethnicity labels, and detailed captions for each detected person.

With this annotated dataset, the paper examines how different gender and ethnic groups are represented and how these identities intersect with themes like crime, sentiment, and broader contextual associations. They also explore how gender-related biases in the dataset correlate to biases in two models trained on LAION-400M, CLIP and Stable Diffusion.

**Strengths:**

* The study proposes an interesting methodology to examine demographic biases in pretraining datasets and their effects on generative models.
* The scope of this work is impressive, both in its scale and in its comprehensive examination of demographic patterns and biases within the dataset.
* The paper presents important findings on how sentiments and topics are associated with gender and ethnic identities, and to what extent gender bias in model generations correlates with biases in the training dataset.
* The additional annotations might facilitate future research, such as studying other forms of dataset-model interaction

**Weaknesses:**

* While the paper ambitiously combines large-scale demographic annotation with multiple layers of analysis, its broad scope makes the presentation overly dense. As a result, some key details and justifications are underdeveloped or omitted from the main paper, limiting clarity and depth in certain areas.
* To generate gender and ethnicity labels, the paper fine-tunes a SigLIP classifier using a subset of the LAION-400M data labelled by three different MLLMs. The labelling process relies on consensus among these models, with training (and testing) data primarily drawn from images where all agreed. While this approach enhances label reliability, it may bias the classifier toward clear-cut examples and limit its robustness to the ambiguous or noisy cases.

Minor typos:
Line 52 mentions the word “intersectionalidentity”
Line 291, the caption says “compound score (orange)”, color is not exactly orange

**Questions:**

Some suggestions:

* The paper uses YOLOv11-l for bounding box generation, relying on evaluations from datasets such as FACET and PHASE. Since LAION is a much noisier web-based dataset with the possibility of multiple people per image, assessing the detector’s accuracy on a subset of LAION would strengthen the work and ensure reliability in this context.

* The paper mentions a qualitative analysis of MLLMs over 4,939 bounding boxes to select one model for generating person-specific captions; however, the description of this process is somewhat vague. Providing more detail on this selection procedure and, if feasible, extending similar qualitative analysis to the gender and ethnicity detection pipelines would strengthen the reliability of the overall methodology.

I am willing to reconsider my assessment based on authors' response to the above.

* (Minor): it would be good to hear authors' take on where they think the annotations could be used in the future

**Details Of Ethics Concerns:**

The paper includes example images of clearly identifiable people from the LAION-400M dataset. Since LAION contains web-scraped content without explicit consent, displaying such images may violate privacy of individuals.

---

> ### Author Response · Authors · 2025-11-20
> **Author Response to Reviewer uTpU (1/2)**
>
> We thank the reviewer for the detailed feedback highlighting the “impressive scope” of our work, our “interesting methodology”, and the “important findings”. We are pleased that the reviewer believes our contributions “might facilitate further research”.
>
> Below, we address the remaining concerns mentioned in the review.
>
> ---
>
> > The paper includes example images of clearly identifiable people from the LAION-400M dataset. Since LAION contains web-scraped content without explicit consent, displaying such images may violate privacy of individuals.
>
> We agree with the reviewer that this is an issue. We have therefore taken action to remove this problem from our paper:
>
> In our revision, we have replaced all images from LAION-400M that show persons. In Fig. 1, we have used AI generated images, based on captions generated from the originals. In Fig. 3, we have blurred faces to protect the identity of shown individuals, as here showing examples from LAION-400M alongside our annotations is important. Likewise, in Tab. 11, Fig. 11, and Fig. 14 we blurred faces.
>
> Please inform us if this addresses the concern.
>
> ---
>
> > The labelling process relies on consensus among these models, with training (and testing) data primarily drawn from images where all agreed. While this approach enhances label reliability, it may bias the classifier toward clear-cut examples and limit its robustness to the ambiguous or noisy cases.
>
> This is a very good observation. When developing our classifiers, we also experimented with including a “disagreement” class to capture cases where MLLMs disagree, but we found this greatly decreases the quality of predicted labels. The most likely reason is that models struggle to handle conflicting cues, as MLLMs might also mislabel images that possess clear cues associated with certain groups. Instead, we think the following directions can increase robustness and potentially leverage disagreeing MLLM predictions:
> * Labels can be created from a majority vote (instead of strict consensus) and weighted during training. In this case, it is beneficial to add more MLLMs beyond the three we use.
> * Additionally, when applying trained classifiers, we can use uncertainty quantification and OOD-detection, possibly on internal activations, to filter predictions. We briefly discuss this in Sec. 3 (line 243 to 246). Although we find no advantages for our analysis, it may be beneficial in other scenarios.
>
> Both are interesting directions for future work on more robust demographic labeling.
>
> ---
>
> > Providing more detail on this selection procedure and, if feasible, extending similar qualitative analysis to the gender and ethnicity detection pipelines would strengthen the reliability of the overall methodology.
>
> Thank you very much for the suggestion. In the revision (Sec .3, “Generating Person-Centric Captions”), we have additionally included a quantitative justification why we chose InternVL3-8B to generate captions for all 200 million bounding boxes. Please see our [general response](https://openreview.net/forum?id=t3ZMiHhqXm&noteId=1qAhljvH85) for details on this experiment.
>
> ---
>
> > assessing the detector’s accuracy on a subset of LAION would strengthen the work and ensure reliability in this context
>
> We asked one human annotator to label 200 images for correctness of the bounding boxes. The annotator decides whether a person is missing (i.e., that did not receive a bounding box), whether a non-human object is assigned a bounding box, or whether the image is correctly annotated, i.e. there are no missing bounding boxes or bounding boxes around non-human objects. We find that 82.5% of images are correct, 10% have a bounding box for a non-human object, and 7.5% of images miss at least one bounding box. Since images can contain multiple bounding boxes, the effective number of missed bounding boxes is actually lower. We have updated Sec. 3 (“Detecting Person Bounding Boxes”) accordingly. Upon inspection, we also found that non-human objects were labeled as “unclear” by our classifiers, and missing individuals in images are small in most cases and appear in the background.
>
> ---
>
> > some key details and justifications are underdeveloped or omitted from the main paper, limiting clarity and depth in certain areas.
>
> We acknowledge that we report many technical details of our methods in the appendix instead of the main paper, but we do provide extensive documentation of our design choices and their justification in the supplementary material. This approach allows the main paper to focus on the key insights and methods. We would be grateful if the reviewer could provide specific pointers to which justifications are underdeveloped or where clarity can be improved. We are happy to revise these and add details that may have been omitted.

---

> > ### Author Response · Authors · 2025-11-20
> > **Author Response to Reviewer uTpU (2/2)**
> >
> > > (Minor): it would be good to hear authors' take on where they think the annotations could be used in the future
> >
> > Thank you for asking this question. We envision several use cases, including:
> > 1. Our annotations can inform studies about how rebalancing the dataset affects model bias, i.e., when we change the data distribution and retrain the model. This analysis is included, for example, in [1]. However, their dataset is unpublished, and the paper does not mention sufficient details to reproduce it. Our dataset, instead, is available to researchers, contributing to further work in this area.
> > 2. Our annotations enable studying how dataset bias transfers to models, i.e., which measurements of dataset bias best fit observed model bias. We have initial experiments on this in Sec. 5, but we are excited to see further work on developing a more mature theory of these relations.
> > 3. Our annotations can serve as a standard for comparison when building new datasets. Comparing against our annotations allows researchers to assess whether the new dataset is more balanced than LAION-400M and which categories are represented differently.
> > 4. The captions we created constitute a significant computational effort, and they contain rich information about people profiles in the dataset. We believe they can be of independent interest, for example, in analyzing social roles in web-scale data or for generative modeling.
> >
> > [1] Alabdulmohsin et al.: Clip the bias: how useful is balancing data in multimodal learning? In ICLR, 2024
> >
> > ---
> >
> > > Minor typos: Line 52 mentions the word “intersectionalidentity” Line 291, the caption says “compound score (orange)”, color is not exactly orange
> >
> > Thank you for pointing out these typos. We have fixed them in the revision. Regarding the color, we have decided on “amber”.
> >
> > ---
> >
> > We would like to thank the reviewer again for the detailed feedback and suggestions. We are looking forward to a constructive discussion.

---

### Official Review · Reviewer_sfn7 · 2025-11-03

**Soundness:** 3
**Presentation:** 3
**Contribution:** 3
**Rating:** 6
**Confidence:** 5

**Summary:**

The paper generated person-centric annotations over LAION-400M, resulting in 270M odd detected person bounding boxes, 200M perceived gender and race/ethnicity labels (after filtering), and person-centric captions generated by MLLMs. The labels are used to quantify the level of demographic imbalances, thematic associations (via sparse autoencoders), and quantify how much of observed gender bias in CLIP and Stable Diffusion can be linearly explained by dataset co-occurrences.

The paper quantifies the extent to which bias in downstream models can be attributed to biases in the dataset. Futhermore, the dataset can be useful for future studies to understand dataset-model interactions in propagating or amplifying biases in large-scale datasets. The reliability of the findings can be improved by auditing potential issues of bias in the proposed process (MLLM ensemble labeling -> classifier training -> dataset labeling -> bias estimation).

**Strengths:**

- While the prevalence of bias in large models, and its attribution to bias in training datasets is well known, there are no large-scale annotated datasets with demographic labels. So the proposed dataset could be a valuable resource for more fine-grained studies seeking to understand and mitigate bias in models trained on large models.

- The workflow (YOLOv11 person detection -> MLLM ensemble labeling -> SigLIP finetuning -> full-dataset labeling -> analyses) for automated labeling is quite reasonable, well described and should be reproducible in principle.

- The attempt to quantitatively relate dataset co-occurrence statistics to measured model biases (CLIP, Stable Diffusion) has been lacking and this paper fills the gap.

**Weaknesses:**

There are several weaknesses in the proposed methodology, which reduces the reliability of the quantitative findings.

- Labeling relies on an MLLM ensemble consensus. However, these MLLMs may have inherent biases that would now propagate through the rest of the method. There is no analysis on the errors and biases of the MLLM ensemble. Similarly, bias in the pre-trained and fine-tuned SigLIP has not been analyzed.

- All the presented results in the paper are point estimates. How reliable are these estimates? Confidence intervals are missing. Similarly, how sensitive are the correlation estimates to hyperparameter choices in the full pipeline?

- Statements like “60–70% of gender bias in CLIP and Stable Diffusion can be linearly explained by direct co-occurrences in the data” are stronger than warranted. The result shows a strong *correlation* but not necessarily *causation*. These claims need to either be substantiated or rephrased.

**Questions:**

- In cases where the MLLM ensemble agree or disagreed, how do they relate to specific demographic groups?

- Bias analysis of the gender and race classifiers.

- To understand the reliability and robustness of the claims, confidence intervals and sensitivity to hyperparameters.

**Details Of Ethics Concerns:**

The paper has proactively addressed potential ethics concerns, especially with respect to automated labeling of perceived gender/race and how it might not reflect reality. I think the authors have given sufficient thought to these concerns and addressed them adequately.

---

> ### Author Response · Authors · 2025-11-20
> **Author Response to Reviewer sfn7**
>
> We thank the reviewer for highlighting our contributions, noting that we provide a “valuable resource” whose construction is “reasonable” and “reproducible”, and that the work “fills the gap” in research for “quantitatively relating dataset co-occurrence statistics to measured model biases”. In our revision, we address the remaining concerns mentioned by the reviewer and discuss them below.
>
> ---
>
> > All the presented results in the paper are point estimates. How reliable are these estimates? Confidence intervals are missing. Similarly, how sensitive are the correlation estimates to hyperparameter choices in the full pipeline?
>
> We have now provided confidence intervals where applicable. Fig. 5 reports gender and race/ethnicity label counts, which do not have a probabilistic interpretation. Fig. 6 already reports the distribution over crime keyword ratios, including the medians, standard deviations, and 95% CIs. Fig. 7 reports average VADER scores, which is also a single scalar score. For Fig. 8 and 9, we now obtained bootstrap estimates of $R^2$ and $\rho$ distributions to calculate the 95% confidence interval. Figures are updated in the revision. Confidence intervals are tight and support our conclusions.
>
> We think it is prohibitively expensive to conduct an extensive hyperparameter study for the full pipeline, because each step (MLLM ensemble prediction, Classifier training, Full dataset labeling) involves significant computational cost. MLLM ensemble prediction requires labeling 3 x 3 million images for gender and 3 x 7 million images for race/ethnicity. Full dataset labeling requires predicting labels for 2 x 200 million bounding boxes. However, we validate each step individually, so labels are accurate. If the reviewer lets us know which specific parameter or setting we should validate, we will be happy to provide experimental results if it is computationally feasible.
>
> ---
>
> > In cases where the MLLM ensemble agree or disagreed, how do they relate to specific demographic groups?
>
> We provided a disagreement analysis for race/ethnicity labels (Appendix E.1, Fig. 17). Here, images are naturally separated into groups by the race/ethnicity keyword used to retrieve them. This analysis shows that “black” is the category with the highest agreement (33.50%), while “Latino” is the category with lowest agreement (2.68%). This justifies why multi-model agreement is necessary to obtain reliable data, i.e., images with features commonly associated with a specific race/ethnicity, in contrast to a single annotator.
>
> For gender labels, this analysis is not available because there is no reference label to compare against. In Appendix D, we include a table (Tab. 6) with pairwise agreement rates between models. In the revision, we added the precise number of instances where all models agree: Models agree on “male” for 703,756 out of 1M bounding boxes, on “female” for 739,915 out of 1M bounding boxes, on “unclear” for 735,713 out of 1M bounding boxes, and on “mixed” for 18,997 bounding boxes among the previous categories. Overall, we see that agreement rates are similar for the “male”, “female”, and “unclear” categories, while there are few bounding boxes where all MLLMs agree on “mixed”. This is, on the one hand, due to its overall lower frequency (generally, bounding boxes contain only one person). On the other hand, if a bounding box contains multiple people, at least one of them is likely to be partially occluded or not fully contained in the bounding box, which creates ambiguity.
>
> We hope this clarifies the question. Please let us know if we can provide further details.
>
> ---
>
> > Bias analysis of the gender and race classifiers.
>
> Thank you for this suggestion. In our revision (Sec. 3), we include a validation study and find that disagreements between our classifier mirror disagreements between human annotators. Please see our general response for details.
>
> > The result shows a strong correlation but not necessarily causation. These claims need to either be substantiated or rephrased.
>
> Thank you for pointing out this issue with our terminology. To clarify, whenever we use the term “explain” in the context of measuring gender bias transfer from data to models, we are referring to the “fraction of variance explained” (FVE; [1]), a common metric to measure regression fit. We acknowledge that this wording might be misleading. We have revised the relevant mentions and clarified them in our revised version. Specifically, in the abstract and introduction, we replaced “is explained” with “a linear fit predicts”. Likewise, in Sec. 5 (last paragraph in Results), we replaced “explains” with “predicts”. Please let us know if you have further suggestions for making this more transparent.
>
> [1] https://en.wikipedia.org/wiki/Fraction_of_variance_unexplained
>
> ---
>
> We would like to thank the reviewer again for the great comments and suggestions. We are looking forward to a constructive discussion.

---

### Author Response · Authors · 2025-11-20
**General Response by Authors**

We thank the reviewers for their time and positive evaluation of our work, with ratings 6, 6, 6, and 4. This confirms the interest of our work to the community and its contributions. In particular, reviewers highlighted the scale and comprehensiveness of our work (uTpU, gDXh), its significant contributions and important findings (DJbC, gDXh, uTpU), and the value of our annotations for future research (sfn7). All four reviewers expressed that our analysis of bias transfer from data to models is interesting and fills an important gap in research.

The main request for clarification mentioned by reviewers (sfn7, uTpU, gDXh) concerns the reliability of automatically inferred labels, and the lack of human validation. Also, reviewers uTpU and DJbC asked for quantitative justification of choosing InternVL3-8B for captioning, and reviewers DJbC and gDXh are interested in further perspectives on bias transfer. In our revision, we have addressed these as follows:

---

## Human Validation
We conducted a human validation study on 648 images, stratified by gender and race/ethnicity labels from our classifier. Two volunteers labeled all images for gender and race/ethnicity. Before the study, we obtained ethics approval from an external provider, and we filtered harmful images containing nudity or other NSFW content. Our main findings are:
* Agreement between models and human annotators on gender is very high, i.e. Cohen’s $\kappa=0.95$  and we find that 98% of male and female labels assigned by human annotators match the labels assigned by the classifier. We have added the corresponding confusion matrix in Fig. 4. We observe a limited number of disagreements primarily on “unclear” and “mixed” labels. Human annotators are more conservative when assigning these labels, but due to the overall lower number (see Fig. 5), this is of less concern.
* Regarding race/ethnicity, we find moderate-to-high agreement (Cohen’s $\kappa\approx 0.65$) between humans and the model and critically also between humans. This confirms perceived race/ethnicity is subjective to some extent, as discussed in Appendix A. Please note that disagreement between human annotators constitutes an upper bound on human-classifier agreement, so we believe our classifier yields reasonable labels.
* We show that disagreement patterns among humans match the disagreement between the model and humans (e.g. East Asian vs. Southeast Asian). A notable exception is the “unclear” category, where the model assigns “unclear” more frequently than humans, which we think is preferable to the opposite.
We believe this additional study adds to the validity of our work, and we hope it satisfies the reviewers. We integrated our findings extensively into our revision (Sec. 3, “Human Validation Study”).
---
## Quantitative Justification of Captioning Model
We selected InternVL3-8B for creating person captions based on qualitative assessment. Now, we also provide a quantitative argument by using GPT-5.1 as a judge to assess model win rates on captions. Given captions of two models, we ask GPT-5.1 to select the better one across 500 images. Results show that GPT-5.1 clearly prefers InternVL3-8B over other models by a wide margin (win rate = 0.756 against Qwen2.5-VL-3B, 0.630 against Qwen2.5-VL-7B and 0.582 against InternVL3-2B). We conclude that InternVL3-8B is a reasonable choice, and we updated Sec. 3 (“Generating Person-Centric Captions”) and App. F with this new experiment.

---

## Bias Transfer beyond Linear Prediction from Co-occurrence Rates
In the paper, we analyze data-to-model bias transfer via linearly predicting model bias from co-occurrence counts of social categories and demographics in the data. In the revision, we added a discussion (Sec. 5, “Discussion & Future Work”) on sources of model bias not covered by direct co-occurrences:
* Indirect co-occurrences influence bias, i.e. bias transfers between co-occurring words where one also frequently cooccurs with a demographic in the visual domain. To evaluate this, we need to model this second-order bias, for example, by label propagation on the cooccurrence graph. We haven’t developed a full theory for this yet, but it is an exciting topic for future research.
* Optimizer or data sampling may affect the extent to which models acquire dataset bias. We aren’t aware of research on this, but it’s well known that SGD-trained models generalize better than Adam-trained models. Similar effects may hold for bias. To evaluate this, we need to train CLIP models from scratch, which is not within our compute budget.
* The model (together with its training, e.g., the loss) induces nonlinear dynamics. Fig. 8 and Fig. 9 clearly show nonlinear relationships of co-occurrence rates and model bias. We found Chebyshev polynomials improve the $R^2$ fit by 1 to 3 points (see App. G.3). However, we think modeling dataset bias beyond direct co-occurrences is the most interesting way to gain further insights into dataset-to-model bias transfer.

---

### Meta-Review · Area_Chair_ypif · 2026-01-06

**Summary:**

The reviewers’ main concerns centered on the reliability of automatically inferred demographic labels, the lack of human validation, the justification for choosing the captioning model, and the interpretation of bias transfer analyses (correlation vs. causation and linearity assumptions). Additional issues included privacy considerations, missing confidence intervals, and requests for deeper discussion of nonlinear bias transfer mechanisms. Importantly, none of the reviewers raised fundamental concerns about novelty or significance; the work was viewed as timely, large-scale, and valuable to the community, supporting acceptance.

**Reviewer Concerns:**

The rebuttal convincingly addressed the core concerns: human validation was added with strong agreement statistics; captioning model choice was quantitatively justified; confidence intervals and privacy safeguards were incorporated; and claims were carefully revised to distinguish correlation from causation. Additional experiments and discussion strengthened the bias transfer analysis beyond linear fits. No major technical or conceptual concerns remain outstanding; any remaining limitations (e.g., higher-order bias modeling or optimizer effects) are clearly acknowledged as future work rather than gaps undermining the current contribution.

**Reviewer Scores:**

The initial scores of 3 out of 4 reviewers were 6 (marginally above the acceptance threshold). Post rebuttal the reviewer who gave 4 mentioned that he has increased the score, indicating that all reviewers would have stayed at 6 score post rebuttal.

---

### Decision · Program_Chairs · 2026-01-26

Accept (Poster)